# Maintenance of p-eIF2α levels by the eIF2B complex is vital for colorectal cancer

Ivana Paskov Škapik[1,2,3,15], Chiara Giacomelli[4,15], Sarah Hahn[1,2,3], Hanna Deinlein[1,2], Peter Gallant[1], Mathias Diebold[1,5], Josep Biayna[1,2,6], Anne Hendricks[1,2], Leon Olimski[1,2], Christoph Otto[2], Carolin Kastner[2], Elmar Wolf[1,7], Christina Schülein-Völk[1], Katja Maurus[8], Andreas Rosenwald[8], Nikolai Schleussner[9,10,11,12], Rene-Filip Jackstadt[9,11,12], Nicolas Schlegel[2], Christoph-Thomas Germer[2,13], Martin Bushell[4,14], Martin Eilers[1,13], Stefanie Schmidt[1,2 ✉] & Armin Wiegering[1,2,3,13 ✉]

## Abstract

**Protein synthesis is an essential process, deregulated in multiple tumor types showing differential dependence on translation factors compared to untransformed tissue. We show that colorectal cancer (CRC) with loss-of-function mutation in the APC tumor suppressor depends on an oncogenic translation program regulated by the ability to sense phosphorylated eIF2α (p-eIF2α). Despite increased protein synthesis rates following APC loss, eIF2α phosphorylation, typically associated with translation inhibition, is enhanced in CRC. Elevated p-eIF2α, and its proper sensing by the decameric eIF2B complex, are essential to balance translation. Knockdown or mutation of eIF2Bα and eIF2Bδ, two eIF2B subunits responsible for sensing p-eIF2α, impairs CRC viability, demonstrating that the eIF2B/p-eIF2α nexus is vital for CRC. Specifically, the decameric eIF2B linked by two eIF2Bα subunits is critical for translating growth-promoting mRNAs which are induced upon APC loss. Depletion of eIF2Bα in APC-deficient murine and patient-derived organoids establishes a therapeutic window, validating eIF2Bα as a target for clinical intervention. In conclusion, we demonstrate how the expression of the oncogenic signature in CRC is crucially controlled at the translational level.**

**Keywords** APC; Colorectal Cancer; eIF2α; eIF2B; Translation
**Subject Categories** Cancer; Translation & Protein Quality

## Introduction

The initiation of mRNA translation is tightly regulated by a well-defined set of translation initiation factors. In unstressed conditions, two rate-limiting steps control canonical cap-dependent translation initiation; first, the recognition of the mRNA cap structure by the cap-binding complex, the eukaryotic initiation factor 4F (eIF4F) complex, and second, recycling of the eIF2/Met-tRNAi/GTP ternary complex (TC) by eIF2B (Hinnebusch and Lorsch, 2012; Merrick and Pavitt, 2018). TC availability is a key point on which a variety of stress signals converge to inhibit global protein synthesis when unfavorable conditions are present. Dependent on the different stress conditions, four kinases (PERK, PKR, HRI, and GCN2) phosphorylate the α-subunit of eIF2 at serine 51 (S51) via a cascade known as the integrated stress response (ISR) (Costa-Mattioli and Walter, 2020). Phosphorylated eIF2α in turn sequesters eIF2B, its own guanine nucleotide exchange factor (GEF), into an inactive form, thus reducing overall TC levels and decreasing translation initiation rates (Kashiwagi et al, 2019; Kenner et al, 2019; Wang and Proud, 2022).

Tumors in general, and CRC in particular, are characterized by altered rates of protein synthesis and differential dependence on specific translation factors (Fabbri et al, 2021; Knight et al, 2021; Schmidt et al, 2020; Schmidt et al, 2019; Wiegering et al, 2015). The ISR itself has received widespread attention in cancer, including CRC, with a variety of outcomes that depend on the activating kinase as well as the downstream pathways induced by phosphorylation of eIF2α (Koromilas, 2015; Schmidt et al, 2020). In previous work, we showed that colorectal tumors deficient for the tumor suppressor gene *APC* are peculiarly dependent on eIF2Bε, an essential subunit among the five subunits that form the eIF2B complex (Schmidt et al, 2019). Moderate depletion of eIF2Bε leads

[1]Theodor Boveri Institute, Biocenter, University of Würzburg, 97074 Würzburg, Germany. [2]Department of General, Visceral, Transplant, Vascular and Pediatric Surgery, University Hospital Würzburg, 97080 Würzburg, Germany. [3]Goethe University Frankfurt, University Hospital, Department of General, Visceral, Transplant and Thoracic Surgery, Frankfurt am Main, Germany. [4]CRUK Scotland Institute, Garscube Estate, Switchback Road, Glasgow G61 1BD, UK. [5]Institute of Pharmacy and Food Chemistry, University of Würzburg, 97074 Würzburg, Germany. [6]Institute of Cardiovascular Regeneration, Centre for Molecular Medicine, Goethe University Frankfurt, 60590 Frankfurt am Main, Germany. [7]Institute of Biochemistry, CAU Kiel, 24118 Kiel, Germany. [8]Institute of Pathology, University of Würzburg, 97074 Würzburg, Germany. [9]Heidelberg Institute for Stem Cell Technology and Experimental Medicine (HI-STEM gGmbH), Heidelberg, Germany. [10]Department of General, Visceral and Transplantation Surgery, University Hospital Heidelberg, University Heidelberg, 69120 Heidelberg, Germany. [11]Cancer Progression and Metastasis Group, German Cancer Research Center (DKFZ) and DKFZ-ZMBH Alliance, Heidelberg, Germany. [12]German Cancer Consortium (DKTK), DKFZ, Core Center Heidelberg, Heidelberg, Germany. [13]Comprehensive Cancer Center Mainfranken, University Hospital Würzburg, 97080 Würzburg, Germany. [14]School of Cancer Sciences, University of Glasgow, Garscube Estate, Switchback Road, Glasgow G61 1QH, UK. [15]These authors contributed equally: Ivana Paskov Škapik, Chiara Giacomelli. ✉E-mail: schmidt_s12@ukw.de; wiegering_a@ukw.de

to dephosphorylation of eIF2α and reduces CRC cell viability with a narrow therapeutic window. Transient inhibition of two of the four kinases responsible for eIF2α phosphorylation, GCN2, and PKR, mimics the phenotype. However, rapid compensatory mechanisms arise and limit the potential of targeting these kinases (Lehman et al, 2015; Schmidt et al, 2019; Tameire et al, 2019).

Here we show that mutations essential for CRC development drive an oncogenic translational program that is tightly regulated by eIF2B/p-eIF2α. Indeed, murine and patient CRC samples harbor elevated levels of p-eIF2α, which are required to balance the translation of this oncogenic signature. As eIF2B is the sensor and downstream target of p-eIF2α, we hypothesized a central role for this interaction in regulating proteostasis in CRC. Therefore, we modulated eIF2Bα and eIF2Bδ, the two subunits essential for sensing p-eIF2α, by depletion or mutation, and evaluated the effects on translational as well as cellular homeostasis. We reveal that the ability of the eIF2B complex to sense p-eIF2α is vital for CRC viability, and that, more specifically, eIF2Bα is required for translation of growth-promoting mRNAs transcriptionally upregulated upon loss of APC. Indeed, without eIF2Bα CRC cells are unable to translate the pro-survival transcriptional program induced by WNT pathway activation, resulting in impaired proliferation. Being the only non-essential subunit of the eIF2B complex (Elsby et al, 2011; Hannig and Hinnebusch, 1988), eIF2Bα targeting in APC-deficient murine intestinal as well as in patient-derived organoids opens a therapeutic window in comparison to wildtype organoids. Thus, we define a possible strategy to target CRC by interfering with the eIF2B/p-eIF2α interaction via eIF2Bα.

# Results

## CRC has increased levels of phosphorylated eIF2α

Protein synthesis is the most energy-consuming cell process and is tightly regulated to maintain homeostasis (Buttgereit and Brand, 1995). Many tumor types exhibit deregulated pathways that converge on translation factors to enhance mRNA translation, as rapid proliferation relies on the continuous delivery of newly synthesized proteins (Schmidt et al, 2020). Interestingly, we and others have described a concomitant increase in phosphorylation of eIF2α at serine 51 (S51), a canonical inhibitor of protein synthesis at the translation initiation stage, in several cancer types (Fabbri et al, 2021; Ghaddar et al, 2021; Schmidt et al, 2020; Schmidt et al, 2019). To investigate whether this holds true also in human CRC, we quantified p-eIF2α in a tissue microarray including samples from colon mucosa ($n = 84$), adenoma ($n = 34$) as well as colon carcinoma ($n = 54$). These samples displayed a clear increase in phosphorylation according to the disease stage (Fig. 1A,B). To validate this increase in eIF2α phosphorylation, we took advantage of genetically well-defined intestinal organoids established from wildtype (WT), $VillinCre^{ER}Apc^{-/-}$ (A), $VillinCre^{ER}Apc^{-/-}Kras^{G12D/+}$ (AK) and $Lgr5^{eGFP-creERT2}Apc^{-/-}Kras^{G12D/+}Tgfbr2^{-/-}Trp53^{-/-}$ (LAKTP) mice (el Marjou et al, 2004; Sansom et al, 2007; Shibata et al, 1997; Tauriello et al, 2018). Phosphorylation of eIF2α was enhanced in A, AK and LAKTP compared to WT organoids (Fig. 1C), which correlates with previously published data showing that intestines of mice deleted for *Apc* (and mutated for *Kras*) have higher levels of p-eIF2α in comparison to WT mice (Schmidt et al,

2019). Puromycin incorporation assays were performed in the same organoid lines to quantify global translation rates. Interestingly, an increase in bulk translation in mutated organoids compared to WT ones was detected (Fig. 1D). Having confirmed that CRC inexplicably displays concomitant high expression of a translation inhibitory signal and also increased protein synthesis, we hypothesized that sensing of p-eIF2α by the eIF2B complex could be the key point deregulated in this process.

The complete eIF2B complex is a decamer composed of two heterotetramers of the four eIF2Bβγδε subunits interacting via an eIF2Bα homodimer (Kashiwagi et al, 2019; Kenner et al, 2019). The tool compound ISR inhibitor (ISRIB) has been shown to render eIF2B insensitive to the phosphorylated status of eIF2α (Kenner et al, 2019; Sidrauski et al, 2013; Sidrauski et al, 2015; Tsai et al, 2018; Zyryanova et al, 2021; Zyryanova et al, 2018). Thus, to investigate whether this drug could functionally regulate the eIF2B/p-eIF2α nexus in CRC, we treated the APC-mutated CRC cell line SW480 with increasing concentrations of ISRIB. While tunicamycin (TM) induced ATF4 expression, as well as downstream transcriptional targets (*ATF3*, *DDIT3* (CHOP)), this was prevented when cells were treated with ISRIB, but no difference was observed in TM non-stressed cells (Appendix Fig. S1A,B). Accordingly, we did not observe any reduction in cell proliferation with ISRIB treatment (Fig. 1E). To confirm this, we replicated the experiments in murine small intestinal organoids carrying the most common mutations in CRC (A, AK, LAKTP), demonstrating that up to 1 μM of ISRIB does not cause a significant effect on cell viability, but reduces TM-induced ISR signaling at low concentrations as shown for LAKTP organoids (Fig. 1F,G; Appendix Fig. S1C). To underscore these findings in a human system, we made use of two CRC patient-derived organoids (PDOs; T4 and HD-3) as well as one established from a familial adenomatous polyposis patient (FAP) (Kastner et al, 2021; Schmidt et al, 2019). PDOs T4 and HD-3 are both *APC*- and *PIK3CA*-mutated and show comparable growth and phenotypic characteristics (Schmidt et al, 2019) (Appendix Table S1). ISRIB treatment did not reduce the viability of any of the three PDOs (Fig. 1F,G), suggesting that the canonical targeting of the eIF2B/p-eIF2α axis via ISRIB might not be a reasonable therapeutic approach in the treatment of CRC.

## Depletion of eIF2Bα and eIF2Bδ differentially affects cellular and translational homeostasis

As ISRIB did not affect CRC viability, we investigated the relevance of the eIF2B complex more closely. Previously, we demonstrated that a moderate depletion of eIF2Bε leads to loss of eIF2α phosphorylation with subsequent induction of MYC-mediated apoptosis in CRC cells (Schmidt et al, 2019). To define the impact of the other four subunits (eIF2Bαβγδ) on CRC cell viability, global translation rates as well as the phosphorylation status of eIF2α, we depleted eIF2Bαβγδ via shRNAs in SW480 cells. All used shRNAs induced a strong knockdown of the respective subunit at both protein and mRNA level (Figs. 2A and EV1A). Interestingly, while depletion of eIF2Bγδ decreased the protein levels of all other subunits to different degrees, and depletion of eIF2Bβ decreased eIF2Bα levels, knockdown of eIF2Bα did not alter protein levels of the other subunits (Fig. 2A). This is in line with previous reports showing that eIF2Bβγδε subunits stabilize each other to ensure the correct stoichiometry of the eIF2B complex (Wang et al, 2012;

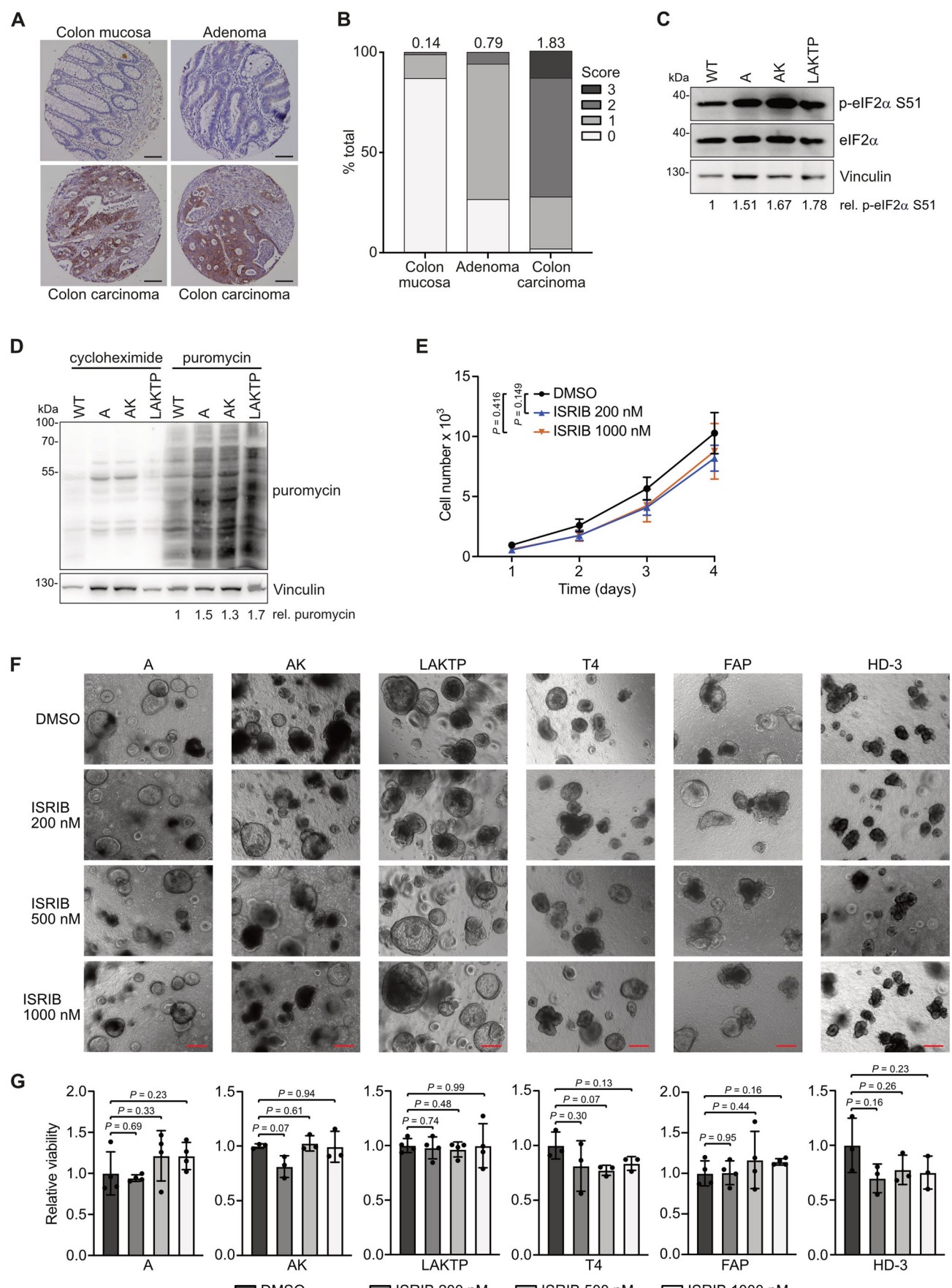

◄ **Figure 1.  CRC has increased levels of phosphorylated eIF2α.**

(A) Immunohistochemical (IHC) staining for p-eIF2α S51 of a tissue microarray from colon mucosa ($n = 84$), adenoma ($n = 34$) and colon carcinoma ($n = 54$). The IHC pictures are representative of the respective samples. Scale bar = 100 μM. (B) Quantification of p-eIF2α S51 IHC described in (A). Staining intensity scores are displayed; 0 = negative, 1 = weak, 2 = intermediate, 3 = strong. Above each stage, the mean of p-eIF2α S51 staining intensity is given. (C) Western blot of indicated proteins in murine WT, A, AK, and LAKTP intestinal organoids, representative of three biological replicates with similar results. Levels of p-eIF2α S51, relative to total eIF2α and normalized to vinculin, are given below the blot. (D) Protein synthesis in murine WT, A, AK, and LAKTP intestinal organoids analyzed by puromycin incorporation. As control, organoids were treated with cycloheximide (50 μg/ml) to inhibit protein synthesis. Western blot is representative of two biological replicates with similar results. Puromycin incorporation, relative to vinculin, is shown below. (E) Growth curve of SW480 cells treated with indicated concentrations of ISRIB or DMSO as control for 4 days. Cell number was analyzed with the Operetta screening microscope. Data show mean ± s.d. ($n = 3$ biological replicates); Student's *t* test. (F) Pictures of A, AK, LAKTP murine intestinal organoids and T4, FAP, HD-3 PDOs treated with indicated concentrations of ISRIB for 5 days, representative of three to four biological replicates with similar results. Scale bar = 200 μm. (G) Relative viability of A, AK, LAKTP murine intestinal organoids and T4, FAP, HD-3 PDOs treated as described in (F). Data show mean ± s.d. ($n = 3$ or 4 biological replicates); Student's *t* test. Source data are available online for this figure.

Wortham et al, 2016). Importantly, eIF2Bα depletion does not affect global translation rates, suggesting that its loss leaves eIF2Bβγδε heterotetramers that retain translation initiation activity (Fig. 2B) (Elsby et al, 2011). In contrast, knockdown of eIF2Bβγδ led to a reduction of global protein synthesis rates, suggesting a disrupted eIF2B complex (Fig. 2A,B). Formation of the full decameric eIF2B complex is a prerequisite for interaction with p-eIF2α as its binding site lies between eIF2Bα and eIF2Bδ. Intriguingly, eIF2Bαβγ knockdown was associated with a reduction in p-eIF2α levels and induction of the ISR protein CHOP (Fig. 2C). On a cellular level, depletion of eIF2Bαβγδ significantly decreased cell numbers (Fig. 2D). Whilst this was associated with strong cell death induction upon knockdown of eIF2Bβγδ, depletion of eIF2Bα elicited only a moderate response (Fig. 2E,F). Instead, cells depleted of eIF2Bα were characterized by a prolonged G1 cell cycle phase of 21.6 h compared to all other conditions that showed a G1 length between 7 and 12 h similar to previously observed G1 length of CRC cells (Fig. 2F,G) (Peter et al, 2014). To confirm the importance of the eIF2B complex for CRC, we replicated these results in a second APC-deficient cell line, namely DLD1 cells. Comparable to the effects observed in SW480 cells, depletion of eIF2Bα did not alter protein levels of the other eIF2B subunits, but knockdown of eIF2Bδ reduced those levels (Fig. EV1B). Accordingly, while eIF2Bδ knockdown induced high percentage of cell death, eIF2Bα depletion affected cell cycle progression (Fig. EV1C–E).

## Disrupting the eIF2Bα homodimer is detrimental for CRC cells

While knockdown of eIF2Bβγδ subunits disrupts the entire eIF2B complex, reduces overall translation rates and induces apoptosis, depletion of eIF2Bα does not affect global protein synthesis but still decreases CRC viability. The eIF2Bα homodimer is essential for the formation of the decameric eIF2B complex and, thus, a prerequisite for sensing eIF2α phosphorylation (Kashiwagi et al, 2019; Kenner et al, 2019). Having shown that loss of eIF2Bα strongly decreases p-eIF2α levels, we investigated whether this effect is exclusively due to the inability of eIF2B to form a decameric complex upon knockdown of eIF2Bα and not due to lowered eIF2Bα levels, thereby excluding any potential moonlighting function. Therefore, we sought to disrupt eIF2B decamer formation without reducing eIF2Bα levels by modulating the eIF2Bα homodimer interface. To identify possible mutations in eIF2Bα that would disrupt homo-dimerization, a cryo-EM structure (PDB: 6O81) (Kenner et al,

2019) was analyzed using the Molecular Operating Environment (MOE 2019) (ULC CCG, 2019) (Fig. 3A). Residues Y185 and E188 were identified to be involved in the strongest stabilizing interactions of the eIF2Bα homodimer. Two additional potential mutation sites, A181 and C218, revealed a high disruption potential as analyzed by visual inspection of the structure. We generated a mutant of eIF2Bα containing four mutations in these residues (eIF2Bα^mut: Y185A, E188A, A181F, C218F) as well as a silent mutation in the *EIF2B1* shRNA recognition site. The eIF2Bα WT form (eIF2Bα^wt) only harbored the mutation in the shRNA target site. The two exogenous eIF2Bα forms showed expression at the protein level, and introduction of sh*EIF2B1* reduced levels of the endogenous eIF2Bα only (Fig. 3B,C). Cells depleted of endogenous eIF2Bα (empty), and cells with expression of eIF2Bα^mut displayed a decrease in p-eIF2α as well as a pronounced increase in CHOP levels (Fig. 3D,E). Conversely, cells expressing eIF2Bα^wt after eIF2Bα knockdown did not display these changes in p-eIF2α and CHOP levels (Fig. 3D,E). Phenotypically, the reduced proliferation as well as the moderate increase in dead cells upon knockdown of eIF2Bα was rescued by stable expression of eIF2Bα^wt, whereas eIF2Bα^mut was not able to revert this phenotype (Fig. 3F,G; Appendix Fig. S2A). The effects of eIF2Bα depletion and mutation are in contrast to ISRIB treatment, which did not induce a reduction in cell viability as analyzed in parallel (Figs. 1E and 3F). Enhanced levels of eIF2α phosphorylation have also been described in other solid tumor entities (e.g., lung cancer, PDAC, breast cancer) pointing to the possibility that modulation of the eIF2B complex might induce a vulnerability in other types of cancer (Bai et al, 2021; Ghaddar et al, 2021; Koromilas, 2015; Shin et al, 2022). We tested this by using the eIF2Bα^mut overexpression approach in a PDAC cell line, PaTu8988T, showing a similarly reduced viability upon depletion of eIF2Bα, which was rescued by eIF2Bα^wt but not eIF2Bα^mut overexpression (Appendix Fig. S2B,C). Likewise, eIF2Bα knockdown reduced p-eIF2α levels which was also apparent in eIF2Bα^mut but not in eIF2Bα^wt PaTu8999 cells (Appendix Fig. S2B).

To investigate the role of p-eIF2α more precisely in the context of eIF2Bα depletion in CRC, we made use of two well-established mutants: the phospho-dead eIF2α S51A and the phospho-mimicking eIF2α S51D mutants (Donze et al, 1995; Perkins and Barber, 2004; Scheuner et al, 2006; Scheuner et al, 2001). We stably overexpressed HA-tagged eIF2α^wt, eIF2α^S51A, or eIF2α^S51D in SW480 cells and depleted eIF2Bα in parallel. Although phosphorylation of both endogenous eIF2α and exogenous HA-eIF2α^wt was reduced upon depletion of eIF2Bα in eIF2α^wt expressing cells, overall eIF2α phosphorylation was still higher compared to eIF2Bα-depleted

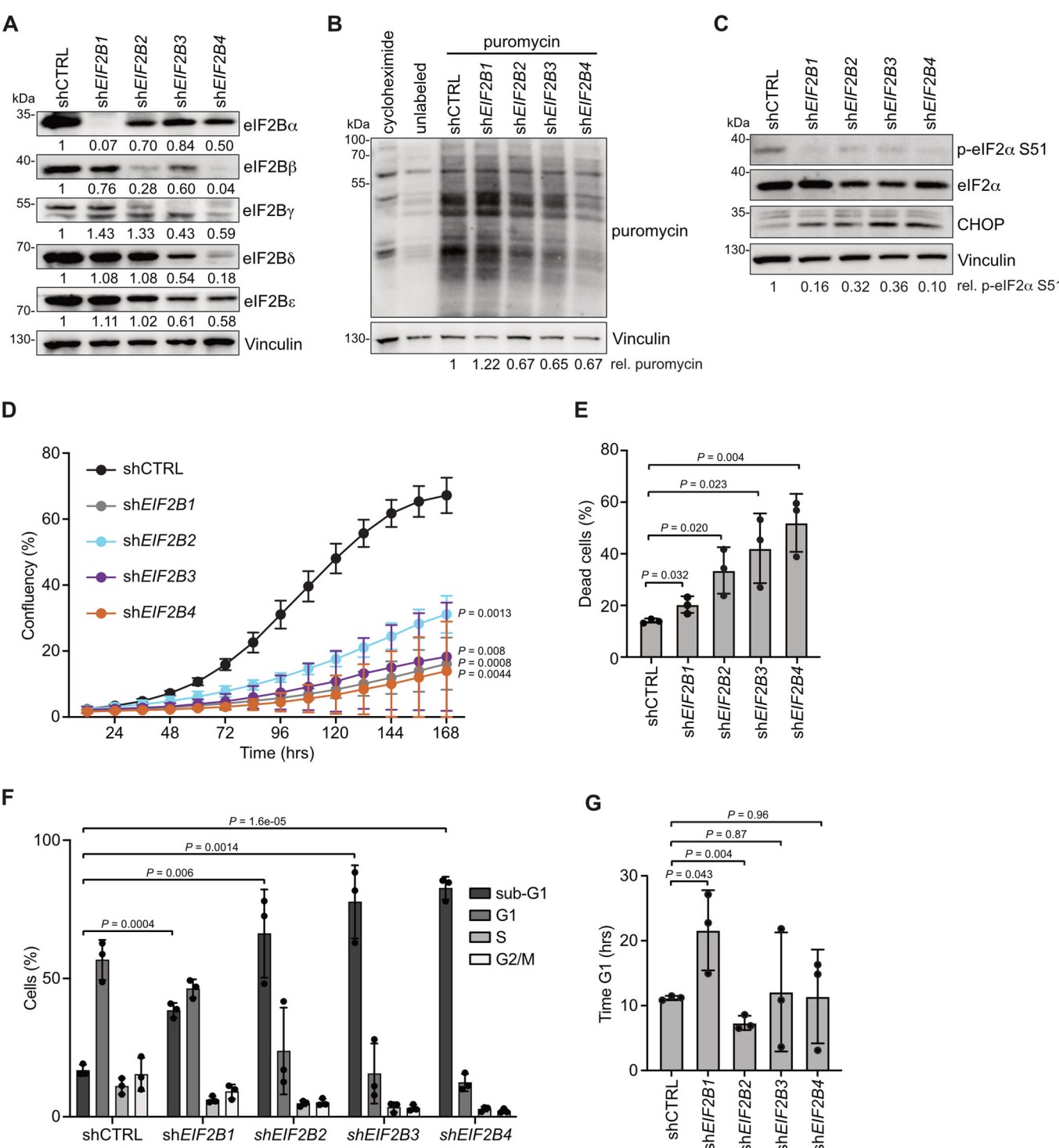

empty cells (Fig. EV2A). Nevertheless, this partial restoration of p-eIF2α levels could not rescue the impaired cell proliferation elicited by eIF2Bα knockdown (Fig. EV2B). Exogenous HA-eIF2α$^{S51A}$ cannot be thoroughly phosphorylated (Fig. EV2A), and it competes with endogenous eIF2α to generate phospho-dead eIF2α$^{S51A}$-containing eIF2/eIF2B complexes over time. Still, this did not impact on cellular viability, either with or without eIF2Bα being present (Fig. EV2B).

Likewise, overexpression of the phospho-mimicking S51D mutant did not revert the viability defect (Fig. EV2C,D). In an additional approach for testing the impact of p-eIF2α on cell viability in the context of eIF2Bα mutation, we depleted GADD34 (encoded by *PPP1R15A*), the inducible regulatory subunit of PP1, which is the relevant phosphatase for p-eIF2α (Choy et al, 2015; Harding et al, 2009). Phosphorylation of eIF2α was restored upon siRNA-mediated depletion of GADD34 confirming that GADD34

**Figure 2. Depletion of eIF2Bα and eIF2Bδ differentially affects cellular and translational homeostasis.**

(A) Western blot of indicated proteins in SW480 cells transduced with shCTRL or shRNAs against *EIF2B1-4*, representative of three biological replicates with similar results. Levels of the respective eIF2B subunits, relative to vinculin, are given below each corresponding panel. (B) Protein synthesis in SW480 cells transduced as described in (A) analyzed by puromycin incorporation. As control, cells were treated with cycloheximide (50 μg/ml) to inhibit protein synthesis. The western blot is representative of three biological replicates with similar results. Levels of puromycin incorporation, relative to vinculin, are given below the blot. (C) Western blot of indicated proteins in SW480 cells transduced as described in (A), representative of three biological replicates with similar results. Levels of p-eIF2α S51, relative to total eIF2α and normalized to vinculin, are given below the blot. (D) Growth curve of SW480 cells transduced as described in (A), measured with Incucyte® live-cell imaging system. Data show mean ± s.d. ($n = 3$ biological replicates); Student's $t$ test. (E) Annexin V/PI FACS analysis of SW480 cells transduced as described in (A). Data show mean ± s.d. ($n = 3$ biological replicates). Student's $t$ test. (F) PI cell cycle FACS analysis of SW480 cells transduced as described in (A). Data show mean ± s.d. ($n = 3$ biological replicates); Student's $t$ test. (G) Length of G1 cell cycle phase of SW480 cells transduced as described in (A), calculated with data acquired from growth curve (D) and PI cell cycle FACS (F). Data show mean ± s.e.m. ($n = 3$ biological replicates); Student's $t$ test. Source data are available online for this figure.

promotes eIF2α dephosphorylation in a context when p-eIF2α is not bound by the eIF2B complex (Fig. EV2E,F). Nevertheless, restoration of p-eIF2α levels by knockdown of GADD34 did not rescue the viability defect elicited by either eIF2Bα depletion or by eIF2Bα mutation (Fig. EV2G,H). These data argue that the levels of p-eIF2α themselves are not the driver for the observed cellular phenotype but rather the inability of the disrupted eIF2B complex to sense p-eIF2α.

To investigate whether the reduction of p-eIF2α levels as well as the proliferation defect upon knockdown of eIF2Bα with or without eIF2Bα$^{mut}$ expression is due to the inability of eIF2Bα$^{mut}$ to form an eIF2B decamer, we analyzed protein complexes by sedimentation on 10–30% sucrose gradients. Samples depleted of eIF2Bα displayed a shift from higher to lower molecular weight fractions, which is consistent with eIF2Bβγδε heterotetramer formation (Appendix Fig. S3A–C). Whereas eIF2Bα$^{wt}$ expression in eIF2Bα-depleted cells rescued eIF2B decamer formation, eIF2Bα$^{mut}$ expression did not, and, thus, is not able to form the eIF2B decamer (Appendix Fig. S3A–C). Moreover, levels of p-eIF2α were lower in eIF2Bα-depleted samples without or with eIF2Bα$^{mut}$ expression compared to control as well as eIF2Bα$^{wt}$ samples with restored eIF2B decamer formation (Appendix Fig. S3A,D). Having shown that eIF2Bα$^{mut}$ cannot assemble the full eIF2B complex, we wanted to directly elucidate whether this in turn leads to an inability to interact with the eIF2 complex. Therefore, we overexpressed HA-tagged versions of eIF2Bα$^{wt}$ and eIF2Bα$^{mut}$ and performed co-immunoprecipitations with endogenous eIF2α. Whereas eIF2Bα$^{wt}$ could be co-immunoprecipitated with endogenous eIF2α, eIF2Bα$^{mut}$ did not show an interaction (Fig. 3H), clearly establishing that the eIF2 complex does not associate with the disrupted eIF2B decamer. In summary, disruption of the decameric eIF2B complex and thus the cells' ability to sense p-eIF2α reveals a weakness in CRC that can be targeted by eIF2Bα modulation.

## The ability to sense eIF2α phosphorylation is essential for CRC cell survival

The eIF2Bδ subunit of the intact eIF2B complex directly binds p-eIF2α, thereby sensing the phosphorylation of eIF2α. Specifically, when bound to eIF2B, p-eIF2α is buried between eIF2Bα and eIF2Bδ, and hydrophobic residues in eIF2Bδ mediate the interaction with p-eIF2α (Kenner et al, 2019). Therefore, to validate the essentiality of this sensing feature for CRC, we hypothesized that disrupting the interface between eIF2Bδ and p-eIF2α would render the cells incapable of sensing p-eIF2α and would negatively affect CRC viability. We examined the corresponding cryo-EM structure

in more detail (PDB: 6O9Z) and reasoned that mutation of two alanine residues in eIF2Bδ, A315 and A318, as has been published (Kenner et al, 2019), to amino acids with large and bulky side chains, had the highest potential to interfere with eIF2Bδ/p-eIF2α interaction (Fig. 4A). Thus, we stably expressed an eIF2Bδ mutant (eIF2Bδ$^{mut}$) construct, harboring A315W and A318W mutations, or an eIF2Bδ WT construct (eIF2Bδ$^{wt}$) and, in parallel, depleted endogenous eIF2Bδ via shRNA. Both constructs contained silent mutations for the respective *EIF2B4* shRNA target site preventing depletion of the exogenous proteins. Western blot and qPCR analysis showed comparable levels of expression of both eIF2Bδ$^{mut}$ and eIF2Bδ$^{wt}$ constructs as well as knockdown of the endogenous eIF2Bδ only upon shRNA transduction (Fig. 4B,C). On a molecular level, depletion of eIF2Bδ led to a decrease of p-eIF2α levels as well as an induction of CHOP (Figs. 2C and 4D,E). Whereas expression of eIF2Bδ$^{mut}$ did not rescue the decrease in p-eIF2α as well as only partially rescued the induction of CHOP upon eIF2Bδ knockdown, expression of eIF2Bδ$^{wt}$ restored both p-eIF2α and CHOP levels to control levels (Fig. 4D,E). The decreased CRC cell numbers upon knockdown of eIF2Bδ, which were associated with strong induction of cell death, could be reverted partially by expression of the eIF2Bδ$^{wt}$ construct (Figs. 4F,G and EV3A). In contrast to eIF2Bδ knockdown alone, parallel expression of eIF2Bδ$^{mut}$ induced moderate levels of cell death comparable to the effects of eIF2Bα depletion (Figs. 2E, 4G, and EV3A). The strong decrease in viability upon eIF2Bδ depletion was associated with diminished global translation (Figs. 2B and EV3B). This was rescued in both eIF2Bδ$^{mut}$ and eIF2Bδ$^{wt}$ expressing cells suggesting that a failure in sensing p-eIF2α by eIF2B has a different impact on cellular and translational homeostasis than disruption of eIF2B complex formation itself (Fig. EV3B).

To confirm that the observed phenotypic and biochemical features of eIF2Bδ$^{mut}$ are exclusively a consequence of an inability to sense p-eIF2α, and not due to interference with eIF2B decameric complex formation, we analyzed eIF2B complex formation by sedimentation on sucrose gradients. Samples depleted of eIF2Bδ displayed a shift from higher to lower molecular weight fractions, reflecting a disrupted eIF2B complex (Appendix Fig. S4A–C). Depletion of endogenous eIF2Bδ and parallel expression of eIF2Bδ$^{mut}$ or eIF2Bδ$^{wt}$ rescued complete eIF2B complex formation (Appendix Fig. S4A–C), which is in concordance with the observed restored global translation rates (Fig. EV3B). We observed a reduction in phosphorylated eIF2α in cells depleted of endogenous eIF2Bδ without and with expression of eIF2Bδ$^{mut}$ compared to non-depleted cells (Appendix Fig. S4A,D). In contrast, higher p-eIF2α levels were detected in the same fractions as the eIF2B decamer in

The EMBO Journal

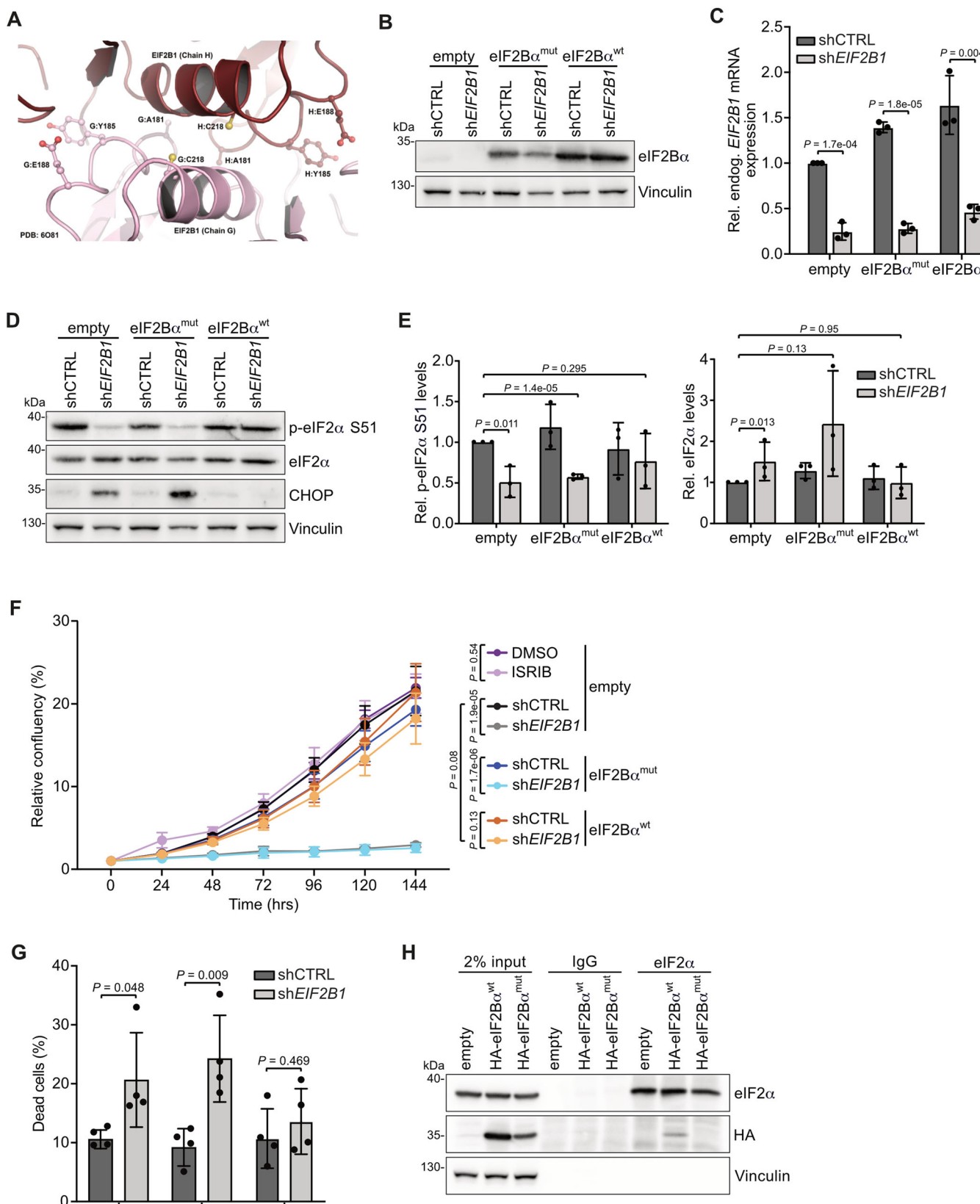

◀  **Figure 3.  Disrupting the eIF2Bα homodimer is detrimental for CRC cells.**

(A) Cryo-EM structure of the eIF2Bα homodimer (PDB: 6O81) with four amino acids of each eIF2Bα monomer, A181, Y185, E188 and C218, shown in detail. (B) Western blot of indicated proteins in shCTRL- or sh*EIF2B1*-transduced SW480 cells stably overexpressing eIF2Bα mutant (eIF2Bα$^{mut}$), eIF2Bα WT (eIF2Bα$^{wt}$) construct, or without any overexpression (empty). The western blot is representative of three biological replicates with similar results. (C) mRNA expression of endogenous *EIF2B1* in SW480 cells transduced as described in (B). Primers targeting the *EIF2B1* 3' UTR were used. Data show mean ± s.d. ($n = 3$ biological replicates); Student's *t* test. (D) Western blot of indicated proteins in SW480 cells transduced as described in (B), representative of three biological replicates. (E) Quantification of p-eIF2α S51 levels and total eIF2α levels, normalized to vinculin, of western blots described in (D). Data show mean ± s.d. ($n = 3$ biological replicates); Student's *t* test. (F) Growth curve of SW480 cells transduced as described in (B) or treated with 1000 nM ISRIB for 7 days (DMSO as control), measured with Incucyte® live-cell imaging system. Data show mean ± s.d. ($n = 6$ biological replicates); Student's *t* test. (G) Annexin V/PI FACS analysis of SW480 cells transduced as described in (B). Data show mean ± s.d. ($n = 4$ biological replicates); Student's *t* test. (H) Immunoprecipitation of eIF2α in SW480 cells stably overexpressing HA-tagged eIF2Bα$^{wt}$ or eIF2Bα$^{mut}$. As input, 2% of lysate was loaded. Co-immunoprecipitated HA-eIF2Bα constructs were detected by western blot. The western blot is representative of four biological replicates with similar results. Source data are available online for this figure.

the non-depleted conditions as well as in cells expressing eIF2Bδ$^{wt}$ arguing that, indeed, eIF2Bδ$^{mut}$ cannot bind p-eIF2α (Appendix Fig. S4A,D). Instead, both eIF2Bδ$^{wt}$ and eIF2Bδ$^{mut}$ are still able to bind total eIF2α as shown in a co-immunoprecipitation, correlating with restoration of a functional eIF2B/eIF2 complex (Fig. 4H).

To exclude the possibility that the GEF activity of eIF2B plays a role in reduction of eIF2α phosphorylation upon eIF2B complex disruption, we overexpressed HA-tagged versions of either wildtype eIF2Bε (eIF2Bε$^{wt}$) or an eIF2Bε R113H mutant (eIF2Bε$^{R113H}$), which is known to have reduced GEF activity (Fogli et al, 2004; Li et al, 2004). We depleted endogenous eIF2Bε in parallel and analyzed p-eIF2α levels as well as cell viability. As observed previously, knockdown of eIF2Bε led to decreased p-eIF2α levels and cell numbers (Schmidt et al, 2019) (Appendix Fig. S5A–D). Expression of both eIF2Bε$^{wt}$ and eIF2Bε$^{R113H}$ restored the phosphorylation of eIF2α and the viability defect (Appendix Fig. S5A–D), suggesting that the GEF function of the eIF2B complex is not involved in the reduction of p-eIF2α but it is solely dependent on disruption of the eIF2B complex. In conclusion, interfering with the function of the eIF2B complex to sense phosphorylation of eIF2α by either mutating eIF2Bδ or modulating eIF2Bα has comparable effects, and it induces a specific vulnerability in CRC.

## eIF2Bα is essential for the translation of growth-promoting mRNAs in CRC

We established that the general ability for sensing phosphorylation of eIF2α by the eIF2B complex is a critical feature for the proliferation and survival of CRC cells. There are fundamental phenotypic differences depending on how the inability for sensing p-eIF2α is achieved: while modulation (knockdown or mutation) of eIF2Bα and mutation of eIF2Bδ had comparable cellular and molecular effects, loss of one of the eIF2Bβγδ subunits had different outcomes. Therefore, to explore these differences in more detail we investigated the effects of eIF2Bα and eIF2Bδ knockdown on translation on a global scale. To achieve this, we performed ribosome profiling experiments (Ribo-seq) in SW480 cells transduced with doxycycline (DOX)-inducible shRNAs against eIF2Bα and eIF2Bδ in four independently repeated experiments. Similar to the previously observed effects with constitutive shRNAs, depletion of eIF2Bα and eIF2Bδ led to induction of the ISR marker ATF4 and reduced cell numbers (Fig. EV4A,B). Quality control of ribosome-protected fragments (RPFs) shows read lengths peaking at 30–31 nt, predominant alignment to coding sequence (CDS) and a single frame preference indicating

reliable RPFs enrichment (Fig. EV4C). In principal component analysis (PCA), non-targeting control samples (shCTRL) grouped together with uninduced (-DOX) sh*EIF2B1* and sh*EIF2B4* samples for total mRNA and RPFs aligned to the coding sequences, whilst induced ( + DOX) shRNA samples clearly clustered differentially (Fig. EV4D). This was true also for RPFs aligned to 5' UTRs, albeit in a less obvious manner (Fig. EV4D). Having established a reliable Ribo-seq dataset, we extracted RPFs reads of length 28 to 33 nt and proceeded to look at their distribution across transcripts. Occupancy of 5' UTRs increased with knockdown of both eIF2Bα and eIF2Bδ, but not upon shCTRL induction, suggesting increased usage of upstream open reading frame (uORF) (Fig. 5A). This is consistent with a reduction of available TC after knockdown of eIF2Bα and eIF2Bδ and with subsequent induction of ATF4 translation (Fig. EV4A; Appendix Fig. S6A). In turn, ATF4 induction increased *DDIT3* (encoding CHOP) and *PPP1R15A* (encoding GADD34) both at the RPFs and total RNA level (Appendix Fig. S6B,C).

Globally, a higher number of genes was regulated upon depletion of eIF2Bα, mostly at the RPFs level, compared to eIF2Bδ-depleted samples, where more genes were affected at the transcriptional level (Fig. 5B). The majority of genes translationally regulated by eIF2Bδ knockdown overlap with those changing upon eIF2Bα knockdown (Fig. 5C). In contrast, depletion of eIF2Bα regulated a much higher number of genes exclusively, hinting that it is able to drive specific translational programs (Fig. 5C). To assess the functional consequences of the differentially regulated expression programs upon individual subunit depletion, we calculated the log2 translation efficiency (TE) (log2 RPFs – log2 total RNA) for each gene and performed gene set enrichment analysis (GSEA) using log2TE ranking. This analysis showed that mRNAs involved in oncogenic pathways associated with cell proliferation and growth were translationally downregulated by knockdown of eIF2Bα specifically, whereas this is not observed upon knockdown of eIF2Bδ (Fig. 5D; Appendix Fig. S6D–G). In contrast, among others, genes belonging to gene sets 'TNFα via NFκB', which include those involved in apoptosis induction, showed a higher TE in both eIF2Bα- and eIF2Bδ-depleted samples (Fig. 5D). Whilst the TE calculation takes into account both the total RNA level changes as well as the RPFs, we wondered whether these pathways were specifically regulated by a change in ribosome occupancy of unchanged transcriptional programs. Thus, we performed over-representation analysis (ORA) of transcripts downregulated at the RPFs level exclusively by eIF2Bα or eIF2Bδ knockdown (1353 and 78, respectively) using the MSigDB hallmark gene set collection.

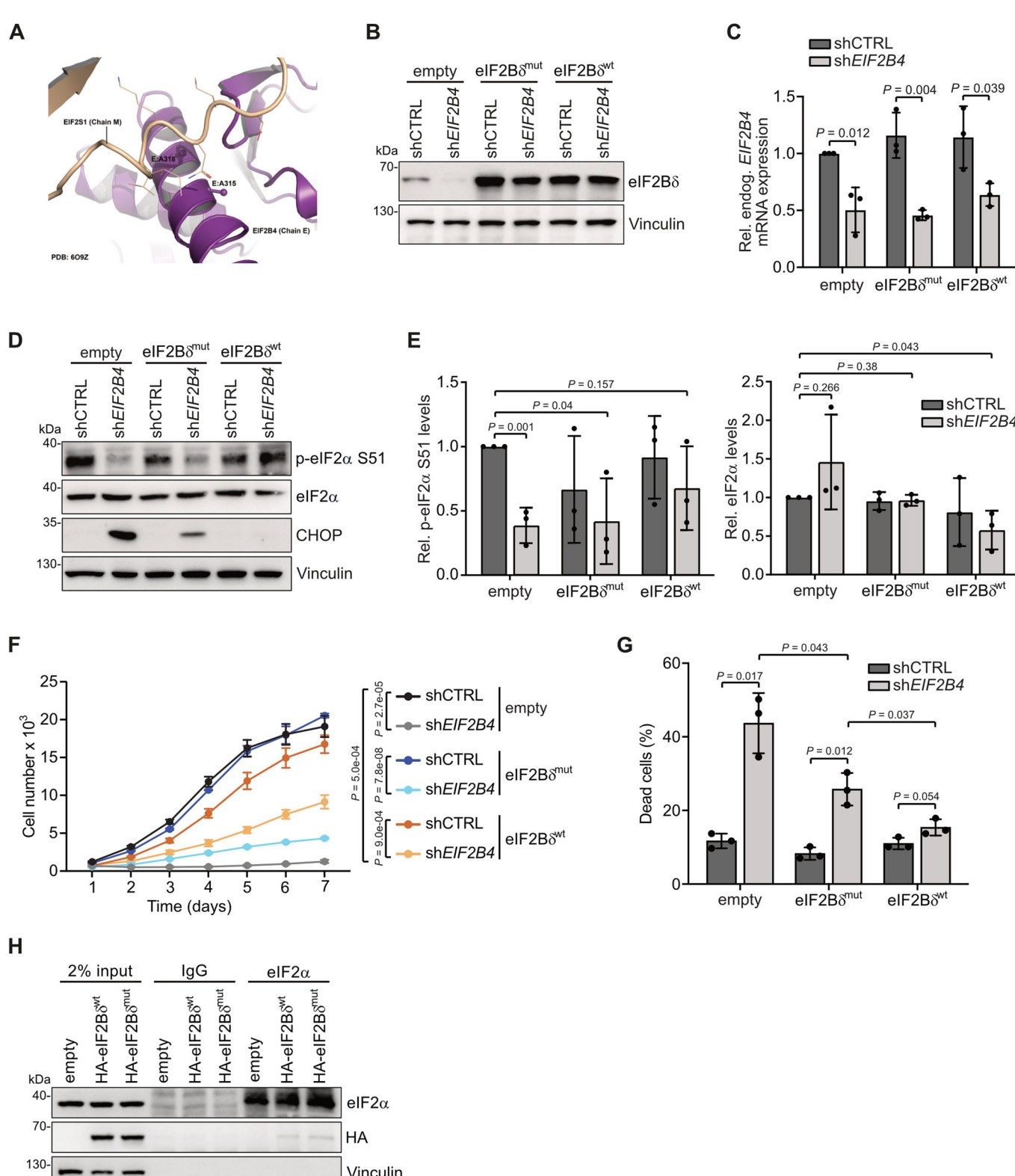

This validated that more oncogenic signaling pathways ("MYC Targets V1", "E2F Targets", "G2M Checkpoint", "MTORC1 Signaling") were translationally repressed upon eIF2Bα compared to eIF2Bδ depletion (Fig. 5E). In previous work, we used APC-mutated SW480 cells (APC$^{def}$), engineered to express doxycycline-inducible full-length APC (APC$^{res}$), and performed Ribo-seq of these two conditions (Schmidt et al, 2019). Strikingly, GSEA of these data showed that the gene sets repressed by depletion of eIF2Bα are the same gene sets found transcriptionally induced in APC$^{def}$ cells (Fig. 5F) (Schmidt et al, 2019).

**Figure 4. The ability of eIF2Bδ to sense eIF2α phosphorylation is essential for CRC cell survival.**

(A) Cryo-EM structure of eIF2Bδ in complex with p-eIF2α (PDB: 6O9Z) with two amino acids of eIF2Bδ, A315 and A318, shown in detail. (B) Western blot of indicated proteins in shCTRL- or sh*EIF2B4*-transduced SW480 cells stably overexpressing eIF2Bδ mutant (eIF2Bδ^mut), eIF2Bδ WT (eIF2Bδ^wt) construct, or without any overexpression (empty). The western blot is representative of three biological replicates with similar results. (C) mRNA expression of endogenous *EIF2B4* in SW480 cells transduced as described in (B). Primers targeting the *EIF2B4* 3′ UTR were used. Data show mean ± s.d. ($n = 3$ biological replicates); Student's $t$ test. (D) Western blot of indicated proteins in SW480 cells transduced as described in (B), representative of three biological replicates with similar results. (E) Quantification of p-eIF2α S51 levels and total eIF2α levels, normalized to vinculin, of western blots described in (D). Data show mean ± s.d. ($n = 3$ biological replicates); Student's $t$ test. (F) Growth curve of SW480 cells transduced as described in (B), measured with Operetta screening microscope. Data show mean ± s.e.m. ($n = 3$ biological replicates); Student's $t$ test. (G) Annexin V/PI FACS analysis of SW480 cells transduced as described in (B). Data show mean ± s.d. ($n = 3$ biological replicates). Student's $t$ test. (H) Immunoprecipitation of eIF2α in SW480 cells stably overexpressing HA-tagged eIF2Bδ^wt or eIF2Bδ^mut. As input, 2% of lysate was loaded. Co-immunoprecipitated HA-eIF2Bδ constructs were detected by western blot. The western blot is representative of three biological replicates with similar results. Source data are available online for this figure.

We already showed that typical ISR genes (*DDIT3*, *PPP1R15A*) are among the translationally upregulated genes upon eIF2Bα and eIF2Bδ depletion (Appendix Fig. S6B,C). Both CHOP and GADD34 are also upregulated on protein level with eIF2Bα^mut and eIF2Bδ^mut expression (Figs. 3D and 4D; Appendix Fig. S7A,B). In addition, we analyzed the Ribo-seq data in more detail and found *CBX4*, an ISR-induced gene (Lee et al, 2008; Sikalidis et al, 2011), as the most upregulated gene at the RPF level upon depletion of eIF2Bδ. Induction of CBX4 was also apparent in eIF2Bδ-depleted eIF2Bδ^mut expressing cells, but not in eIF2Bδ^wt cells (Appendix Fig. S7A). Thus, we assume that ISR activation contributes to the apoptotic phenotype observed upon eIF2Bδ modulation. To also relate the effects observed upon depletion and mutation of eIF2Bα, we focused on hallmarks gene sets which were translationally-repressed (Fig. 5D,E), selecting genes which were preferentially downregulated by sh*EIF2B1* only. Among these, the HNRNP group of proteins, RNA-binding proteins mainly involved in RNA metabolism, was highly represented. Indeed, HNRNPD and HNRNPA3 were decreased by both depletion and mutation of eIF2Bα, but levels were restored with eIF2Bα^wt (Appendix Fig. S7C). Several HNRNP proteins are involved in p-eIF2α-mediated stress granule formation, and HNRNPD, among others, is found to be down-regulated in eIF2B-mutant vanishing white matter (VWM) patient samples (Huyghe et al, 2012; Wu et al, 2014; Zhou et al, 2024).

Thus, we showed that knockdown of eIF2Bα and eIF2Bδ affects different sets of genes dependent on the level of remaining eIF2B complex activity, and only eIF2Bα is capable of regulating specific translational programs without affecting bulk protein synthesis. Importantly, our data suggest that intact eIF2B, capable of sensing p-eIF2α, is required for translation of mRNAs whose expression is induced upon APC loss.

## The eIF2Bα translational program is characterized by GA content of coding sequences

As we hypothesized that increased ribosome occupancy in the 5′ UTR of all transcripts was due to reduced TC levels (Fig. 5A), similar to what happens during the ISR, we examined ribosome coverage in transcripts regulated at the RPFs level by eIF2Bα or eIF2Bδ knockdown (Appendix Fig. S8A,B). Interestingly, transcripts with reduced RPFs in their CDS upon eIF2Bα knockdown showed an increase in their 5′ UTR footprints (Appendix Fig. S8A, upper panel), whilst this was not visible for eIF2Bδ knockdown (Appendix Fig. S8B, upper panel). To further characterize the translational programs regulated by eIF2Bα or eIF2Bδ, we employed gradient-boosting regression analysis, predicting the contribution of transcripts' molecular features to changes in TE.

Surprisingly, the presence of uORFs in the 5′ UTR, 5′ UTR length, or 5′ UTR complexity (probed as %GC) did not have a contribution to log2TE, scoring cumulatively less than 2% in relative influence (Appendix Fig. S8C). Also, the usage of non-annotated translation start-sites determined by QTI-seq (Gao et al, 2015) (upstream: uTIS, downstream: dTIS) did not have an influence on log2TE (Appendix Fig. S8C). Thus, eIF2Bα and eIF2Bδ knockdown induce specific translational signatures not associated with features predicted to be regulated by reduced TC availability. Unexpectedly, the log2FC TE was mostly explained by the GC content of the CDS (Appendix Fig. S8C). Correlation between log2FC and GC content shows that genes with higher TE also display higher GC content in their CDS in both eIF2Bα- or eIF2Bδ-depleted conditions (Appendix Fig. S8D). Relative Synonymous Codon Usage (RSCU) analysis shows that this trend is driven by the third nucleotidic position (wobble) of codons. Indeed, upon eIF2Bα or eIF2Bδ knockdown, genes with RPFs increased, compared to genes with RPFs decreased, have a preference for codons displaying a G or a C in the wobble position (GC3 codons) (Appendix Fig. S8E). Codon usage has previously been associated to opposite gene sets: GO terms related to proliferation (such as M-phase) were found to be enriched in AT-ending (AT3) codons, whilst differentiation-associated terms (such as Pattern Specification Process) were characterized by GC3 codons (Gillen et al, 2021; Gingold et al, 2014). Thus, the preference for GC3 codon translation corroborates the idea that eIF2B decameric complex loss and reduced p-eIF2α sensing ability are detrimental to molecular programs driving cell proliferation. To identify the molecular characteristics that could distinguish between transcripts regulated only at the RPF level in eIF2Bα compared to eIF2Bδ knockdown only (Fig. 5C), we took advantage of gradient-boosting classification capabilities (Appendix Fig. S8F,G). The predictive model indicates that transcripts can be classified as being regulated by eIF2Bα rather than eIF2Bδ on the basis of the GA content of their coding sequences (Appendix Fig. S8F,G) (Mudge et al, 2022). In summary, the interaction between p-eIF2α and the eIF2B complex is necessary to balance the translation of growth-promoting mRNAs and a translation inhibitory signal for growth-limiting mRNAs, the latter of which are characterized by a high GC content in the wobble position.

## APC-deficient, but not WT, intestinal organoids are dependent on eIF2Bα

In yeast, eIF2Bα is the only eIF2B subunit whose complete loss is not lethal (Hannig and Hinnebusch, 1988). In addition, eIF2Bα is dispensable for mammalian cell viability (Elsby et al, 2011). Analysis

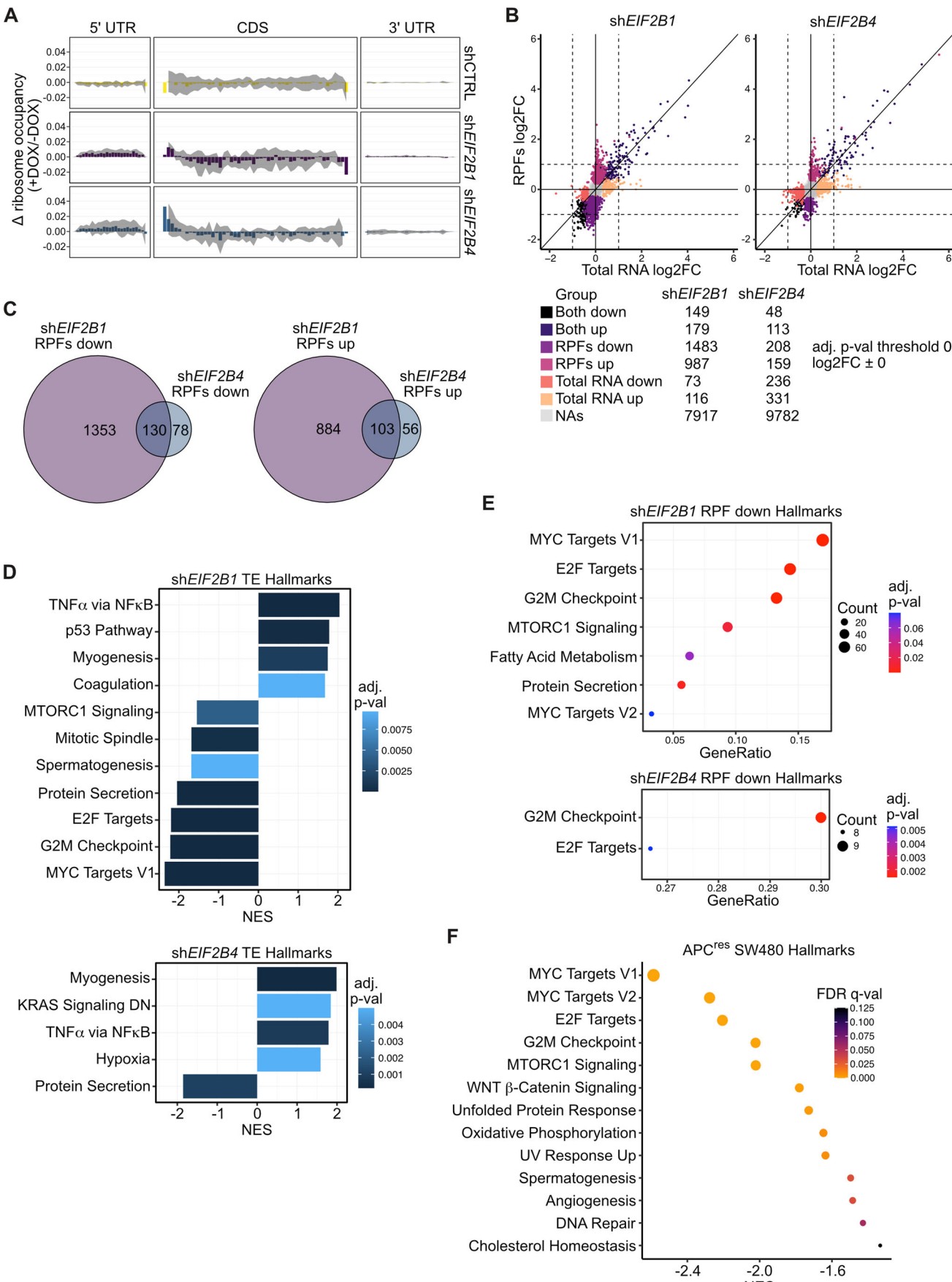

◀ **Figure 5.  eIF2Bα is essential for the translation of growth-promoting mRNAs in CRC.**

(A) Metagene plots representing the variation in ribosomal density upon doxycycline-mediated induction of shRNAs, indicated right each plot. All transcripts detected in the Ribo-seq experiment with >10 reads/sample are represented. Each portion of the transcript is binned (5′ and 3′ untranslated regions—5′ UTR/3′ UTR = 25 bins; coding sequence—CDS = 50 bins). (B) Differential expression analysis of total cytoplasmic RNA (x axes) and RPFs (y axes) upon shRNA induction (sh*EIF2B1* left, sh*EIF2B4* right), corrected for shCTRL effects. The number of genes passing adjusted *P* value threshold of 0.1 are listed in the table and color coded as the plots. (C) Venn diagrams showing the overlap of transcripts in the RPFs down (left) and up (right) categories upon sh*EIF2B1* and sh*EIF2B4* induction (shCTRL corrected). Groups are defined in (B). (D) Gene Set Enrichment Analysis (GSEA) for Hallmark pathways of transcripts ranked by log2 Translation Efficiency (TE) upon sh*EIF2B1* (top) or sh*EIF2B4* (bottom) induction (shCTRL corrected). Pathways passing enrichment adjusted *P* value threshold of 0.01 are depicted. Normalized enrichment scores (NES) are represented on the x axis. Gene Set Enrichment statistics are calculated with the fGSEA R package. (E) Overrepresentation analysis for Hallmark pathways of transcripts belonging to the RPFs down category only in sh*EIF2B1* (top) or sh*EIF2B4* (bottom). Pathways passing enrichment adjusted *P* value threshold of 0.1 are depicted. GeneRatios are represented on the x axis and dot sizes depict the number of transcripts belonging to the categories listed. Overrepresentation statistics are calculated with a hypergeometric test followed by Benjamini–Hochberg correction. (F) GSEA for Hallmark pathways of transcripts ranked by log2FC total RNAs in SW480 cells upon full-length APC re-expression (APC[res]). Pathways passing enrichment FDR *q* value threshold of 0.25 are depicted. NES are represented on the x axis.

of cancer dependency data using DepMap showed that cancer cells commonly depend on all subunits of the eIF2 complex (*EIF2S1-3*) as well as eIF2B subunits βγδε (Appendix Fig. S9A) (Behan et al, 2019; McDonald et al, 2017; Tsherniak et al, 2017). In contrast, eIF2Bα displays the lowest dependency among all analyzed factors and, thus, is not necessary for unperturbed cellular homeostasis in general (Appendix Fig. S9A). Together with the fact that depletion of eIF2Bα does not destabilize the other eIF2B subunits (Fig. 2), targeting of eIF2Bα suggests a therapeutic window.

To validate eIF2Bα as potential candidate for the treatment of CRC in a physiological context, we used the murine WT, A, AK and LAKTP organoids transduced with one or two independent doxycycline-inducible shRNAs against mouse *Eif2b1* (sh*Eif2b1-1* and sh*Eif2b1-2*) or, as control, a non-targeting shRNA (shCTRL). Doxycycline-induced eIF2Bα depletion reduced the viability of A, AK and LAKTP tumor organoids, but did not affect WT organoids (Figs. 6A–D and EV5A–D). Similar to what we observed in APC-deficient SW480 cells, p-eIF2α levels were decreased upon eIF2Bα knockdown in A, AK (except for sh*Eif2b1-2*) and LAKTP organoids (Figs. 6C,D and EV5C,D). This was also apparent in WT organoids, but to a much lesser degree. For further validation, also PDOs T4, FAP and HD-3 were transduced with two independent doxycycline-inducible shRNAs targeting *EIF2B1* (sh*EIF2B1-1* and sh*EIF2B1-2*) or a non-targeting shCTRL. The viability of all three PDOs was reduced upon doxycycline-induced eIF2Bα depletion (Figs. 6E–H and EV5E–H). More specifically, analyzing the growth of PDO HD-3 via live imaging over 7 days revealed a significant reduction in organoid size upon eIF2Bα knockdown by sh*EIF2B1-1* (Appendix Fig. S9B). In addition, knockdown of eIF2Bα led to reduction of eIF2α phosphorylation in all PDOs (Figs. 6G,H and EV5G,H). To validate the tumor-specific effects of eIF2Bα knockdown, we depleted eIF2Bα via sh*EIF2B1-1* in one WT human organoid derived from normal colon tissue (WT Ko165). Whereas knockdown efficiency of eIF2Bα was comparable to tumor PDOs, WT Ko165 did not show a reduction in viability (Fig. 6E–H). Furthermore, we investigated the effects of eIF2Bδ depletion in murine WT and LAKTP organoids as well as in human PDO T4. Knockdown of eIF2Bδ did reduce the viability of tumor-derived organoids, but also of murine WT organoids (Appendix Fig. S9C–H), suggesting that modulation of eIF2Bδ has rather unspecific, toxic cellular effects. To conclude, by decreasing eIF2Bα levels in clinically relevant models we provide a proof-of-concept demonstrating it as a target for therapeutic intervention.

## Discussion

Here we demonstrate that interfering with sensing of p-eIF2α by the eIF2B complex induces a specific vulnerability in CRC by repressing translation of the transcriptional signature driven by APC mutation (Fig. 7). We show that CRC harbors increased levels of phosphorylated eIF2α compared to normal tissue, accompanied by higher bulk translation, which are needed to ensure balanced translation of this oncogenic mRNA signature. Elevated p-eIF2α has also been observed in different gastrointestinal malignancies as well as other tumor entities including pancreatic cancer, breast cancer, prostate cancer and MYC-driven lymphoma where it is partially involved in tumor progression (Bai et al, 2021; Guo et al, 2017; Hart et al, 2012; Koromilas, 2015; Lobo et al, 2000; Nguyen et al, 2018; Shin et al, 2022). In contrast, phosphorylation of eIF2α has a potential growth suppressive effect in other cellular contexts, where expression of its phospho-mimicking form, eIF2α S51D, results in apoptosis and cell cycle arrest (Darini et al, 2019; Perkins and Barber, 2004; Scheuner et al, 2006; Teng et al, 2014). Our data argue that elevated phosphorylation of eIF2α represents a key vulnerability in CRC and that altering cellular capabilities to sense it is a viable therapeutic option. This can also be transferred to other cancer types, as shown in this study for PDAC cells, which is potentially dependent on the level of eIF2α phosphorylation in the respective tumor type.

Targeting eIF2α/p-eIF2α directly is not a promising anti-cancer strategy, as eIF2α is a common essential gene for cellular homeostasis (Appendix Fig. S9A). Previously, we established that two of the four eIF2α kinases, GCN2 and PKR, are activated in CRC in a MYC-dependent manner and, thus, are involved in increased eIF2α phosphorylation (Schmidt et al, 2019). Nevertheless, all four respective kinases, GCN2/PKR/PERK/HRI, show redundancies and unanticipated feedback loops, as well as their inhibition has adverse side effects (Lehman et al, 2015; Schmidt et al, 2019; Szaruga et al, 2023; Tameire et al, 2019). Another method to target this pathway is via rendering eIF2B insensitive to p-eIF2α. This can be achieved via the small molecule ISRIB, which we tested and showed a peculiar absence of downstream effects in CRC cells. Therefore, we reasoned that a different method to interfere with the ability of the eIF2B complex to sense p-eIF2α was required. We hypothesized that targeting either eIF2Bα or eIF2Bδ, both of which are essential for sensing phosphorylated eIF2α (Kashiwagi et al, 2019; Kenner et al, 2019), could be a viable strategy. Surprisingly, depletion of either one of these eIF2B

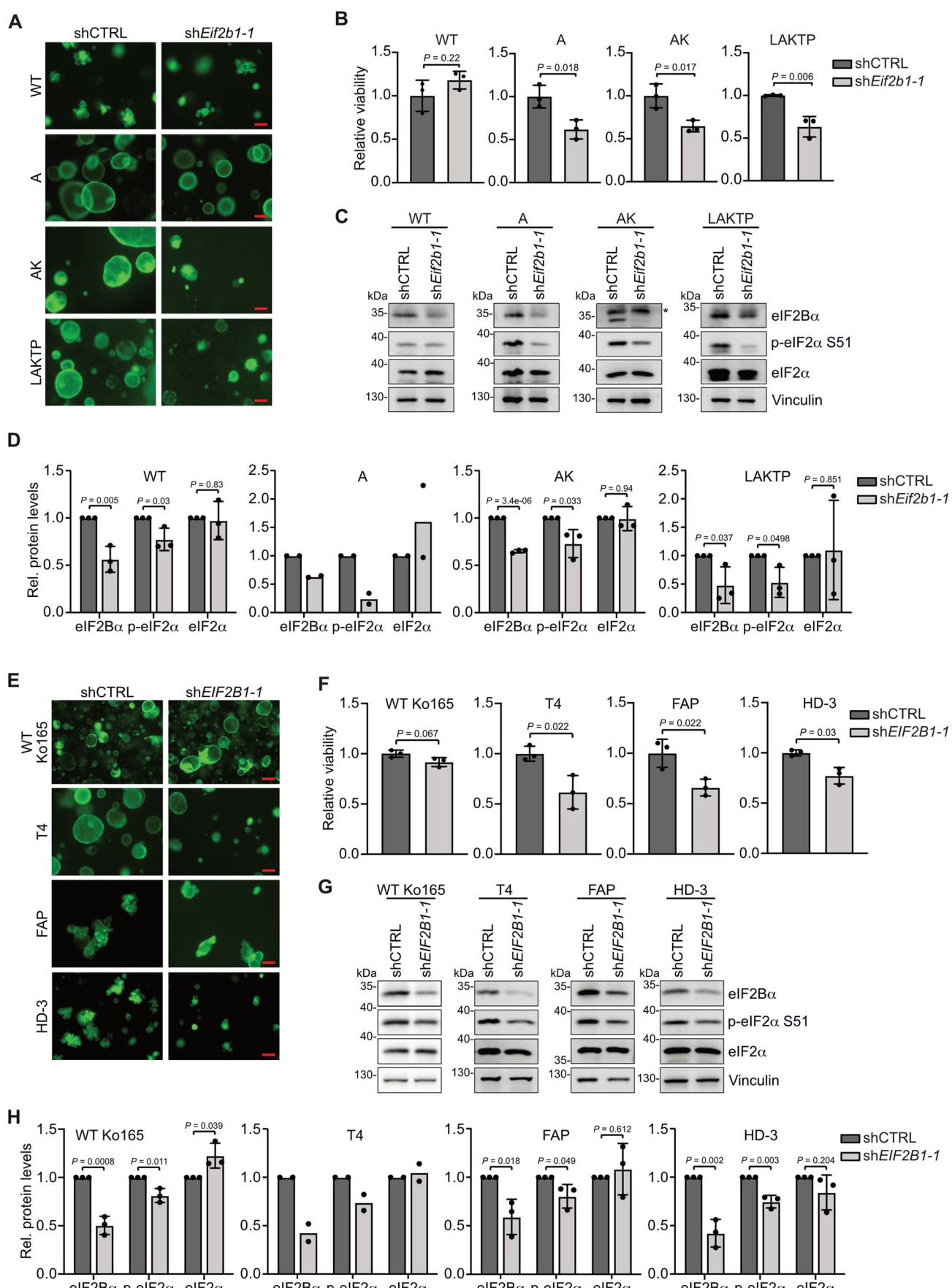

**Figure 6.  APC-deficient, but not WT, intestinal organoids are dependent on eIF2Bα.**

(A) Pictures of murine WT, A, AK and LAKTP intestinal organoids transduced with doxycycline-inducible shCTRL or sh*Eif2b1-1* (7 days of doxycycline treatment), representative of three biological replicates with similar results. Green signal (GFP) indicates shRNA induction. Scale bar = 200 μm. (B) Relative viability of WT, A, AK, and LAKTP organoids transduced and treated as described in (A). Data show mean ± s.d. ($n = 3$ biological replicates); Student's *t* test. (C) Western blot of indicated proteins in WT, A, AK, and LAKTP organoids transduced as described in (A), representative of two or three biological replicates with similar results (96 h of doxycycline treatment); *unspecific bands. (D) Quantification of eIF2Bα, p-eIF2α S51 and total eIF2α levels, normalized to vinculin, of western blots described in (C). Data show mean ± s.d. of two to three biological replicates; Student's *t* test. (E) Pictures of human WT Ko165 organoids and T4, FAP, HD-3 PDOs transduced with doxycycline-inducible shCTRL or sh*EIF2B1-1* (7 days of doxycycline treatment), representative of three biological replicates with similar results. Green signal (GFP) indicates shRNA induction. Scale bar = 200 μm. (F) Relative viability of WT Ko165, T4, FAP, HD-3 organoids transduced and treated as described in (E). Data show mean ± s.d. ($n = 3$ biological replicates); Student's *t* test. (G) Western blot of indicated proteins in WT Ko165, T4, FAP, HD-3 organoids transduced as described in (E), representative of two or three biological replicates with similar results (96 h of doxycycline treatment). (H) Quantification of eIF2Bα, p-eIF2α S51 and total eIF2α levels, normalized to vinculin, of western blots described in (G). Data show mean ± s.d. ($n = 2$ or 3 biological replicates); Student's *t* test. Source data are available online for this figure.

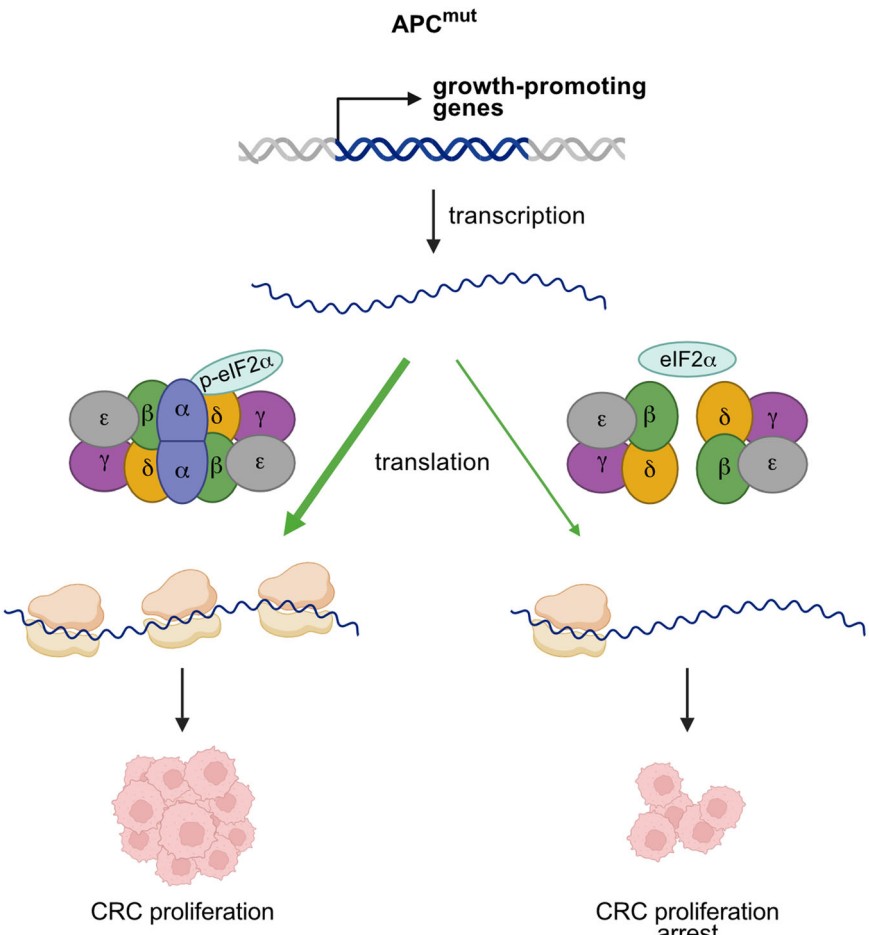

**Figure 7.  Sensing of p-eIF2α levels by the eIF2B complex is vital for CRC.**

Model explaining our findings. Loss of APC induces deregulated transcription of growth-promoting genes whose translation is dependent on an intact eIF2B complex capable of sensing p-eIF2α. Disruption of this interaction by modulation of eIF2Bα specifically reduces translation of this oncogenic mRNA signature leading to CRC proliferation arrest. Created in BioRender. Schmidt, 2025. https://BioRender.com/f09b706.

subunits resulted in strikingly different cellular and biochemical outcomes. Although eIF2Bδ has oncogenic relevance in other cancer types, e.g., breast cancer (Gupta et al, 2023), it seems an unsuitable target in CRC as modulation of its levels results in overall disrupted translational homeostasis leading to uncontrolled cell death. On the contrary, loss of eIF2Bα induces cell cycle arrest,

has no effect on bulk translation rates, and does not interfere with expression of the other eIF2B subunits. When we characterized the transcripts which are translationally deregulated upon eIF2Bα depletion, we found that they belong to the same functional categories as those transcriptionally upregulated upon APC loss. This argues that there is a second layer of translational control

required to promote oncogenic expression programs following loss of APC. The molecular consequences of removing eIF2Bα and eIF2Bδ were strikingly different, despite the fact that they both are required for sensing p-eIF2α. Indeed, knockdown of eIF2Bδ disrupts completely the decamer and thus, upon reduction of TC availability, cells are unable to maintain basal translation levels. We show that the effect on bulk translation is entirely dependent on maintenance of the decameric structure, as eIF2Bδ$^{mut}$ cells retain basal protein synthesis rates upon endogenous eIF2Bδ depletion. In contrast, upon eIF2Bα depletion, there is still remaining eIF2B activity and the reduction of TC that induces ISR genes is not associated with strong global protein synthesis reduction. Therefore, depletion of either eIF2Bα or eIF2Bδ regulates different expression programs and this feature depends on the degree to which the eIF2B decamer is disrupted and, consequently, on the levels of available TC remaining in the cells.

Importantly, WT intestinal epithelial cells were not dependent on eIF2Bα levels, establishing it as an attractive therapeutic target in CRC. This observation correlates with data from yeast showing that GCN3 (the yeast eIF2Bα homolog) per se is dispensable under non-starved conditions. Only under stressed conditions (e.g., tumor growth) its deletion or mutation makes the eIF2B complex insensitive to the inhibitory effects of eIF2α phosphorylation (Dever et al, 1993; Hannig et al, 1990; Pavitt et al, 1997). Furthermore, mutations in eIF2B subunits are causal for VWM disease, a leukoencephalopathy with severe neurological disorders, and different mutations within different subunits cause variable severities of the disease (Leegwater et al, 2001; van der Knaap et al, 2002). Among all eIF2B subunits, eIF2Bα shows the lowest mutation rate and is least associated with severe effects (Ohlenbusch et al, 2005; Slynko et al, 2021). Therefore, modulation of eIF2Bα, among all other subunits, promises to have the least side effects. As the ability of eIF2Bα to sense p-eIF2α lacks enzymatic activity, its function cannot be directly inhibited via small molecules. There are two possible strategies to interfere with the function of eIF2Bα. First, reducing the levels of eIF2Bα could be achieved via a proteolysis-targeting chimera (PROTAC) approach that has emerged as alternative strategy to target proteins of interest with potential clinical use during the last years (Adhikari et al, 2020; Bekes et al, 2022; Otto et al, 2019). Second, design of a peptide inhibitor that disrupts the eIF2Bα homodimer and, thus, interferes with eIF2B decamer formation and p-eIF2α sensing, could be of potential value. A similar strategy has been successfully implemented against ERK or BRAF dimers (Gunderwala et al, 2019; Tomasovic et al, 2020; Yao et al, 2019).

# Methods

### Reagents and tools table

| Reagent/resource | Reference or source | Identifier or catalog number |
|---|---|---|
| **Experimental models** | | |
| HEK293T cells (*H. sapiens*) | ATCC | CRL-3216 |
| SW480 cells (*H. sapiens*) | ATCC | CCL-228 |
| DLD1 cells (*H. sapiens*) | ATCC | CCL-221 |

| Reagent/resource | Reference or source | Identifier or catalog number |
|---|---|---|
| PaTu8988 cells (*H. sapiens*) | Mathias Rosenfeldt, EKO, Oberhausen | N/A |
| HEK293-Noggin (*H. sapiens*) | Elmar Wolf, CAU Kiel | N/A |
| L17-R-spondin (*H. sapiens*) | Owen J. Sansom, CRUK Scotland Institute, Glasgow | |
| WT organoids (*M. musculus*) | This study | N/A |
| A organoids (*M. musculus*) | Owen J. Sansom, CRUK Scotland Institute, Glasgow (el Marjou et al, 2004; Sansom et al, 2007; Shibata et al, 1997) | N/A |
| AK organoids (*M. musculus*) | Owen J. Sansom, CRUK Scotland Institute, Glasgow (el Marjou et al, 2004; Sansom et al, 2007; Shibata et al, 1997) | N/A |
| LAKTP organoids (*M. musculus*) | Eduard Batlle, IRB, Barcelona (Tauriello et al, 2018) | N/A |
| T4 organoids (*H. sapiens*) | Schmidt et al, 2019 | N/A |
| FAP organoids (*H. sapiens*) | This study | N/A |
| HD-3 organoids (*H. sapiens*) | Rene-Filip Jackstadt, HiStem Heidelberg | N/A |
| Ko165 organoids (*H. sapiens*) | This study, Nicolas Schlegel, University Hospital Würzburg | N/A |
| **Recombinant DNA** | | |
| psPAX2 | Didier Trono, Addgene | #12260 |
| pMD2.G | Didier Trono, Addgene | #12259 |
| pLeGO Hygro sh*EIF2B1-3* | This study | N/A |
| pLeGO Hygro sh*EIF2B2-1* | This study | N/A |
| pLeGO Hygro sh*EIF2B3-5* | This study | N/A |
| pLeGO Hygro sh*EIF2B4-2* | This study | N/A |
| pLeGO Hygro sh*EIF2B5-3* | Schmidt et al, 2019 | N/A |
| pLeGO Hygro shLuciferase (shCTRL) | Schmidt et al, 2019 | N/A |
| LT3-GEPIR-sh*EIF2B1-1* | This study | N/A |
| LT3-GEPIR-sh*EIF2B1-2* | This study | N/A |
| LT3-GEPIR-sh*Eif2b1-1* | This study | N/A |
| LT3-GEPIR-sh*Eif2b1-2* | This study | N/A |
| LT3-GEPIR-sh*Eif2b4* | This study | N/A |
| LT3-GEPIR-sh*EIF2B4* | This study | N/A |
| LT3-GEPIR-shCTRL | This study | N/A |
| pRRL-SFFV-puro-eIF2Bα$^{wt}$ | This study | N/A |
| pRRL-SFFV-puro-eIF2Bα$^{mut}$ | This study | N/A |
| pRRL-SFFV-puro-eIF2Bδ$^{wt}$ | This study | N/A |
| pRRL-SFFV-puro-eIF2Bδ$^{mut}$ | This study | N/A |
| pRRL-SFFV-puro-6xHis-HA-eIF2Bα$^{wt}$ | This study | N/A |
| pRRL-SFFV-puro-6xHis-HA-eIF2Bα$^{mut}$ | This study | N/A |
| pRRL-SFFV-puro-HA-eIF2Bδ$^{wt}$ | This study | N/A |
| pRRL-SFFV-puro-HA-eIF2Bδ$^{mut}$ | This study | N/A |
| pRRL-SFFV-puro-HA-eIF2α$^{wt}$ | This study | N/A |
| pRRL-SFFV-puro-HA-eIF2α$^{S51A}$ | This study | N/A |
| pRRL-SFFV-puro-HA-eIF2α$^{S51D}$ | This study | N/A |
| pLV-Bsd-EIF2Bε$^{wt}$-HA | Schmidt et al, 2019 | N/A |
| pLV-Bsd-EIF2Bε$^{R113H}$-HA | This study | N/A |

| Reagent/resource | Reference or source | Identifier or catalog number |
|---|---|---|
| **Antibodies** | | |
| ATF4 | Santa Cruz Biotechnology | sc-200 |
| ATF4 | Cell Signaling Technology | 11815 |
| Actin | Merck Millipore | MAB1501 |
| β-actin | Sigma | A1978 |
| CHOP | Cell Signaling Technology | 2895 |
| eIF2α | Bethyl | A300-721A-M |
| eIF2Bα | Proteintech | 18010-1-AP |
| eIF2Bβ | Proteintech | 11034-1-AP |
| eIF2Bγ | Santa Cruz Biotechnology | sc-137248 |
| eIF2Bδ | Proteintech | 11332-1-AP |
| eIF2Bδ | Santa Cruz Biotechnology | sc-271332 |
| eIF2Bε | Santa Cruz Biotechnology | sc-28854 |
| eIF2Bε | abcam | |
| p-eIF2α S51 | Cell Signaling Technology | 9721 |
| p-eIF2α S51 | Cell Signaling Technology | 3398 |
| p-eIF2α S51 | Abcam | ab32157 |
| Puromycin | Sigma Aldrich | MABE343 |
| Vinculin | Sigma | V9131 |
| GAPDH | Santa Cruz Biotechnology | sc-32233 |
| HA | Abcam | ab9110 |
| GADD34 | Proteintech | 10449-1-AP |
| HNRNPD | Proteintech | 12770-1-AP |
| HNRNPA3 | Proteintech | 25142-1-AP |
| CBX4 | Proteintech | 18544-1-AP |
| IRDye® 680RD Goat-Anti-Rabbit (IgG) | Li-Cor | 926-68071 |
| IRDye® 800CW Goat-Anti-Mouse (IgG) | Li-Cor | 926-32210 |
| ECL-Anti-rabbit IgG HRP linked | GE Healthcare | 1079-4347 |
| ECL-Anti-mouse IgG HRP linked | GE Healthcare | 1019-6124 |
| **Oligonucleotides and other sequence-based reagents** | | |
| qPCR primers | This study | Table EV1 |
| cgtcccagataagtttaagta | #DHPAC-M1-P, Cellecta (Yang et al, 2011) | sh*EIF2B1-1* |
| gcagggcttaacttgttgatt | #DHPAC-M1-P, Cellecta (Yang et al, 2011) | sh*EIF2B2-1* |
| gcccacctctacttgtttgaaa | Fellmann et al, 2013 | sh*EIF2B3-5* |
| cggttgttgaatctagtctat | #DHPAC-M1-P, Cellecta (Yang et al, 2011) | sh*EIF2B4-2* |
| tgcacgtaacagctcaaggaat | Schmidt et al, 2019 | sh*EIF2B5-3* |
| cttcgaaatgttcgtttggtt | Schmidt et al, 2019 | shLuciferase (shCTRL) |
| ttatacttaaacttatctgggga | Fellmann et al, 2013 | sh*EIF2B1-1* (Dox-inducible) |
| GTCCCAGATAAGTTTAAGTATA | Fellmann et al, 2013 | sh*EIF2B1-2* (Dox-inducible) |
| tagactagattcaacaaccgta | Fellmann et al, 2013 | sh*EIF2B4* (Dox-inducible) |
| cttactctcgcccaagcgagag | This study | shCTRL (Dox-inducible) |

| Reagent/resource | Reference or source | Identifier or catalog number |
|---|---|---|
| tttctatagacaacattctgtg | Fellmann et al, 2013 | sh*Eif2b1-1* mouse (Dox-inducible) |
| TCCCAGATAAGTTTAAGTACAA | Fellmann et al, 2013 | sh*Eif2b1-2* mouse (Dox-inducible) |
| CTCCAGGGATCTTGTAAATAAA | Fellmann et al, 2013 | sh*Eif2b4* mouse (Dox-inducible) |
| Ugguuuacaugucgacuaa Ugguuuacauguuguguga Ugguuuacauguuuucuga ugguuuacauguuuuuccua | Horizon Discovery | siNTC, #D-001810-10 |
| Guuaacaaguucucuuauc Gaaaggugcgcuucuccga Gcagggaagucaauuugca uggcauguauggugagcga | Horizon Discovery | si*PPP1R15A* (siGADD34), #L-004442-02 |
| **Chemicals, enzymes, and other reagents** | | |
| Phosphatase inhibitor cocktail 2 | Sigma Aldrich | P5726 |
| Phosphatase inhibitor cocktail 3 | Sigma Aldrich | P0044 |
| Ribolock RNase Inhibitor | Thermo Fisher Scientific | P0044 |
| dNTPs | Roth | K039.1 |
| M-MLV | Promega | M1705 |
| Protease Inhibitor | Sigma Aldrich | P8340 |
| peq GOLD TriFast | peqlab/VWR International | VWRC30 |
| GlycoBlue | Ambion | AM9516 |
| Dynabeads A/G | Thermo Fisher Scientific | 10009D, 10009D |
| Propidium iodide (PI) | Sigma Aldrich | 81845 |
| Annexin V-Pacific Blue | Life Sciences | |
| L-Cystein, L-Methionin S35-label | Hartmann Analytics | SCIS103/74 |
| Page Ruler Prestained Protein Ladder | Thermo Fisher Scientific | 26617 |
| CellTiter-Blue Cell Viability Assay | Promega | G8080 |
| CellTiter-Glo 3D Cell Viability Assay | Promega | G9683 |
| Immobilon Western HRP Substrate | Millipore | WBKL S0 500 |
| PowerUp SYBR Green Master Mix | Thermo Fisher Scientific | A25742 |
| Restriction enzymes | New England Biolabs | |
| GELTREX LDEV FREE RGF BME | Thermo Fisher Scientific | A1413202 |
| Cultrex RGF Basement Membrane Extract,Type 2, Select | Biotechne | 3536-005 |
| Advanced DMEM/F12 | Thermo Fisher Scientific | 12634028 |
| N2 | Invitrogen | 17502-048 |
| B27 | Invitrogen | 12587-010 |
| Rec. murine EGF | Peprotech | 315-09 |
| Nicotinamide | Sigma | N0636 |
| N-Acetyl-L-Cysteine | Sigma | A9165 |
| Gastrin | Sigma | G9145 |
| A 83-01 | Tocris Bioscience | 2939 |
| IGF-I | Biolegend | 590906 |
| Wnt surrogate Fc fusion recombinant protein | Gibco | PHG0401 |
| FGF basic | Peprotech | 100-18B |
| Y-27632 | SEL-S1049 Selleckchem / Biozol | SEL-S1049 |

| Reagent/resource | Reference or source | Identifier or catalog number |
|---|---|---|
| IntestiCult™ Organoid Growth Medium (Human) | Stemcell Technologies | 6010 |
| DMEM | Capricorn Scientific | DMEM-HPA |
| RPMI 1640 | Thermo Fisher Scientific | 21875091 |
| FBS | Capricorn Scientific | FBS-11A |
| doxycycline | Sigma | D9891 |
| cycloheximide | Sigma | C7698 |
| tunicamycin | Sigma | T7765 |
| ISRIB | Sigma | SML0843 |
| RNaseI | Ambion (Invitrogen) | AM2295 (lot# 00791236) |
| SUPERaseIn™ | Ambion (Invitrogen)csx | AM2696 |
| NEXTflex small RNA kit (v3 - discontinued) | Perkin Elmer (now Revvity) | NOVA-5132-05 |
| CORALL total RNA seq kit (v1 - discontinued) | Lexogen | 095.96 |
| RiboCop (H/M/R) (v2) | Lexogen | 144.96 |
| NextSeq 500/550 High Output Kit v2.5 (75 Cycles) | Illumina | 20024906 |
| **Software** | | |
| ImageJ v1.53t | https://imagej.net/ij/ | |
| GraphPad Prism v10.0.3 for Mac | https://www.graphpad.com/scientific-software/prism/ | |
| Harmony High Content Imaging and Analysis Software v4.8 | http://www.perkinelmer.de/product/harmony-4-8-office-hh17000001 | |
| StepOne software v2.3 | https://www.thermofisher.com/de/de/home/technical-resources/software-downloads/StepOne-and-StepOnePlus-Real-Time-PCR-System.html | |
| Multi Gauge software v3.0 | https://cloudfront.ualberta.ca/-/media/science/departments/biological-sciences/mbsu/fla-5000/mulitgauge20.pdf | |
| BD FACSDIVA Software v9.0.1 | http://www.bdbiosciences.com/us/instruments/research/software/flow-cytometry-acquisition/bd-facsdiva-software/m/111112/overview | |
| TecanSpark Control v3.1 | Tecan Trading AG | |
| IncuCyte 2021C | Sartorius | |
| Fusion FX Evolution Capt Edge v18.14.0.0 | Vilber | |
| Microsoft Office version 16.16.27 | University Wuerzburg | |
| Affinity Designer version 2.2.1 | https://affinity.serif.com/de/ | |
| Molecular Operating Environment 2019.01 | ULC CCG (2019) Chemical Computing Group ULC, 1010 Sherbrooke St. West, Suite #910, Montreal, QC, Canada, H3A 2R7, 2019 | |
| ZEN software | Zeiss | |
| R | v 4.3.2 (2023-10-31) | |
| RStudio | v 2023.12.0, Build 369 | |
| **Other** | | |
| Fusion FX | Vilber | |
| Image Reader LAS 4000 | Fujifilm | |

| Reagent/resource | Reference or source | Identifier or catalog number |
|---|---|---|
| TecanSpark | Tecan | |
| Incucyte SX5 | Sartorius | |
| StepOne Plus Real-Time PCR System | Applied Biosystems | |
| NextSeq 500 sequencer | Illumina | |
| Operetta™ High-Content Screening System | Perkin Elmer (now Revvity) | |
| BD FACS Canto™ II | BD Biosciences | |

## Cell culture

HEK293T, DLD1, PaTu8988T, HEK293-Noggin and L17-R-spondin cells were cultured in DMEM (#DMEM-HPA, Capricorn Scientific), SW480 cells were cultured in RPMI 1640 (#21875091, Thermo Fisher Scientific), supplemented with 10% heat-inactivated FBS (#FBS-11A, Capricorn Scientific) and 1% penicillin/streptomycin (#P4333, Sigma). HEK293-Noggin and L17-R-spondin cells were selected with 0.5 mg/ml and 1 mg/ml, respectively, G418 (#4727878001, Sigma). HEK293-Noggin cells were provided by Elmar Wolf, L17-R-spondin cells were provided by Owen J. Sansom (CRUK Scotland Institute, Glasgow). HEK293T and SW480 cell lines were authenticated via STR analysis. All cells were regularly tested for mycoplasma contamination, last testing was July 2024.

Following reagents were added where specified: doxycycline (#D9891, Sigma), cycloheximide (#C7698, Sigma), puromycin (#ant-pr, Invivogen), tunicamycin (#T7765, Sigma), ISRIB (#SML0843, Sigma).

## Murine organoid and patient-derived organoid (PDO) culture

Mice were housed in pathogen-free conditions on a 12-h dark/light cycle with unlimited access to food and water and were used for small intestine isolation. Animal procedures were conducted according to the German Animal Protection Law §4 Abs. 1 TierSchG. Isolation of WT small intestinal crypts from C57BL/6 mice was performed as previously described (Schmidt et al, 2019). *VillinCre*^ER^*Apc*^-/-^ (RZF42.3c) and *VillinCre*^ER^*Apc*^−/−^*Kras*^G12D/+^ (RZF42.3f) murine small intestinal organoid lines were provided by Owen J. Sansom (CRUK Scotland Institute, Glasgow). *Lgr5*^eGFP-creERT2^*Apc*^−/−^*Kras*^G12D/+^*Tgfbr2*^−/−^*Trp53*^−/−^ (LAKTP) organoid line (MTO204) was provided by Eduard Batlle (IRB, Barcelona) (Tauriello et al, 2018). Isolation and culture of PDOs (T4, FAP, Ko165) was approved by the ethic committee of the University of Würzburg (#142/16-ge; amendment 20). Human WT organoid Ko165 was provided by Nicolas Schlegel. All patients gave written consent prior to surgery. Organoid isolation and culture were performed as described previously (Schmidt et al, 2019). PDO HD-3 was provided by Rene-Filip Jackstadt, isolation and culture was approved by the ethic committee of the University of Heidelberg (S-136/2021). All organoid lines were cultured in Cultrex BME Type 2 (#3536, Biotechne) or Geltrex™ (#A1413202, Thermo Fisher Scientific) as previously described (Schmidt et al, 2019; Tauriello et al, 2018). Organoid culture medium contained 10% R-

spondin-conditioned medium and 1% Noggin-conditioned medium produced in L17-R-spondin and HEK293-Noggin cells, respectively. PDO Ko165 was cultured in IntestiCult™ Organoid Growth Medium (Stemcell Technologies) including 10 µM Y-27632 ROCK inhibitor (#SEL-S1049, Selleckchem).

## Lentiviral transduction and transfection

All shRNA experiments were carried out by stable lentiviral transduction. Lentiviruses were generated by transfection of HEK293T cells with LeGO-iG2 or LT3-GEPIR together with packaging plasmids psPAX2 and pMD2.G. shRNA sequences were from the DECIPHER shRNA Library Module I (#DHPAC-M1-P, Cellecta) (Yang et al, 2011) or from (Fellmann et al, 2013). Plasmids psPAX2 and pMD2.G were a gift from Didier Trono (Addgene plasmid # 12260; http://n2t.net/addgene:12260; RRID: Addgene_12260 and Addgene plasmid # 12259; http://n2t.net/addgene:12259; RRID: Addgene_12259). LeGO-iG2 was a gift from Boris Fehse (Addgene plasmid # 27341; http://n2t.net/addgene:27341; RRID: Addgene_27341) (Weber et al, 2008). LT3-GEPIR was a gift from Johannes Zuber (Addgene plasmid # 111177; http://n2t.net/addgene:111177; RRID: Addgene_111177) (Fellmann et al, 2013). Infections were carried out using 8 µg/ml polybrene (#107689, Sigma). Two days after infection, cells were selected with 6 µg/ml blasticidin (#ant-bl, Invivogen), 1–2 µg/ml puromycin (#ant-pr, Invivogen) or 600 µg/ml hygromycin (#ant-hg, Invivogen), and pools of selected cells were used for downstream analyses.

## Crystal violet staining and fluorescence-activated cell sorting (FACS)

For crystal violet staining, 100,000 cells per condition were seeded in six-well plates. Cells were stained with 0.1% crystal violet in 20% ethanol 7 days after seeding. Propidium iodide (PI) and annexin V/PI FACS analysis was performed as previously described (Schmidt et al, 2019).

## Operetta™ growth curve

Cells were trypsinized, counted and 2000 cells were seeded in 96-well plates (CellCarrier Ultra, Perkin Elmer) in three technical replicates. Images were taken with the Operetta™ High-Content Screening System (Perkin Elmer, now Revvity) with the 20× long WD objective. Digital phase contrast (DPC) and brightfield images were acquired (38 image fields per well) and analysis was performed with the Harmony® Software. DPC images were used for cell counting.

## Live-cell imaging via Incucyte®

Cells were trypsinized, counted and 2000 cells were seeded on a 96-well plate in triplicates in 100 µl of appropriate medium. Live imaging was performed with the Incucyte® SX5 Live-Cell Analysis System (Sartorius) with images acquired every 12 h over 7 days. For organoid imaging, PDOs were disrupted mechanically, and organoids were seeded in one drop of 70% Cultrex BME/organoid medium into the middle of a well of a 24-well plate (#CLS3526, Corning), and human organoid medium was added. Images were

acquired every 8 h over 7 days. Analysis was performed using the Incucyte® organoid software module.

## Viability assay of murine and patient-derived organoids

For analysis of the viability of organoids, organoids were mechanically disrupted and centrifuged at 700 rcf for 5 min. Organoids were seeded either in 50 µl 50% Cultrex/organoid medium in a 96-well plate or in two 10 µl 70% Cultrex/organoid medium drops in a 24-well plate in at least four technical replicates, and the respective organoid medium was added. shRNAs were induced immediately after seeding by the addition of 1 µg/ml of doxycycline. Every second day, the medium was changed and doxycycline was supplemented freshly. A viability assay was performed 6–7 days post-seeding using CellTiter-Blue® (#G8080, Promega) or CellTiter-Glo® 3D (#G9242, Promega).

## eIF2Bα and eIF2Bδ mutant design

To identify possible mutations in eIF2Bα that would disrupt homo-dimerization, a cryo-EM structure (PDB: 6O81) (Kenner et al, 2019) was analyzed using the Molecular Operating Environment (MOE 2019) (ULC CCG, 2019). All chains except G and H were removed. All termini and the 17 residue breaks present in both chains (Y252 to H270) were capped with acetyl and N-methyl caps. A total of 102 missing side chains were added using MOE's structure preparation module. The interactions between the resulting monomers were studied using the Contacts function. The four strongest stabilizing effects were found to be formed from the acidic side chain of E188 to Q243 and N242 in both monomers. Hydrogen bonds formed between the phenolic hydroxyl group of Y185 and K251 were found on ranks five and six of the interaction analysis. Therefore, mutations of E188 and Y185 to alanine were thought to destabilize the homodimer. In addition, visual inspection of the structure revealed that A181 and C218 were mutation sites with high disruption potential when replaced by large, bulky amino acids. Y185A, E188A, A181F, C218F (eIF2Bα$^{mut}$) mutations were introduced into the eIF2Bα CDS.

For mutations that would abolish the interaction between eIF2Bδ and p-eIF2α S51, the corresponding cryo-EM structure (PDB: 6O9Z) was examined (Kenner et al, 2019). All chains except for E and M were removed. As a large number of side chains had to be added, chain E was prepared first (41 side chains) and followed by chain M (149 side chains). Due to the high amount of unresolved amino acids in the structure, numerous clashes were generated during the structure preparation workflow. Therefore, interaction analysis was not performed and efforts were focused on mutations that would potentially disrupt the interaction interface between eIF2Bδ and p-eIF2α. Two alanine residues, A315 and A318, as has been published before (Kenner et al, 2019), were identified in the interface that could potentially affect the interaction if mutated to amino acids with large and bulky side chains. Thus, A315W and A318W mutations were introduced into the eIF2Bδ CDS (eIF2Bδ$^{mut}$).

Mutated CDS (eIF2Bα$^{mut}$, eIF2Bδ$^{mut}$) and corresponding WT CDS (eIF2Bα$^{wt}$, eIF2Bδ$^{wt}$) were cloned into the pRRL-SFFV-puro vector (Doyle et al, 2018). All CDS contained silent mutations in the respective shRNA targeting sites to avoid the knockdown of the exogenous proteins.

## eIF2α^S51A and eIF2α^S51D mutant design

Mutated CDS (eIF2α^S51A, eIF2α^S51D) and corresponding WT CDS (eIF2α^wt) carrying a C-terminal HA-tag were cloned into the pRRL-SFFV-puro vector (Doyle et al, 2018).

## eIF2Bε^R113H mutant design

The vector for eIF2Bε^wt overexpression has been described before (Schmidt et al, 2019). The CDS of this construct was used to introduce the R113H mutation which was integrated into the lentiviral overexpression vector pLV-Bsd_SV40 (VectorBuilder).

## Western blotting

Cells were lysed in RIPA buffer (50 mM Tris pH 7.5, 150 mM NaCl, 1% NP-40, 0.5% DOC, 0.1% SDS) containing protease inhibitors (#P8340, Sigma) and phosphatase inhibitors (#P5726, #P0044, Sigma). Cleared protein lysates, normalized to protein concentrations, were separated by SDS-PAGE and transferred to a PVDF membrane (Millipore). Chemi-luminescent images were taken using LAS3000 imager (Fuji) or Fusion FX (Vilber). Images were scanned in Near-Infrared 700 and 800 channels on an Odyssey scanner (LI-COR).

## eIF2α immunoprecipitation (IP)

Immunoprecipitation of endogenous eIF2α was performed as described previously (Schmidt et al, 2019).

## ^35S-methionine labeling

Cells were cultured in methionine-free medium (#R7513, Sigma) supplemented with 10% dialyzed FBS (#F0392, Sigma) and 1% GlutaMAX™ (#35050061, Thermo Fisher) for 30 min. In total, 1 μCi/ml of ^35S-methionine (#SCIS-103, Hartmann Analytic) was added for 1 h. After NaOH-mediated cell lysis and TCA precipitation incorporated ^35S-methionine was measured by scintillation counting using a 300 SL counter (Hidex). Counts were normalized to protein concentration.

## Puromycin labeling

Puromycin labeling for quantification of global translation was performed according to (Schmidt et al, 2009). Briefly, cells or organoids were treated with 1 μg/ml puromycin for 10 min at 37 °C, followed by washing with PBS and 50 min incubation at 37 °C in normal medium (for organoids only). For cycloheximide controls, cells or organoids were treated with 50 μg/ml cycloheximide for 2 h before puromycin labeling. Then, cells or organoids were washed once with PBS and harvested. Cell pellets were lysed in RIPA buffer, and SDS-PAGE and western blotting was performed as described in the respective section. Membranes were incubated with anti-puromycin antibody.

## Sucrose gradients

Pellets with an equal number of cells were resuspended in 500 μl of gradient lysis buffer (100 mM KCl, 20 mM Tris pH 7.5, 5 mM MgCl_2, 1 mM DTT, 1 mM Leupeptin, 1 mM Pepstatin, 1 mM Aprotinin,

0.1 mM PMSF, 0.1 mM AEBSF, 0.5% NP-40, 100 μg/ml cycloheximide, 40 U/ml RNAsin (Promega)), and incubated for 10 min on ice. Samples were centrifuged for 10 min at 10,000 rpm and 4 °C, and cleared lysates were collected and loaded on 10–30% sucrose gradients. Ultracentrifugation was performed for 18 h at 37,000 rpm and 4 °C in a SW60 rotor (Beckmann Coulter). Fractionation was performed manually, and 24 fractions were collected. For western blotting, pooled fractions were loaded onto gels. Deoxycholate-trichloracetic acid (DOC-TCA) protein precipitation was used for isolation of proteins from sucrose gradient fractions. Two gradient fractions were pooled before precipitation, and 100 μl of 0.15% DOC was added to 1 ml of pooled samples. Samples were vortexed and incubated for 10 min at room temperature. Next, 50 μl of 100% TCA was added, and samples were incubated for 30 min on ice. Fractions were centrifuged at 12,000 rpm for 15 min at 4 °C. Pellets were washed with 500 μl ice-cold acetone to remove remaining TCA, centrifuged at 12,000 rpm for 10 min at 4 °C. All samples were resuspended in equal amount of 2× Laemmli buffer.

## RNA isolation and quantitative PCR (qPCR)

RNA isolation and qPCR were performed as described previously (Schmidt et al, 2019).

## Tissue microarray (TMA) immunohistochemical staining (IHC)

TMAs were prepared and stained as previously described (Maier et al, 2023).

## Ribosomal profiling (Ribo-seq)

Ribo-seq was performed as previously described (Wilczynska et al, 2019), with variations described below. Sample preparation was performed as follows: SW480 cells transduced with inducible non-targeting control (CTRL), *EIF2B1*, or *EIF2B4* shRNAs were grown in the absence (DMSO) or presence of 1 μg/ml doxycycline for 5 days. Two 15-cm dishes per condition were then harvested in cold PBS, washed twice in cold PBS, and the resulting pellet was snap-frozen in liquid nitrogen. One additional 15-cm dish per condition was harvested in parallel to check efficient knockdown by western blotting. The experiment was performed four times in separate weeks. Each replicate was further processed independently.

Pellets for Ribo-seq were lysed in 500 μl of lysis buffer (15 mM Tris-Cl pH 7.5, 150 mM NaCl, 15 mM MgCl_2, 100 μg/ml cycloheximide, 1% Triton X-100, 0.05% Tween-20, 2% n-Dodecyl-beta-Maltoside Detergent (DDM), 0.5 mM DTT, 5 mM NaF, 1× cOmplete mini (#04693124001, Roche)) and then cleared by centrifugation (13,000 rpm, 5 min, 4 °C). An aliquot of 30 μl of undigested cleared lysates was supplemented with 1 ml TRIzol™ (#15596026, Invitrogen) for total cytoplasmic RNA extraction. TRIzol extraction was performed following the manufacturer's instructions. Total cytoplasmic RNAs had an average 9.7 RINe value (RNA ScreenTape, Agilent Technologies) prior to library preparation.

In total, 400 μl of cleared lysate were transferred to a fresh tube and digested with 8 μl of RNAseI (#AM2295, Lot# 00791236, Ambion) at 22 °C for 20 min. Digestion was stopped with 16 μl of SUPERaseIn™ (#AM2696, Invitrogen). Digested lysates were loaded onto a 10–50% sucrose gradient and ultracentrifuged for 2 h at 38,000 rpm at 4 °C in a

Sw40Ti rotor (Beckman Coulter). Spun samples were fractionated on a TRIAX gradientmaster and fractions corresponding to monosomes and disomes were harvested in Acid-Phenol:Chloroform, pH 4.5 (with IAA, 125:24:1, Invitrogen), and chloroform extracted. RNA was then run on a gel and processed as previously described (Wilczynska et al, 2019). No rRNA depletion was performed. Ribosome-protected fragments (RPFs) size were measured on a small RNA bioanalyzer chip (Agilent Technologies) before proceeding to library preparation. RPFs concentration was quantified with RNA High Sensitivity Qubit assays (Invitrogen). RPFs libraries were prepared using the NEXTflex small RNA kit v3 (Perkin Elmer) following the manufacturer's instructions, using 10.5 ng of input and 11 PCR cycles in the amplification step. Alternative Step F and Step H2 were used. Total cytoplasmic RNAs were depleted of rRNA with RiboCop v2 (Lexogen) and then libraries were prepared using the CORALL kit (Lexogen), all following the manufacturer's instructions. 900 µg of total cytoplasmic RNAs were used as input for the RiboCop. 13 PCR cycles were used in the library amplification stage.

RPFs libraries and total libraries were quantified using DNA High Sensitivity Qubit assays (Invitrogen) and size-checked using D1000 High-sensitivity ScreenTape (Agilent Technologies). After equimolar pooling, libraries were sequenced on a NextSeq 500 High Output 75 cycle kit (Illumina). Demultiplexed sequencing results are available at the GEO accession number GSE249128.

### Ribo-seq analysis

Sequencing analysis was performed following the publicly available pipeline at the Bushell's lab GitHub page https://github.com/Bushell-lab/Ribo-seq/tree/main. Custom scripts listed below are available on the github page.

Briefly, total cytoplasmic RNA sequencing data were demultiplexed, adaptors removed, deduplicated, and then aligned to a filtered protein-coding transcriptome (derived from GENECODE, v38) using bowtie2 (Li and Dewey, 2011). Transcript isoforms were quantified using rsem-calculate-expression function. The most expressed transcript per gene was extracted and the associated FASTA file generated. RPFs sequencing data were demultiplexed, adaptors removed, deduplicated and then aligned to rRNAs, tRNAs, and mitochondrial RNAs (GENECODE, v38) to remove all contaminant reads. Lastly, they were aligned to the FASTA of most abundant protein-coding transcripts for SW480 cells, generated by using the total cytoplasmic RNA results mentioned above. RPFs reads were then checked for QC with custom R scripts and then the correct sizes with corresponding offsets were extracted using custom python scripts. Metagene plots were generated with a custom R script for binning, normalizing, and plotting the data. Differential expression analysis was performed with DESeq2 (Love et al, 2014), correcting for shCTRL effects using a multifactorial design that combines the induction of shRNAs (condition: +DOX) with the sh*EIF2B1* or sh*EIF2B4*. Normalized read counts as well as differential expression results obtained using DEseq2 are available as Dataset EV1 (total cytoplasmic RNA normalized reads), Dataset EV2 (RPFs normalized reads), and Dataset EV3 (DE genes and TE analysis). Functional analyses were performed with the packages fgsea (Korotkevich et al, 2021) and enrichr (Wu et al, 2021). Molecular feature analysis for regression and for classification were performed with xgboost (Chen and Guestrin, 2016) and caret (Kuhn, 2008) packages.

### Statistics and reproducibility

Experiments were repeated at least three times unless stated otherwise. As indicated in the figure legends, data are presented as mean ± s.d. or s.e.m. of at least three biological replicates, unless stated otherwise, or as one representative analysis ± s.d. Statistical analyses were performed using Prism 10 (GraphPad), Excel (Microsoft) and R. Statistical significance was tested by unpaired, two-tailed Student's *t* test. A *P* value less than 0.05 was considered statistically significant, unless otherwise stated.

## Data availability

The datasets produced in this study are available in the following databases: Ribo-seq data: Gene Expression Omnibus (GEO) GSE249128.

The source data of this paper are collected in the following database record: biostudies:S-SCDT-10_1038-S44318-025-00381-9.

## Peer review information

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

## Acknowledgements

This study was supported by grants from the Deutsche Forschungsgemeinschaft (DFG) (WI 5037/2-1 and WI 5037/4-1 to AW), the Deutsche Krebshilfe (# 70114050 to AW) and the Wilhelm Sander-Stiftung (# 2021.114.1 to AW). Additional personal financial support was given by Mr. Kratz. We thank Eduard Batlle for providing the murine LAKTP organoids and Owen J Sansom for providing A and AK organoids as well as L17-R-spondin cells. The technical expertise of Sabine Roth, Hecham Marouf and Cornelius Schneider is gratefully acknowledged. MB and CG were funded by CRUK core funding to the CRUK Scotland Institute (A31287) and to MB (A29252). The invaluable support of the Biological Services Unit, the Molecular Technologies Unit, and all the core services at the Cancer Research UK Scotland Institute is greatly appreciated. The manuscript was critically reviewed by Catherine Winchester at the CRUK Scotland Institute.

## Author contributions

**Ivana Paskov Škapik**: Formal analysis; Validation; Investigation; Visualization; Writing—original draft; Writing—review and editing. **Chiara Giacomelli**: Data curation; Software; Formal analysis; Validation; Investigation; Visualization; Writing—original draft; Writing—review and editing. **Sarah Hahn**: Investigation. **Hanna Deinlein**: Investigation. **Peter Gallant**: Software; Writing—review and editing. **Mathias Diebold**: Investigation; Writing—review and editing. **Josep Biayna**: Writing—review and editing. **Anne Hendricks**: Writing—review and editing. **Leon Olimski**: Investigation; Writing—review and editing. **Christoph Otto**: Writing—review and editing. **Carolin Kastner**: Writing—review and editing. **Elmar Wolf**: Resources. **Christina Schülein-Völk**: Formal analysis; Writing—review and editing. **Katja Maurus**: Formal analysis; Investigation. **Andreas Rosenwald**: Resources; Formal analysis; Investigation. **Nikolai Schleussner**: Resources. **Rene-Filip Jackstadt**: Resources; **Nicolas Schlegel**: Resources. **Christoph-Thomas Germer**: Project administration. **Martin Bushell**: Conceptualization; Supervision; Funding acquisition; Writing—original draft; Project administration; Writing—review and editing. **Martin Eilers**: Funding acquisition; Project administration; Writing—review and editing. **Stefanie Schmidt**: Conceptualization; Formal analysis; Supervision; Investigation; Visualization; Writing—original draft; Project administration; Writing—review and editing. **Armin Wiegering**: Conceptualization; Supervision; Funding acquisition; Writing—original draft; Project administration; Writing—review and editing.

Source data underlying figure panels in this paper may have individual authorship assigned. Where available, figure panel/source data authorship is listed in the following database record: biostudies:S-SCDT-10_1038-S44318-025-00381-9.

## Funding

## Disclosure and competing interests statement

The authors declare no competing interests.

# Expanded View Figures

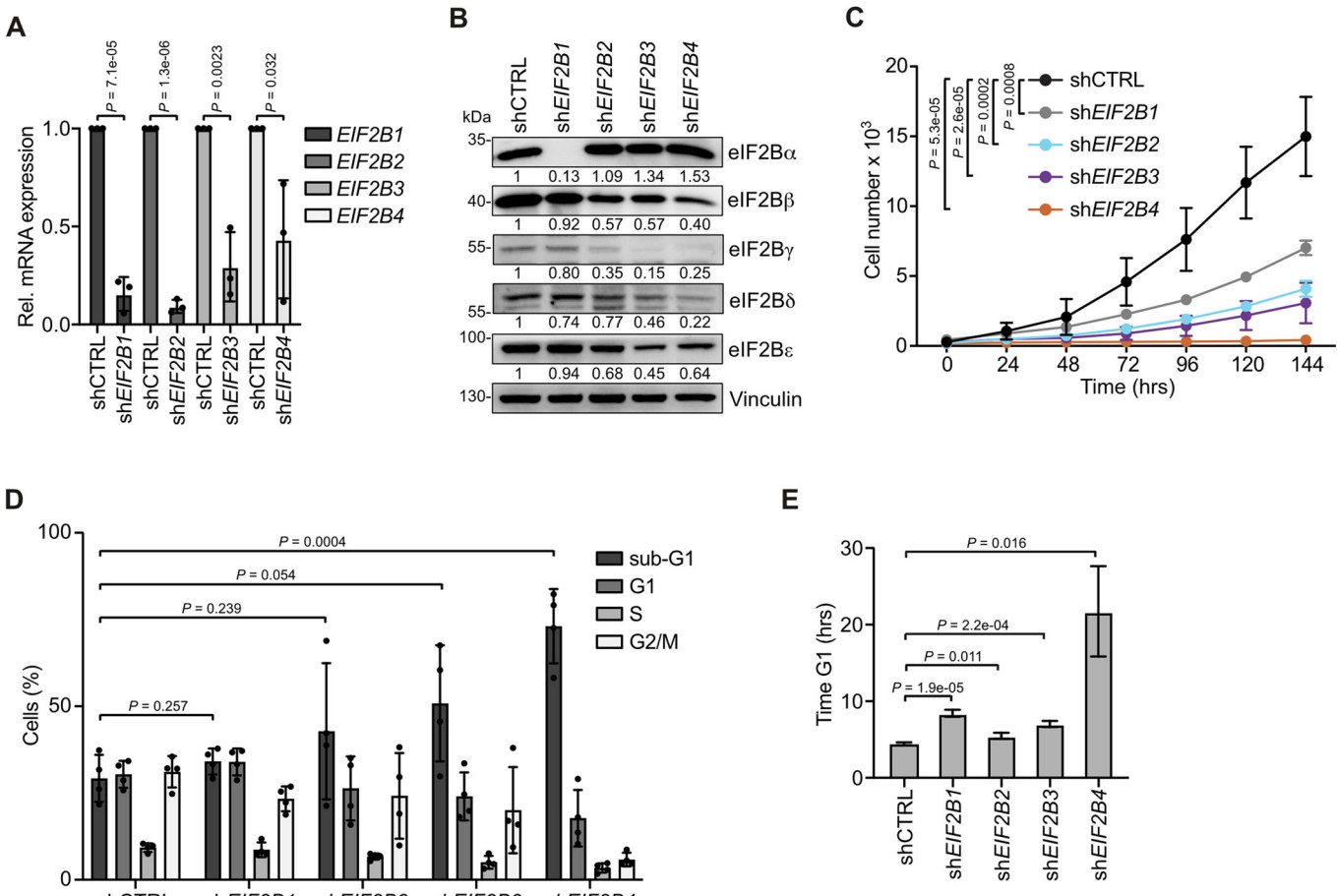

**Figure EV1.  DLD1 cells show reduced viability upon eIF2B subunit depletion.**

(A) mRNA expression of indicated genes in SW480 cells transduced with shCTRL or shRNAs against *EIF2B1-4*. Data show mean ± s.d. ($n = 3$ biological replicates); Student's *t* test. (B) Western blot of indicated proteins in DLD1 cells transduced with shCTRL or shRNAs against *EIF2B1-4*, representative of three biological replicates with similar results. Levels of the respective eIF2B subunits, relative to vinculin, are given below each corresponding panel. (C) Growth curve of DLD1 cells transduced as described in (B), measured with Operetta screening microscope. Data show mean ± s.d. ($n = 6$ biological replicates); Student's *t* test. (D) PI cell cycle FACS analysis of DLD1 cells transduced as described in (B). Data show mean ± s.d. ($n = 4$ biological replicates); Student's *t* test. (E) Length of G1 cell cycle phase of DLD1 cells transduced as described in (B), calculated with data acquired from growth curve in (C) and PI cell cycle FACS in (D). Data show mean ± s.e.m.; Student's *t* test. Source data are available online for this figure.

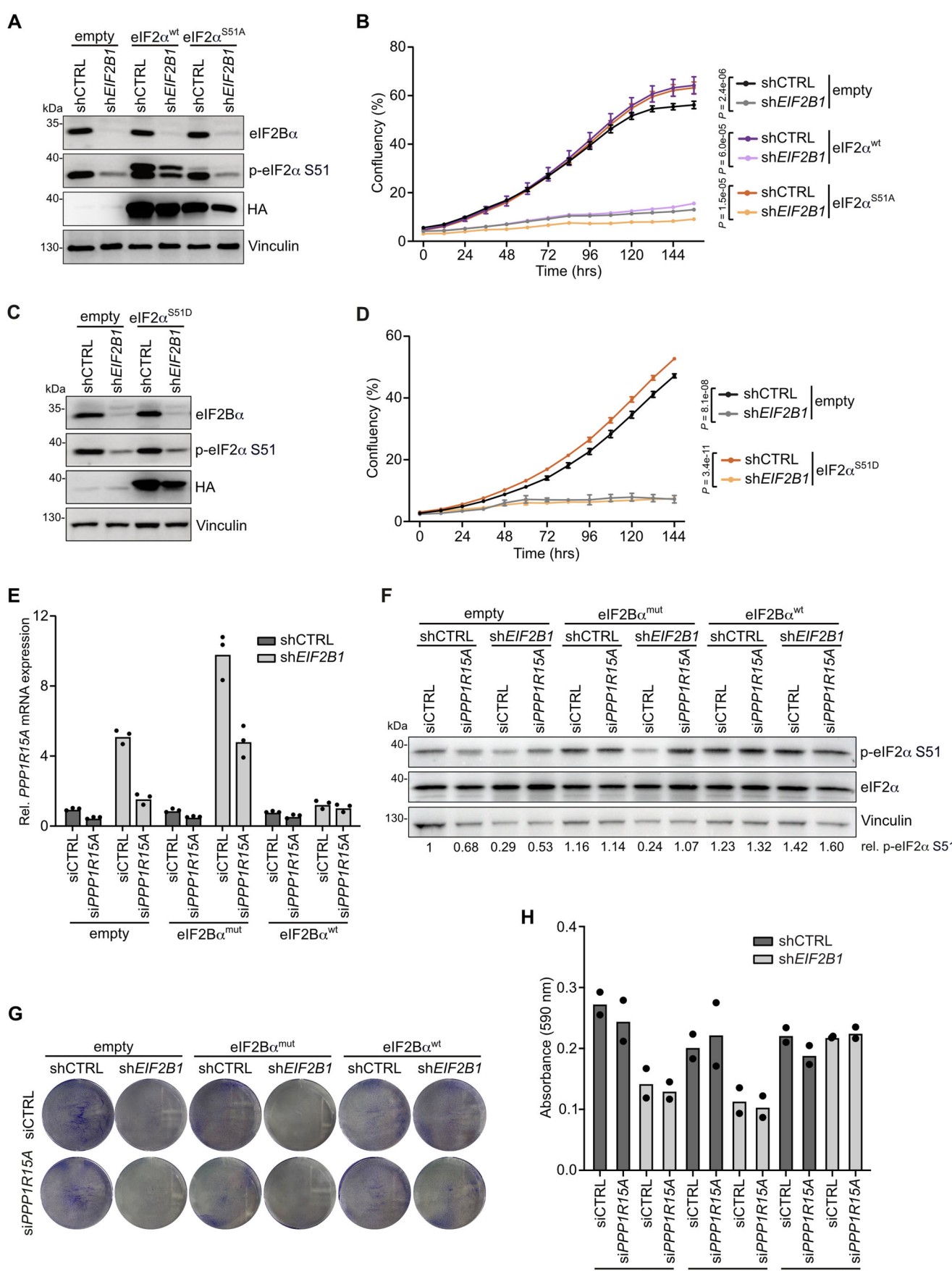

◀  **Figure EV2.   Restoration of eIF2α phosphorylation does not rescue the viability defect upon eIF2Bα modulation.**

(A) Western blot of indicated proteins in shCTRL- or sh*EIF2B1*-transduced SW480 cells stably overexpressing eIF2α WT (eIF2α^wt), eIF2α S51A mutant (eIF2α^S51A) construct, or without any overexpression (empty). The western blot is representative of three biological replicates with similar results. (B) Growth curve of SW480 cells transduced as described in (A), measured with Incucyte® live-cell imaging system. Data show mean ± s.d. ($n = 4$ biological replicates); Student's $t$ test. (C) Western blot of indicated proteins in shCTRL- or sh*EIF2B1*-transduced SW480 cells stably overexpressing eIF2α S51D mutant (eIF2α^S51D) construct, or without any overexpression (empty). The western blot is representative of three biological replicates with similar results. (D) Growth curve of SW480 cells transduced as described in (C), measured with Incucyte® live-cell imaging system. Data show mean ± s.d. ($n = 4$ biological replicates); Student's $t$ test. (E) mRNA expression of *PPP1R15A* in shCTRL- or sh*EIF2B1*-transduced SW480 cells stably overexpressing eIF2Bα mutant (eIF2Bα^mut), eIF2Bα WT (eIF2Bα^wt) construct, or without any overexpression (empty), transfected with siCTRL or si*PPP1R15A* for 72 hr. Data show mean of technical triplicates of one representative experiment ($n = 2$ biological replicates). (F) Western blot of indicated proteins in SW480 cells transduced and transfected as described in (E). The western blot is representative of two biological replicates with similar results. Levels of p-eIF2α S51, relative to total eIF2α, are given below. (G) Crystal violet staining of SW480 cells transduced and transfected as described in (E). Staining was done 7 days after seeding. Pictures are representative of two biological replicates with similar results. (H) Quantification of crystal violet staining described in (G). Data show mean ($n = 2$ biological replicates). Source data are available online for this figure.

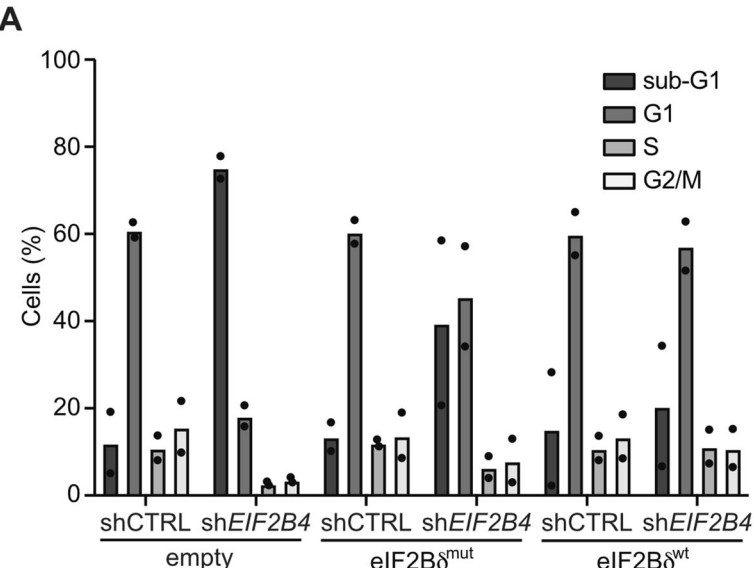

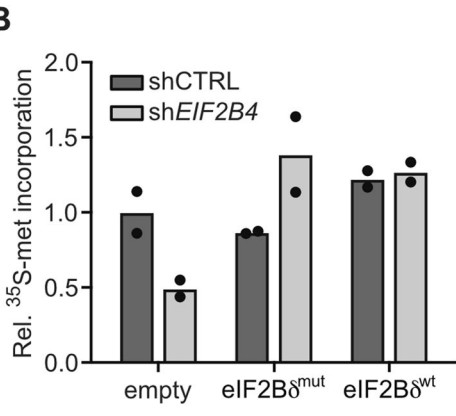

**Figure EV3.** **eIF2Bδ^mut expression rescues protein synthesis rates but not the viability defect upon eIF2Bδ depletion.**

(A) PI cell cycle FACS analysis of shCTRL- or sh*EIF2B4*-transduced SW480 cells stably overexpressing eIF2Bδ mutant (eIF2Bδ^mut), eIF2Bδ WT (eIF2Bδ^wt) construct, or without any overexpression (empty). Data show mean ($n = 2$ biological replicates). (B) Relative ^35S-methionine incorporation of SW480 cells transduced as described in (A). Data show mean ($n = 2$ biological replicates). Source data are available online for this figure.

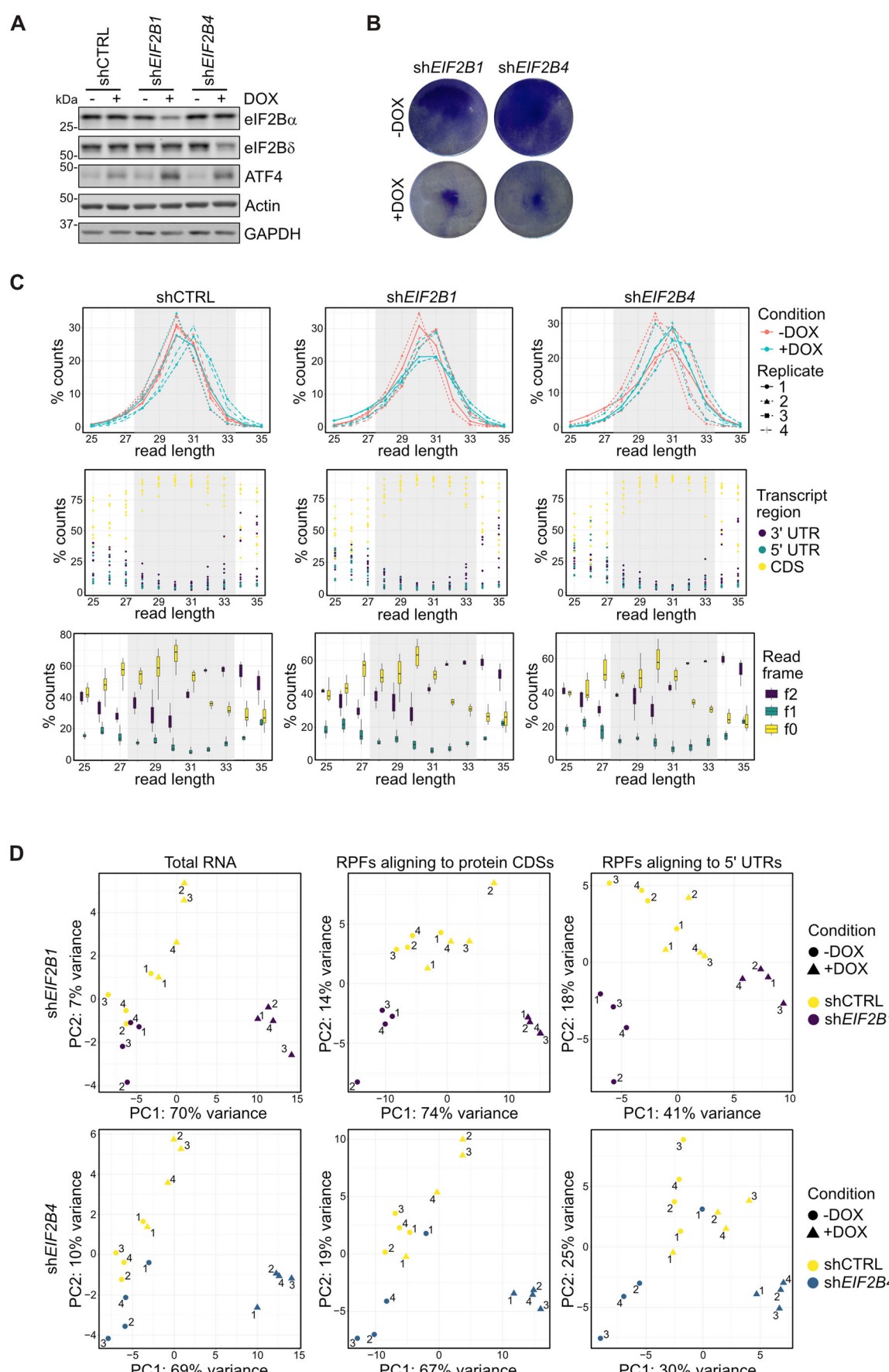

◀  **Figure EV4.  Ribo-seq quality controls (QC).**

(A) Western blot of indicated proteins in SW480 cells transduced with doxycycline-inducible shCTRL, sh*EIF2B1* or sh*EIF2B4* (5 days of doxycycline treatment ( + DOX), DMSO as control (-DOX)), representative of four biological replicates with similar results. GAPDH is probed on the same membrane as eIF2Bδ, and is used as its loading control; Actin is probed on the same membrane as ATF4 and eIF2Bα, and is used as loading control for these two targets. (B) Crystal violet staining of SW480 cells transduced with doxycyline-inducible sh*EIF2B1* or sh*EIF2B4* after 7 days of DOX treatment ( + DOX, EtOH as control (-DOX)). The experiment was performed once. (C) Ribo-seq QC plots; the gray shaded area in all graphs indicates reads of length 28–33 nt, representing the extracted reads for downstream analyses. Columns show results from cells transduced with respective doxycyline-inducible shRNA. The first row shows distribution of ribosome-protected fragments (RPFs) reads for -DOX and +DOX samples, with four replicates plotted independently. The second row shows the transcript region where the RPF reads align to; each dot represents an individual sample. The third row shows the read frame respective to the known codon position within the RPFs. Box plot summary statistics: boxes' lower and upper hinges represent first and third quantiles, lines represent the median, whiskers extend until last data point to a maximum of 1.5* inter-quantile range. (D) PCA plots for total cytoplasmic RNA sequencing (first column), RPFs aligning to protein-coding sequences (second column), or RPFs aligning to the 5' UTRs (third column). Data from each shRNA against either of the two subunits is plotted with the data for the shCTRL (yellow in all plots). Variance is displayed on each axis; size of axes is not variance-scaled. Source data are available online for this figure.

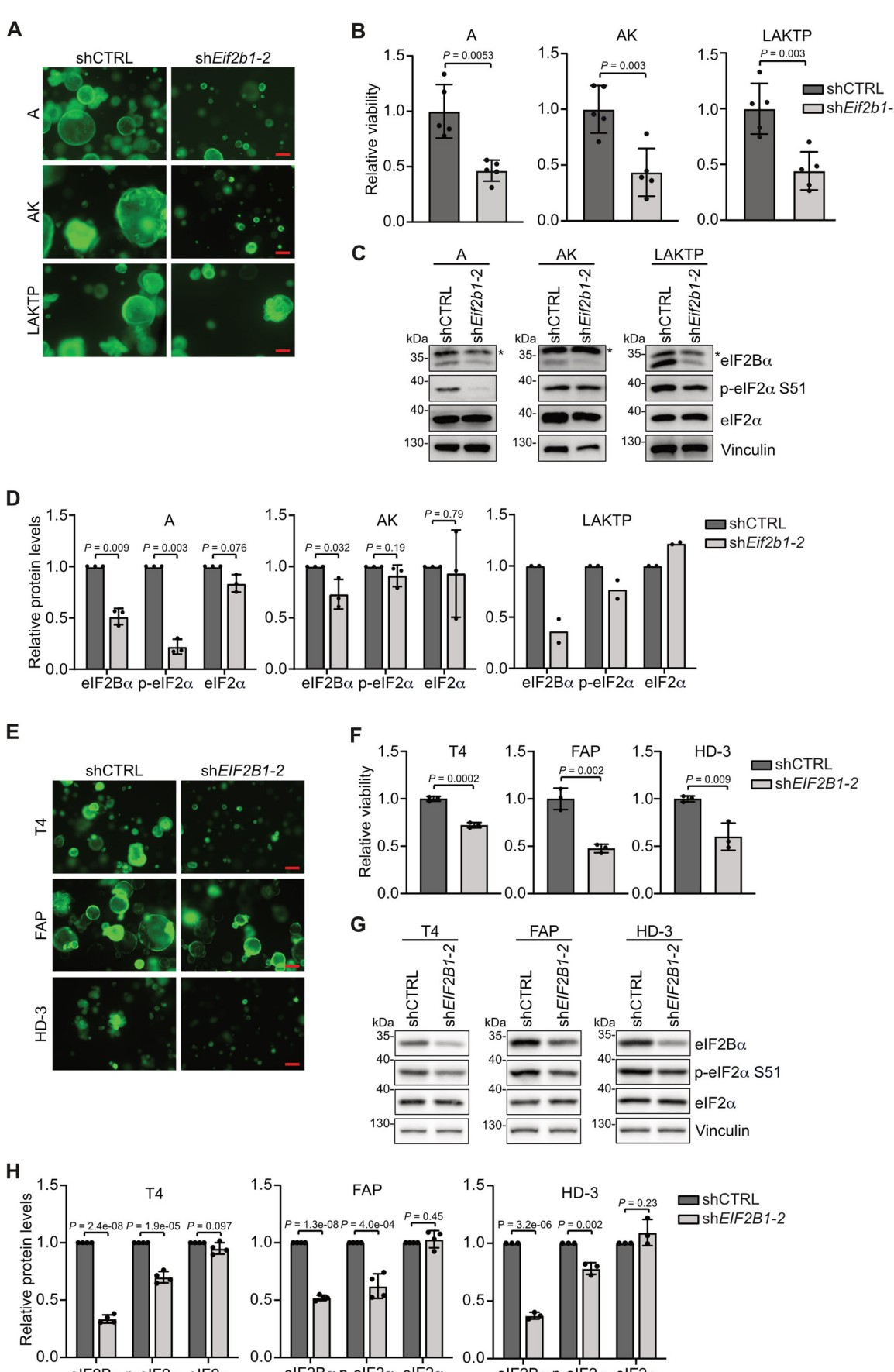

◀ **Figure EV5. Depletion of eIF2Bα by a second independent shRNA reduces viability of murine and human tumor organoids.**

(**A**) Pictures of murine A, AK and LAKTP intestinal organoids transduced with doxycycline-inducible shCTRL or sh*Eif2b1-2* (7 days of doxycycline treatment), representative of five biological replicates with similar results. Green signal (GFP) indicates shRNA induction. Scale bar = 200 μm. (**B**) Relative viability of A, AK and LAKTP organoids transduced and treated as described in (**A**). Data show mean ± s.d. (*n* = 5 biological replicates); Student's *t* test. (**C**) Western blot of indicated proteins in A, AK and LAKTP organoids transduced as described in (**A**), representative of two or three biological replicates with similar results (96 h of doxycycline treatment); *unspecific bands. (**D**) Quantification of eIF2Bα, p-eIF2α S51 and total eIF2α levels, normalized to vinculin, of western blots described in (**C**). Data show mean ± s.d. (*n* = 2 or 3 biological replicates); Student's *t* test. (**E**) Pictures of human T4, FAP, HD-3 PDOs transduced with doxycycline-inducible shCTRL or sh*EIF2B1-2* (7 days of doxycycline treatment), representative of three biological replicates with similar results. Green signal (GFP) indicates shRNA induction. Scale bar = 200 μm. (**F**) Relative viability of T4, FAP, HD-3 PDOs transduced and treated as described in (**E**). Data show mean ± s.d. (*n* = 3 biological replicates); Student's *t* test. (**G**) Western blot of indicated proteins in T4, FAP, HD-3 PDOs transduced as described in (**E**), representative of three or four biological replicates with similar results (96 h of doxycycline treatment). (**H**) Quantification of eIF2Bα, p-eIF2α S51 and total eIF2α levels, normalized to vinculin, of western blots described in (**G**). Data show mean ± s.d. (*n* = 3 or 4 biological replicates); Student's *t* test. Source data are available online for this figure.

