## [Peer Review File · The EMBO Journal]

Maintenance of p-eIF2 α levels by the eIF2B complex is vital for colorectal cancer

Ivana Paskov Škapik, Chiara Giacomelli, Sarah Hahn, Hanna Deinlein, Peter Gallant, Mathias Diebold, Josep Biayna, Anne Hendricks, Leon Olimski, Christoph Otto, Carolin Kastner, Elmar Wolf, Christina Schüle-Völk, Katja Maurus, Andreas Rosenwald, Nikolai Schleussner, Rene-Filip Jackstadt, Nicolas Schlegel, Christoph Germer, Martin Bushell, Martin Eilers, Stefanie Schmidt, and Armin Wiegering

Corresponding authors: Armin Wiegering (Wiegering_A@ukw.de) , Stefanie Schmidt (schmidt_s12@ukw.de)

Review Timeline:

Submission Date:	19th Feb 24
Editorial Decision:	19th May 24
Revision Received:	29th Sep 24
Editorial Decision:	22nd Dec 24
Revision Received:	17th Jan 25
Accepted:	23rd Jan 25

Editor: Daniel Klimmeck

Transaction Report:

Dear Dr. Wiegering,

Thank you for the submission of your manuscript (EMBOJ-2024-116846) to The EMBO Journal and in addition providing us with a preliminary revision plan. As mentioned earlier your study was assessed by three reviewers with expertise in protein translation and cancer biology, and I enclose their comments below.

As you will see from the reviewers' reports, they acknowledge the analysis and potential interest and value of your findings. However, they also express major concerns regarding insufficient mechanistic detail provided on the context and regulation of enhanced eIF2alpha phosphorylation and its sensing by eIF2B. Further, they request clarification on specificity of this scenario for APC-driven colorectal cancer. Further, the reviewers raise a number of points related to the presentation of the findings, additional controls required, improved methods annotation and data processing and overall discussion of related literature, that would need to be conclusively addressed to achieve the level of robustness and clarity needed for The EMBO Journal.

In light of the overall interest stated and broader angle of your findings, we are able to invite you to revise your manuscript experimentally to address the referees' comments, along the lines detailed in your preliminary response. I need to stress though that we do require strong support from the referees on a revised version of the study in order to move on to publication of the work.

Please feel free to also contact me if you have any questions or need further input on the referee comments.

When submitting your revised manuscript, please carefully review the instructions below.

Please feel free to approach me any time should you have additional questions related to this.

Thank you for the opportunity to consider your work for publication.

I look forward to your revision.

Kind regards,

Daniel Klimmeck

Daniel Klimmeck, PhD
Senior Editor
The EMBO Journal

Instruction for the preparation of your revised manuscript:

2) individual production quality figure files as .eps, .tif, .jpg (one file per figure).

3) a .docx formatted letter INCLUDING the reviewers' reports and your detailed point-by-point response to their comments. As part of the EMBO Press transparent editorial process, the point-by-point response is part of the Review Process File (RPF), which will be published alongside your paper.

4) a complete author checklist, which you can download from our author guidelines ([https://wol-prod-cdn.literatumonline.com/pb-assets/embo-site/Author Checklist%20-%20EMBO%20J-1561436015657.xlsx](https://wol-prod-cdn.literatumonline.com/pb-assets/embo-site/Author%20Checklist%20-%20EMBO%20J-1561436015657.xlsx)). Please insert information in the checklist that is also reflected in the manuscript. The completed author checklist will also be part of the RPF.

6) It is mandatory to include a 'Data Availability' section after the Materials and Methods. Before submitting your revision, primary datasets produced in this study need to be deposited in an appropriate public database, and the accession numbers and database listed under 'Data Availability'. Please remember to provide a reviewer password if the datasets are not yet public (see <https://www.embopress.org/page/journal/14602075/authorguide#datadeposition>).

7) Our journal encourages inclusion of *data citations in the reference list* to directly cite datasets that were re-used and obtained from public databases. Data citations in the article text are distinct from normal bibliographical citations and should directly link to the database records from which the data can be accessed. In the main text, data citations are formatted as follows: "Data ref: Smith et al, 2001" or "Data ref: NCBI Sequence Read Archive PRJNA342805, 2017". In the Reference list, data citations must be labeled with "[DATASET]". A data reference must provide the database name, accession number/identifiers and a resolvable link to the landing page from which the data can be accessed at the end of the reference. Further instructions are available at .

8) At EMBO Press we ask authors to provide source data for the main and EV figures. Our source data coordinator will contact you to discuss which figure panels we would need source data for and will also provide you with helpful tips on how to upload and organize the files.

Numerical data can be provided as individual .xls or .csv files (including a tab describing the data). For 'blots' or microscopy, uncropped images should be submitted (using a zip archive or a single pdf per main figure if multiple images need to be supplied for one panel). Additional information on source data and instruction on how to label the files are available at .

9) We replaced Supplementary Information with Expanded View (EV) Figures and Tables that are collapsible/expandable online (see examples in <https://www.embopress.org/doi/10.15252/embj.201695874>). A maximum of 5 EV Figures can be typeset. EV Figures should be cited as 'Figure EV1, Figure EV2" etc. in the text and their respective legends should be included in the main text after the legends of regular figures.

11) For data quantification: please specify the name of the statistical test used to generate error bars and P values, the number (n) of independent experiments (specify technical or biological replicates) underlying each data point and the test used to calculate p-values in each figure legend. The figure legends should contain a basic description of n, P and the test applied. Graphs must include a description of the bars and the error bars (s.d., s.e.m.).

We realize that it is difficult to revise to a specific deadline. In the interest of protecting the conceptual advance provided by the work, we recommend a revision within 3 months (17th Aug 2024). Please discuss the revision progress ahead of this time with the editor if you require more time to complete the revisions.

Referee #1:

In this study, Skapik and colleagues demonstrate that p-eIF2alpha is elevated in APC deficient cells, and its proper sensing by the decameric eIF2B complex are essential to balance translation. Knockdown or mutation of eIF2Balpha and eIF2Bdelta, two eIF2B subunits responsible for sensing p-eIF2alpha, impairs CRC viability, demonstrating that the eIF2B/p-eIF2alpha axis is critical in CRC. The authors further show that decameric eIF2B is critical for translating growth-promoting mRNAs that are induced upon APC loss, although the mechanism is not clear. Finally, depletion of eIF2Balpha in APC-deficient murine and patient-derived organoids establishes a therapeutic window and identifies a possible strategy to target CRC by interfering with the eIF2B/p-eIF2alpha interaction via eIF2Balpha. Overall, this study is clearly described, and the manuscript is well-written. However, there are several concerns that need to be addressed and additional experiments added to merit publication in the EMBO Journal. Major and minor comments are included below:

Major Comments:

1. What is the mechanism of increased eIF2alpha phosphorylation in APC deficient cells?
2. The authors should provide quantification of western blots in Supplementary Figure S1A and S1C.
3. Cell cycle analysis in Figure 2F is unclear. The authors state that cells depleted of eIF2Balpha were characterized by a prolonged G1 cell cycle phase, yet all cells shown appear to have a prolonged G1 cell cycle phase. The authors should replicate this in a different CRC cell line to ensure reproducibility of results.
4. Do the authors think that interfering with the function of the eIF2B complex to sense phosphorylation of eIF2alpha (by either mutating eIF2Bdelta or modulating eIF2Balpha), induces a vulnerability in tumor types other than CRC? They may want to elaborate on this point in the Discussion.

Minor Comments:

1. Have the two CRC patient-derived organoids T4 and HD-3 been characterized? What is their oncogenotype? Was there rationale for selecting these particular PDOs? If so, this information should be provided.

Referee #2:

Skapik and colleagues have presented evidence indicating that disrupting the eIF2B complex through genetic methods, such as siRNA or shRNA, in CRC cell lines results in the suppression of eIF2alpha phosphorylation (p-eIF2) and inhibits tumor cell growth. Despite lacking a mechanistic explanation for this intriguing finding, the authors propose a hypothesis suggesting that eIF2B integrity enables tumor cells to sense p-eIF2 levels. Since p-eIF2 is linked to promoting tumor growth in CRC and various other tumor types, targeting eIF2B could potentially reduce tumor reliance on p-eIF2 for survival and stress adaptation.

Employing sophisticated methodologies, the researchers demonstrate that the alpha and delta subunits of eIF2B are potential candidates for this purpose. By disrupting either the eIF2B alpha homodimerization or the eIF2Bdelta:p-eIF2 interface, the authors illustrate the involvement of both subunits in regulating p-eIF2 and the translation of genes associated with tumorigenic pathways.

The study presents numerous experiments with intriguing findings, but it does not thoroughly investigate the role of p-eIF2 in this context. While the reduction of p-eIF2 is correlated with the proliferative and translational effects described, the study leaves open the possibility of a moonlighting function of eIF2B, as suggested in the text. One key question is how eIF2B disruption functions in cells with a phosphorylation-defective eIF2, such as the Ser52Ala mutant. It would be expected that the loss of p-eIF2 would decrease, but not completely abolish, cell growth, allowing for the analysis of the effects of eIF2B disruption on cell proliferation and protein synthesis. Alternatively, it would be informative to examine the effects of eIF2B disruption in cells where

p-eIF2 is induced by treatment with eIF2-specific phosphatase inhibitors or through the expression of a phosphomimic eIF2alpha Ser52Asp mutant.

The study does not clearly establish whether the effects of eIF2B downregulation on the inhibition of p-eIF2 are specific to CRC tumors. For instance, eIF2Balpha knockdown does not seem to have detectable effects on p-eIF2 in K562 cells, as reported in Schoof et al. *eLife* 2021;10:e65703.

As a final note, while experiments with cultured cells indicate the inability of ISRIB to impair the proliferation of CRC cells (Figure 1), there have been numerous studies, nicely cited in the manuscript, demonstrating the anti-tumor effects of ISRIB in mouse models of cancer. It might be of interest to compare the growth of CRC tumor cells shown in Fig. 3B or 4B with the growth of parental cells treated with ISRIB in mice. This comparison could help determine whether targeting eIF2Balpha or eIF2Bdelta is indeed a superior approach to ISRIB treatment for inhibiting CRC tumor growth.

Minor comment: It would be of interest to determine whether the GEF activity of eIF2B indeed plays a role in the reduction of p-eIF2 by the disruption of eIF2B complex. Figure 2 would be better with inclusion of eIF2Bepsilon KD in parallel with other subunits. Can the authors test the effects of a catalytic mutant eIF2Bepsilon in a similar approach shown in Figure 3B or 4B? It would be great.

Referee #3:

In their study, Skapik et al. propose that the targeting of eIF2B subunits, in particular eIF2B1 provides a potential therapeutic strategy for the treatment of CRC, especially when APC-deficient. On the remarkable fact that APC-deficient CRC is somewhat addicted to elevated eIF2a phosphorylation, the authors focus on the eIF2B complex, which is essential for translation initiation and the sensing of eIF2alpha phosphorylation. Two subunits, eIF2B1 and eIF2B4, are thought to sense this phosphorylation. By applying a transient depletion recovery analysis the authors claim that manipulating the role of both the factors by abundance or mutation of the phosphorylation sensing surface provides a vulnerability of CRC-derived organoids. Although the sensing function is not monitored directly, the viability studies support at least a pivotal role of eIF2B1/4 in the process. Settling on this, the authors proceed with exploring distinct translation deregulation by depleting eIF2B1 vs 4. These studies suggest that mRNAs encoding factors involved in promoting proliferation are particularly susceptible to manipulating eIF2B1. In sum this leads the authors to propose that this may provide a therapeutic window for CRC treatment. In sum the study provides interesting insights, but various aspects require further investigations, more controls and need to address the major concerns raised in the following:

1. Studies presented in Figs.1-4 mainly address the potentially distinct phenotypic roles of eIF2B subunits with a focus on eIF2B1/4. However, these studies do not provide substantial insights relevant for the remainder of the studies where only depletion studies are utilized. This is even more severe since the sensing of eIF2alpha is not directly addressed and the connection between monitoring this modification and sensing remains largely elusive. Thus, despite a huge, admirable experimental effort, there is little to learn from studies presented in Figs.2-4.
2. A general problem of the study is that only one shRNA is used for depletion studies. Thus, how can the authors exclude - except for recovery studies in Figs.2-4 but not later (!) - bias by off-target effects. Along these lines, the authors demonstrate that depleting eIF2B2-4 disrupt the stability of other factors except eIF2B1 (Fig.2). This essentially limits the conclusiveness of findings and raises the question why this is not seen for eIF2B1 knockdown. Moreover, studies presented after Fig.4 require at least one additional, independent shRNA.
3. In Fig.3B the mutant clearly shows lower expression. This could as well explain modestly affected (reduced by ~40%) eIF2alpha phosphorylation. Moreover, errors are missing in the quantification of eIF2alpha phosphorylation, which should be assessed together with monitoring changes in the overall protein abundance of eIF2alpha.
4. Puzzling is why the authors do not present the same analyses in Fig.3/4, in particular altered proliferation rates (see Fig.3E) but only present the percentage of dead cells in Fig.4. These analyses require further in-depth analyses to evaluate how eIF2B1 vs 4 influence cell cycle progression, apoptosis, and organoid growth.
5. The ribosomal profiling studies are very interesting but surprising given the fact that both factors are claimed essential for sensing eIF2alpha phosphorylation by prior presented studies and have strong phenotypic impact, at least on the number of dead cells. How come then that eIF2B4 depletion has comparatively marginal effect on translation? Is this potentially a problem of higher apoptosis and the lack of these cells in polysomal analyses?
6. The provided GSEA analyses (Fig.5) apparently support a role of disturbed translation on overall cell viability but how these data explain the strong impact of eIF2B4 on apoptosis (?) or viability remains unclear. For eIF2B1 depletion the studies appear reasonable, but some gene sets show remarkable differences, e.g. MYC Targets V1 vs V2, why? Unfortunately, the ribosome profiling studies at best describe observations which would require further in-depth testing of at least some candidate effectors claimed for eIF2B1/4 and need to connect more precisely to more detailed phenotypic characterizations.
7. How the data presented in Fig.6 suggest eIF2B1 as a candidate target for CRC treatment remains vague for various reasons. (1) At least two shRNAs should be used for these studies; (2) the depletion of eIF2B4 should be used as a control since this would be expected to greatly impact on viability as well; (3) the change in viability on A, AK, LAKTP models versus wild type is obvious but could be explained by substantial distinct growth rates - also note the huge errors in the wild type population; (4)

Major concerns are raised by Fig.6C in the AK model where barely any depletion of eIF2B1 is observed yet strong effects are reported; (5) finally, if eIF2B1 is a promising therapeutic target one would expect to have tests on other non CRC-derived cells (in addition to WT organoids) to exclude a broad toxicity of eIF2B1 targeting.

Point-by-point reply to Reviewers' Comments EMBOJ-2024-116846

We would very much like to thank all reviewers for their detailed and insightful comments and for their effort in helping us to further improve the manuscript. Please find our detailed responses below.

Referee #1:

In this study, Skapik and colleagues demonstrate that p-eIF2alpha is elevated in APC deficient cells, and its proper sensing by the decameric eIF2B complex are essential to balance translation. Knockdown or mutation of eIF2Balpha and eIF2Bdelta, two eIF2B subunits responsible for sensing p-eIF2alpha, impairs CRC viability, demonstrating that the eIF2B/p-eIF2alpha axis is critical in CRC. The authors further show that decameric eIF2B is critical for translating growth-promoting mRNAs that are induced upon APC loss, although the mechanism is not clear. Finally, depletion of eIF2Balpha in APC-deficient murine and patient-derived organoids establishes a therapeutic window and identifies a possible strategy to target CRC by interfering with the eIF2B/p-eIF2alpha interaction via eIF2Balpha. Overall, this study is clearly described, and the manuscript is well-written. However, there are several concerns that need to be addressed and additional experiments added to merit publication in the EMBO Journal. Major and minor comments are included below:

Major comments:**1. What is the mechanism of increased eIF2alpha phosphorylation in APC deficient cells?**

Colorectal cancer cells deficient of APC are characterized by deregulated high MYC expression. In our previous publication (Schmidt *et al*, 2019), we established MYC-dependent GCN2 activation and concurrent eIF2 α phosphorylation as feedback that couples MYC-driven RNA synthesis to the availability of amino acids (see Fig. 7a, 8e in Schmidt *et al*, 2019). In addition, we observed that MYC also contributes to PKR activation (Fig. 7a in Schmidt *et al*, 2019). Inhibition of both kinases with the respective small molecule inhibitors A-92 (GCN2i) and C16 (PKRi) reduced p-eIF2 α levels and the viability of APC-deficient SW480 CRC cells as well as mouse and patient-derived organoids, whereas we did not see any effects upon inhibition of PERK with the inhibitor GSK2606414 (Fig. 7,8 and ED Fig. 10 in Schmidt *et al*, 2019). These results were also validated by using shRNAs against PKR and GCN2 (ED Fig. 9 in Schmidt *et al*, 2019). We included this in more detail in the discussion of the revised manuscript (Word file: page 19, line 480ff; merged PDF file: page 20, 488ff).

2. The authors should provide quantification of western blots in Supplementary Figure S1A and S1C.

We included the respective quantifications below the Western blots in new Appendix Fig. S1A,C (previous Supplementary Fig. S1). Furthermore, to strengthen the data we increased the number of replicates for Appendix Fig. S1A,C to $n = 2$ biological replicates.

3. Cell cycle analysis in Figure 2F is unclear. The authors state that cells depleted of eIF2Balpha were characterized by a prolonged G1 cell cycle phase, yet all cells shown appear to have a prolonged G1 cell cycle phase. The authors should replicate this in a different CRC cell line to ensure reproducibility of results.

The length of the cell cycle and the respective cell cycle phases differs greatly between different cells. Typically, rapidly dividing eukaryotic cells with a total cell cycle duration of 24 hrs stay in the G1 phase for approximately 11 hrs (Cooper, 2000). Accordingly, as shown in new Fig. 2G (previous Fig. 2F), SW480 cells transduced with non-targeting shCTRL have a mean length of the G1 cell cycle phase of 11.2 hrs. This is comparable to previous cell cycle analysis of another CRC cell line, e.g. Ls174t, that show a G1 phase length of 9.6 hrs (Peter *et al*, 2014). Cells depleted of eIF2B subunits $\beta\gamma\delta$ do not show significant changes in G1 length (shEIF2B2: 7.3 hrs, shEIF2B3: 12.1 hrs, shEIF2B4: 11.4 hrs), whereas only cells depleted of eIF2B α show a significant prolongation of the G1 phase to 21.6 hrs.

For clarification, we adjusted the graph in new Fig. 2G (previous Fig. 2F) and only show the G1 phase with respective P values. Additionally, we explain it more precisely in the revised text (page 7, line 167ff). Furthermore, to ensure reproducibility, we replicated this experiment in a second APC-deficient CRC cell line, DLD1, which show comparable reduction in cell viability after depletion of the respective eIF2B subunits (Figure EV1B-E; Word file: page 7, line 170ff, merged PDF file: page 7, line 173ff). More specifically, eIF2B δ knockdown induced high percentage of cell death, whereas eIF2B α depletion only affected cell cycle progression, which is similar to the effects observed in SW480 cells.

4. Do the authors think that interfering with the function of the eIF2B complex to sense phosphorylation of eIF2 α (by either mutating eIF2B δ or modulating eIF2B α), induces a vulnerability in tumor types other than CRC? They may want to elaborate on this point in the Discussion.

Enhanced levels of eIF2 α phosphorylation have been described in other solid tumor entities (e.g. lung cancer, PDAC, breast cancer) pointing to the possibility that modulation of the eIF2B complex might also induce a vulnerability in other types of cancer (Bai *et al*, 2021; Ghaddar *et al*, 2021; Koromilas, 2015; Shin *et al*, 2022). We elaborated on this in the discussion of the revised manuscript (Word file: page 19, line 468ff, merged PDF file: page 20, line 476ff). Additionally, we analyzed the effects of eIF2B α modulation in a pancreatic cancer setting by using the PDAC cell line PaTu8988T, validating that interfering with the function of the eIF2B complex to sense p-eIF2 α by depletion or mutation of eIF2B α indeed reduces the viability of PDAC cells in a similar manner as observed in CRC (Appendix Fig. S2B,C; Word file: page 9, line 205ff; merged PDF file: page 9, line 208ff).

Minor comments:

1. Have the two CRC patient-derived organoids T4 and HD-3 been characterized? What is their oncogenotype? Was there rationale for selecting these particular PDOs? If so, this information should be provided.

Both PDOs T4 and HD-3 are characterized and sequenced. T4 PDO has already been used in our previous publication (Schmidt *et al.*, 2019). Sequencing results of HD-3 are provided in Appendix Table S1 in this revised manuscript. We selected these two PDOs as they are both APC- and PI3KCA-mutated and they show similar and reproducible growth characteristics. We included this information in the revised manuscript (Word file: page 6, line 139ff; merged PDF file: page 6, line 140ff).

Referee #2:

Skapik and colleagues have presented evidence indicating that disrupting the eIF2B complex through genetic methods, such as siRNA or shRNA, in CRC cell lines results in the suppression of eIF2 α phosphorylation (p-eIF2) and inhibits tumor cell growth. Despite lacking a mechanistic explanation for this intriguing finding, the authors propose a hypothesis suggesting that eIF2B integrity enables tumor cells to sense p-eIF2 levels. Since p-eIF2 is linked to promoting tumor growth in CRC and various other tumor types, targeting eIF2B could potentially reduce tumor reliance on p-eIF2 for survival and stress adaptation.

Employing sophisticated methodologies, the researchers demonstrate that the alpha and delta subunits of eIF2B are potential candidates for this purpose. By disrupting either the eIF2B alpha homodimerization or the eIF2B δ :p-eIF2 interface, the authors illustrate the involvement of both subunits in regulating p-eIF2 and the translation of genes associated with tumorigenic pathways.

Major comments:

1. The study presents numerous experiments with intriguing findings, but it does not thoroughly investigate the role of p-eIF2 in this context. While the reduction of p-eIF2 is correlated with the proliferative and translational effects described, the study leaves open the possibility of a moonlighting

function of eIF2B, as suggested in the text. One key question is how eIF2B disruption functions in cells with a phosphorylation-defective eIF2, such as the Ser52Ala mutant. It would be expected that the loss of p-eIF2 would decrease, but not completely abolish, cell growth, allowing for the analysis of the effects of eIF2B disruption on cell proliferation and protein synthesis. Alternatively, it would be informative to examine the effects of eIF2B disruption in cells where p-eIF2 is induced by treatment with eIF2-specific phosphatase inhibitors or through the expression of a phosphomimic eIF2alpha Ser52Asp mutant.

To address this important comment, we performed the following experiments:

1) To investigate the role of p-eIF2 α in the context of eIF2B disruption by depletion of eIF2B α more precisely, we performed experiments with SW480 cells stably overexpressing HA-tagged eIF2 α WT (eIF2 α^{wt}), the phospho-dead eIF2 α S51A mutant (eIF2 α^{S51A}) or the phospho-mimicking eIF2 α S51D mutant (eIF2 α^{S51D}). Although p-eIF2 α levels of both endogenous and exogenous HA-eIF2 α^{wt} were reduced upon depletion of eIFB2 α , overall eIF2 α phosphorylation was still higher compared to empty cells (Fig. EV2A; Word file: page 9, line 214ff; merged PDF file: page 9, line 218ff). These increased p-eIF2 α levels did not rescue the proliferation defect elicited by eIFB2 α depletion (Fig. EV2B). Exogenous eIF2 α^{S51A} cannot be thoroughly phosphorylated (Fig. EV2A), and it competes with endogenous eIF2 α to generate phospho-dead eIF2 α^{S51A} -containing eIF2/eIF2B complexes over time. Still, this did not impact on cellular viability, either with or without eIF2B α being present (Fig. EV2B). Likewise, overexpression of the phospho-mimicking S51D mutant did not revert the viability defect (Fig. EV2C,D).

2) Additionally, to investigate the effects of eIF2B disruption by mutation of eIF2B α where p-eIF2 α is induced, we performed experiments with siRNA-mediated depletion of GADD34, the regulatory subunit of PP1, which is the relevant phosphatase for p-eIF2 α . In eIF2B α -depleted cells with or without expression of the eIF2B α mutant, phosphorylation of eIF2 α was restored upon depletion of GADD34 confirming that GADD34 promotes eIF2 α dephosphorylation in a context when p-eIF2 α is not bound by the eIF2B complex (Figure EV2E,F; Word file: page 10, line 228ff; merged PDF file: page 10, line 232ff). Nevertheless, restoration of p-eIF2 α levels by knockdown of GADD34 in the context of a disrupted eIF2B complex did not rescue the viability defect (Figure EV2G,H).

These sets of experiments suggest that the levels of p-eIF2 α themselves are not the driver for the observed cellular phenotype but rather the inability of the disrupted eIF2B complex (by either depletion or mutation of eIF2B α) to sense p-eIF2 α .

2. The study does not clearly establish whether the effects of eIF2B downregulation on the inhibition of p-eIF2 are specific to CRC tumors. For instance, eIF2Balpha knockdown does not seem to have detectable effects on p-eIF2 in K562 cells, as reported in Schoof et al. eLife 2021;10:e65703.

We assume that the effect of eIF2B downregulation on p-eIF2 α is highly dependent on the steady-state levels of phosphorylated eIF2 α in the respective tumor cell type. As we demonstrate in Fig. 1, CRC tumors harbor greatly elevated p-eIF2 α levels per se, whereas K562 cells used in the cited publication seem to have much lower or almost undetectable levels of eIF2 α phosphorylation (Schoof et al, 2021) (Fig. 3A, see below). In addition, having a very close look at the presented Western blot below it confirms our data: 1) Strong degradation of eIF2B α (28 and 83 nM dTag13) further reduces the per se low levels of p-eIF2 α . 2) This reduction in p-eIF2 α correlates with induction of ATF4. 3) Degradation of eIF2B α does not affect the stability of the eIF2B δ subunit. In other solid tumor entities, enhanced levels of eIF2 α phosphorylation have been described pointing to the possibility that modulation of the eIF2B complex might also induce a vulnerability in other types of cancer (Bai et al., 2021; Ghaddar et al., 2021; Koromilas, 2015; Shin et al., 2022). We elaborated on this in the discussion of the revised manuscript (Word file: page 19, line 468ff; merged PDF file: page 20, line 476ff). Additionally, we analyzed the effects of eIF2B α modulation in a pancreatic cancer setting by using the PDAC cell line PaTu8988T, validating that interfering with the function of the eIF2B complex

to sense p-eIF2 α by depletion or mutation of eIF2B α indeed reduces the viability of PDAC cells in a similar manner as observed in CRC (Appendix Fig. S2B,C; Word file: page 9, line 205ff; merged PDF file: page 9, line 208ff).

Schoof *et al*, 2021, Figure 3A

3. As a final note, while experiments with cultured cells indicate the inability of ISRIB to impair the proliferation of CRC cells (Figure 1), there have been numerous studies, nicely cited in the manuscript, demonstrating the anti-tumor effects of ISRIB in mouse models of cancer. It might be of interest to compare the growth of CRC tumor cells shown in Fig. 3B or 4B with the growth of parental cells treated with ISRIB in mice. This comparison could help determine whether targeting eIF2B α or eIF2B δ is indeed a superior approach to ISRIB treatment for inhibiting CRC tumor growth.

We understand the importance of mouse models for the evaluation of new potential anti-cancer strategies. Nevertheless, 3D organoid models have been widely established as alternative models that greatly resemble the respective primary organ and are suitable for testing potential treatments for various diseases including cancer (Berkers *et al*, 2019; Lo *et al*, 2020). More specifically, intestinal organoid models were the first 3D models to be established which have been characterized in detail and are widely used for pre-clinical investigations of anti-tumor treatments (Farin *et al*, 2023; Kastner *et al*, 2021; Sato *et al*, 2011; Sato *et al*, 2009). According to the 3R (reduction, refinement, replacement) commitment we aim at reducing and replacing animal experiments whenever possible. Therefore, we have invested a lot of effort to characterize the anti-tumor effects of targeting especially eIF2B α in a variety of different *in vitro* models and conducted the following additional experiments to strengthen our conclusions:

1. As 2D model we use the APC-mutated SW480 CRC cell line. We extended this analysis of eIF2B subunit depletion to one more APC-mutated CRC cell line, DLD1, which show comparable reduction in cell viability after depletion of the respective eIF2B subunits (Figure EV1B-E; Word file: page 7, line 170ff; merged PDF file: page 7, line 173ff). More specifically, eIF2B δ knockdown induced high percentage of cell death, whereas eIF2B α depletion only affected cell cycle progression, which is similar to the effects observed in SW480 cells. Furthermore, we repeated previous Fig. 3E with ISRIB treatment as additional condition to provide a side-by-side comparison validating that eIF2B α modulation might indeed be a superior approach to ISRIB (new Fig. 3F; Word file: page 8, line 200ff; merged PDF file: page 9, line 204ff).
2. As 3D model, we use three different mouse intestinal organoid lines each with a genetically defined background harboring the most common mutations in CRC (A, AK, LAKTP), and showed that LAKTP organoids were not affected in their viability by ISRIB treatment (previous Fig. 1F,G). We extended this analysis and treated also murine A and AK organoids with ISRIB which also did not show any viability reduction by ISRIB treatment (Fig. 1F,G; Word file: page 6, line 132ff; merged PDF file: page 6, line 133ff). This is in contrast to our eIF2B α depletion studies, where knockdown of eIF2B α by two independent shRNAs resulted in decreased

viability of all three murine tumor organoids, whereas WT organoids were not affected (Fig. 6A-D and Fig. EV5A-D; Word file: page 18, line 439ff; merged PDF file: page 18, line 446ff).

3. As additional 3D model with high therapeutic relevance, we use three different human APC-mutated PDOs (FAP, T4, HD-3). To underscore our findings that targeting eIF2B α is a superior approach to ISRIB treatment in this human setting, we also treated T4, FAP and HD-3 PDOs with ISRIB which did not impair the growth of all human PDOs (Fig. 1F,G; Word file: page 6, line 136ff; merged PDF file: page 6, line 137ff). Instead, all tumor PDOs displayed decreased viability upon eIF2B α depletion by two independent shRNAs (Fig. 6E-H and Fig. EV5E-H, Word file: page 18, line 447ff, merged PDF file: page 18, line 454ff). We additionally included one human WT organoid from normal colon mucosa (WT Ko165), that was not affected by knockdown of eIF2B α (Fig. 6E-H).

Minor comment:

1. It would be of interest to determine whether the GEF activity of eIF2B indeed plays a role in the reduction of p-eIF2 by the disruption of eIF2B complex. Figure 2 would be better with inclusion of eIF2Bepsilon KD in parallel with other subunits. Can the authors test the effects of a catalytic mutant eIF2Bepsilon in a similar approach shown in Figure 3B or 4B? It would be great.

We characterized the effects of eIF2B ϵ knockdown extensively in our previous publication (Schmidt *et al.*, 2019). Depletion of eIF2B ϵ reduced p-eIF2 α levels, CRC cell viability as well as induced ISR genes similarly to depletion of the other subunits. Therefore, we did not include it in the recent manuscript to avoid being repetitive. We refer to our previous publication in the manuscript (page 6, line 148ff). Regarding Fig. 2A, we also observed the decrease of the levels of the other subunits upon eIF2B ϵ knockdown, which is consistent with previous publications (Reply Fig. 1) (Wang *et al.*, 2012; Wortham *et al.*, 2016).

To address the question whether the GEF activity of eIF2B plays a role in reduction of p-eIF2 α by disruption of the eIF2B complex, we overexpressed HA-tagged versions of either wildtype eIF2B ϵ (eIF2B ϵ^{wt}) or an eIF2B ϵ R113H mutant (eIF2B ϵ^{R113H}), which is known to have reduced GEF activity (Fogli *et al.*, 2004; Li *et al.*, 2004). We depleted endogenous eIF2B ϵ in parallel and analyzed p-eIF2 α levels and also cell viability. As observed previously, knockdown of eIF2B ϵ led to decreased p-eIF2 α levels and cell numbers ((Schmidt *et al.*, 2019); Appendix Fig. S5A-D; Word file: page 12, line 302ff; merged PDF file: page 13, line 306ff). Expression of both eIF2B ϵ^{wt} and mutant eIF2B ϵ^{R113H} restored the phosphorylation of eIF2 α and viability of SW480 cells (Appendix Fig. S5A-D), suggesting that the GEF function of the eIF2B complex is not involved in the reduction of p-eIF2 α but it is solely dependent on disruption of the eIF2B complex.

Referee #3:

In their study, Skapik et al. propose that the targeting of eIF2B subunits, in particular eIF2B1 provides a potential therapeutic strategy for the treatment of CRC, especially when APC-deficient. On the remarkable fact that APC-deficient CRC is somewhat addicted to elevated eIF2a phosphorylation, the authors focus on the eIF2B complex, which is essential for translation initiation and the sensing of eIF2alpha phosphorylation. Two subunits, eIF2B1 and eIF2B4, are thought to sense this phosphorylation. By applying a transient depletion recovery analysis the authors claim that manipulating the role of both the factors by abundance or mutation of the phosphorylation sensing surface provides a vulnerability of CRC-derived organoids. Although the sensing function is not monitored directly, the viability studies support at least a pivotal role of eIF2B1/4 in the process. Settling on this, the authors proceed with exploring distinct translation deregulation by depleting eIF2B1 vs 4. These studies suggest that mRNAs encoding factors involved in promoting proliferation are particularly susceptible to manipulating eIF2B1. In sum this leads the authors to propose that this may provide a therapeutic window for CRC treatment. In sum the study provides interesting insights,

but various aspects require further investigations, more controls and need to address the major concerns raised in the following:

Major comments

1. Studies presented in Figs.1-4 mainly address the potentially distinct phenotypic roles of eIF2B subunits with a focus on eIF2B1/4. However, these studies do not provide substantial insights relevant for the remainder of the studies where only depletion studies are utilized. This is even more severe since the sensing of eIF2 α is not directly addressed and the connection between monitoring this modification and sensing remains largely elusive. Thus, despite a huge, admirable experimental effort, there is little to learn from studies presented in Figs.2-4.

In our study we utilized different ways to interfere with the sensing of p-eIF2 α , which we explained more comprehensively in the revised text (Word file: page 13, line 316ff; merged PDF file: page 13, line 320ff): Depletion or mutation of eIF2B α , mutation of eIF2B δ and depletion of one of the eIF2B $\beta\gamma\delta$ subunits. In general, assembly of the full eIF2B decamer is essential for sensing p-eIF2 α (Kashiwagi *et al*, 2019; Kenner *et al*, 2019). Therefore, we recognized that especially depletion of eIF2B α mimics the eIF2B α mutant phenotype, and thus the p-eIF2 α non-sensing phenotype, since with both types of modulation (depletion or mutation) the complete eIF2B decamer cannot be assembled properly as shown in the sucrose gradients analysis (Appendix Fig. S3; Word file: page 10, line 238ff; merged PDF file: page 10, line 242ff). Furthermore, also mutation of eIF2B δ has comparable effects to depletion and mutation of eIF2B α . Instead, the loss of eIF2B δ had strikingly different phenotypic outcomes.

To connect the mutation and depletion studies better, and to address the sensing of p-eIF2 α more directly, we conducted the following experiments:

1) We overexpressed HA-tagged versions of eIF2B α (wt and mutant) and performed co-immunoprecipitations of endogenous eIF2 α with the two overexpressed eIF2B α forms. eIF2B α^{wt} interacts with immunoprecipitated endogenous eIF2 α whereas no interaction was observed between eIF2B α^{mut} and eIF2 α (Fig. 3H, Word file: page 10, line 247ff; merged PDF file: page 10, line 249ff), clearly validating that eIF2B α^{mut} is not able to bind eIF2 α , suggesting also an inability to sense the phosphorylated form of eIF2 α .

2) To investigate the role of p-eIF2 α in the context of eIF2B disruption by depletion of eIF2B α more precisely, we performed experiments with SW480 cells stably overexpressing HA-tagged eIF2 α WT (eIF2 α^{wt}), the phospho-dead eIF2 α S51A mutant (eIF2 α^{S51A}) or the phospho-mimicking eIF2 α S51D mutant (eIF2 α^{S51D}). Although p-eIF2 α levels of both endogenous and exogenous HA-eIF2 α^{wt} were reduced upon depletion of eIF2B α in eIF2 α^{wt} expressing cells, overall eIF2 α phosphorylation was still higher compared to empty cells (Fig. EV2A; Word file: page 9, line 214ff; merged PDF file: page 9, line 218ff). These increased p-eIF2 α levels did not rescue the proliferation defect elicited by eIF2B α depletion (Fig. EV2B). Exogenous eIF2 α^{S51A} cannot be thoroughly phosphorylated (Fig. EV2A), and it competes with endogenous eIF2 α to generate phospho-dead eIF2 α^{S51A} -containing eIF2/eIF2B complexes over time. Still, this did not impact on cellular viability, either with or without eIF2B α being present (Fig. EV2B) Likewise, overexpression of the phospho-mimicking S51D mutant did not revert the viability defect (Fig. EV2C,D). Additionally, to investigate the effects of eIF2B disruption by mutation of eIF2B α where p-eIF2 α is induced, we performed experiments with siRNA-mediated depletion of GADD34, the regulatory subunit of PP1, which is the relevant phosphatase for p-eIF2 α . In eIF2B α -depleted cells with or without expression of the eIF2B α mutant, phosphorylation of eIF2 α was restored upon depletion of GADD34 confirming that GADD34 promotes eIF2 α dephosphorylation in a context when p-eIF2 α is not bound by the eIF2B complex (Figure EV2E,F; Word file: page 10, line 228ff; merged PDF file: page 10, line 232ff). Nevertheless, restoration of p-eIF2 α levels by knockdown of GADD34 in the context of a disrupted eIF2B complex did not rescue the viability defect (Figure EV2G,H).

3) We already showed that typical ISR genes (*DDIT3*, *PPP1R15A*) are among the translationally upregulated genes upon eIF2B α and eIF2B δ depletion (Appendix Fig. S6B,C; Word file: page 15, line 368ff; merged PDF file: page 15, line 373ff). Both *DDIT3* (CHOP) and *PPP1R15A* (GADD34) are also upregulated on protein level with eIF2B α^{mut} and eIF2B δ^{mut} expression (Fig. 3D, 4D and Appendix Fig. S7A,B). Additionally, we analyzed the Ribo-seq data in more detail and found *CBX4* as the most upregulated gene at the RPF level upon depletion of eIF2B δ . Importantly, we detected an increase in *CBX4* expression also in eIF2B δ^{mut} cells upon eIF2B δ knockdown (Appendix Fig. S7A). This indicates that the mutated subunits are included in eIF2B complexes rendering them incapable to sense p-eIF2 α . *CBX4* has also previously been shown to be upregulated upon ISR induction (Lee *et al*, 2008; Sikalidis *et al*, 2011). This reinforces the concept that mutations rendering decameric eIF2B insensitive to p-eIF2 α in a context where its phosphorylation status is elevated, activate an incomplete ISR.

Concerning the effects of eIF2B α modulation, we focused on hallmarks gene sets which were translationally repressed, selecting genes which were preferentially regulated by induction of sh*EIF2B1* only. Among these, the HNRNPs group of proteins, RNA-binding proteins mainly involved in RNA metabolism, was highly represented and strongly downregulated at the translational level. Indeed, HNRNPD (also called AUF1) and HNRNPA3 were decreased by both depletion and mutation of eIF2B α , but their levels were restored upon eIF2B α^{wt} expression (Appendix Fig. S7C, Word file: page 15, line 376ff; merged PDF file: page 15, line 381ff). Several HNRNP proteins are involved in p-eIF2 α -mediated stress granule formation, and HNRNPD, among others, is found to be downregulated in eIF2B-mutant VWM patient samples (Huyghe *et al*, 2012; Wu *et al*, 2014).

Overall, these data clearly show a relation between eIF2B depletion and mutation studies and validate that the observed effects are due to the p-eIF2 α non-sensing characteristic of the disrupted eIF2B complex.

2. A general problem of the study is that only one shRNA is used for depletion studies. Thus, how can the authors exclude - except for recovery studies in Figs.2-4 but not later (!) - bias by off-target effects. Along these lines, the authors demonstrate that depleting eIF2B2-4 disrupt the stability of other factors except eIF2B1 (Fig.2). This essentially limits the conclusiveness of findings and raises the question why this is not seen for eIF2B1 knockdown. Moreover, studies presented after Fig.4 require at least one additional, independent shRNA.

For initial constitutive depletion experiments in SW480 cells, we used three to four independent shRNAs for eIF2B α and eIF2B δ , respectively. We could validate the decrease in viability for all shRNAs against eIF2B α and for two out of four shRNAs against eIF2B δ , which correlated very well with knockdown efficiency making off-target effects unlikely (Reply Fig. 2A-D). For all subsequent mechanistic analysis, we chose one shRNA each that showed a strong knockdown (sh*EIF2B1.3* and sh*EIF2B4.2*). These shRNAs were also used in another CRC cell line, DLD1, showing similar cellular and molecular effects as in SW480 cells (Fig. EV1B-E; Word file: page 7, line 170ff; merged PDF file: page 7, line 173ff). Finally, we could validate the reduction in p-eIF2 α as well as in cell viability with sh*EIF2B1.3* in a different cancer cell type, for which we used the PDAC cell line PaTu8988T (Appendix Fig. S2B,C, Word file: page 9, line 205ff, merged PDF file: page 9, line 208ff). These effects were also rescued by overexpression of eIF2B α^{wt} (Appendix Fig. S2B,C), comparable to the recovery studies in SW480 cells, making off-target effects highly unlikely.

Moreover, in preparation for the Riboseq studies, where DOX-inducible shRNAs were used, we tested two independent shRNAs for each eIF2B α and eIF2B δ . Both shRNAs for each subunit showed similar depletion efficiencies and we could validate the decrease in p-eIF2 α and the increase in ISR protein ATF4 for all shRNAs (Reply Fig. 2E,F). Riboseq preparation and analysis is very time-consuming and needs a big amount of cellular material, therefore we decided to continue with only one shRNA for each subunit. Nevertheless, we performed a transcriptome-wide analysis to search for motifs complementary to the seed-sequences of the shRNAs used. We searched for the strongest possible seed-match, called an 8mer. Please see below Fig. 1A (bottom panel) and Fig. 1B (right panel) from Grimson *et al*, 2007, which describe the features of an 8mer site as well as the effect that a single 8mer (or other) site in the 3'UTR of a transcript has on its level (Grimson *et al*, 2007).

Grimson et al, 2007, Figure 1A,B

In our case, instead of the microRNA sequence, we considered the inducible shRNA sequences for *EIF2B1*, *EIF2B4*, and CTRL. Thus, we searched for the 8mer sites in both coding sequences (CDSs) and 3’UTRs and checked for their effect on total RNA levels (Reply Fig. 3). As it can be seen by the cumulative fraction distributions, the presence of 8mer sites – even when more than single ones – for the respective inducible shRNA has negligible effects, comparable to the ones which are caused by shCTRL. Thus, we conclude that the Ribo-seq data represents knockdown-specific events with virtually no off-target effects detected in these experimental conditions.

To address the concern regarding off-target effects of eIF2B α depletion in organoids, we repeated the experiments shown in Figure 6 with an additional independent shRNA against eIF2B α (see also response to comment 7). Comparable to the effects observed with the previously used mouse sh*Eif2b1-1* or human sh*EIF2B1-1*, both mouse sh*Eif2b1-2* and human sh*EIF2B1-2* reduced the viability and p-eIF2 α levels in tumor-derived organoids (mouse A, AK, LAKTP and human T4, FAP, HD-3) (Fig. EV5A-H; Word file: page 18, line 439ff; merged PDF file: page 18, line 446ff). Only in AK organoids, no reduction in p-eIF2 α was observed with the second shRNA (Fig. EV5C). We extensively tried to infect murine WT organoids with sh*Eif2b1-2*, but we were not successful unfortunately. Therefore, we included one human WT organoid established from normal colon mucosa (Ko165), which, similar to murine WT organoids, did not show a reduction in viability upon knockdown of eIF2B α (Fig. 6E-H; Word file: page 18, line 454ff; merged PDF file: page 18, line 461ff), strongly arguing against off-target effects.

Finally, the decreased stability of the eIF2B subunits upon depletion of eIF2B $\beta\gamma\delta$, but not eIF2B α , is well established in the literature and we explain and cite this accordingly (Word file: page 7, line 153ff; merged PDF file: page 7, line 156ff). We did not only observe this in SW480 cells (Fig. 2A), but also in DLD1 cells (Fig. EV1B). Additionally, we performed the same experiments in another tumor cell line, Hela, by using siRNAs. For this, we used siRNA pools, each consisting of four independent siRNAs to reduce off-target effects. We could validate the observed effects on the stability of the different eIF2B subunits also in Hela cells (Reply Fig. 4A,B).

3. In Fig.3B the mutant clearly shows lower expression. This could as well explain modestly affected (reduced by ~40%) eIF2alpha phosphorylation. Moreover, errors are missing in the quantification of eIF2alpha phosphorylation, which should be assessed together with monitoring changes in the overall protein abundance of eIF2alpha.

The eIF2B α mutant is not able to form an eIF2B α homodimer and single eIF2B α subunits are highly unstable which could explain the observed lower expression. We also analyzed the effects of eIF2B α

modulation in a pancreatic cancer setting by using the PDAC cell line PaTu8988T. Overexpression levels of eIF2B α^{mut} and eIF2B α^{wt} are more similar compared to levels in SW480 cells (Appendix Fig. S2B; Word file: page 9, line 205ff; merged PDF file: page 9, line 208ff). Still, we observed the reduction in p-eIF2 α upon knockdown of eIF2B α and eIF2B α^{mut} expression (Appendix Fig. S2B,C).

To enhance the robustness of the data we quantified the levels of both p-eIF2 α and total eIF2 α in all respective biological replicates and provide graphs with error bars showing a consistent reduction in p-eIF2 α levels in all relevant experiments (Fig. 3E, 4E, 6D, 6H; Fig. EV5D,H).

4. Puzzling is why the authors do not present the same analyses in Fig.3/4, in particular altered proliferation rates (see Fig.3E) but only present the percentage of dead cells in Fig.4. These analyses require further in-depth analyses to evaluate how eIF2B1 vs 4 influence cell cycle progression, apoptosis, and organoid growth.

We replicated the growth curves shown in previous Fig. 3E for modulation of eIF2B δ (Fig. 4F; Word file: page 11, line 277ff; merged PDF file: page 12, line 281ff). The strong decrease in cell numbers by depletion of eIF2B δ could be partially restored by eIF2B δ^{wt} but not by eIF2B δ^{mut} expression (Fig. 4F). In Fig. EV3A, we also show in-depth cell cycle analysis via PI FACS for the modulation of eIF2B δ (as shown for eIF2B α modulation in Appendix Fig. S2A). This shows a major increase of dead cells by knockdown of eIF2B δ , reflected by the high fraction of cells in sub-G1, but no change in distribution of cell cycle phases (Fig. EV3A). Whereas eIF2B δ^{wt} abolished this induction of cell death, eIF2B δ^{mut} also moderately induced cell death comparable to the effects of eIF2B α depletion, which correlated with the Annexin V/PI FACS data (Fig. 4G and Fig. EV3A). With those detailed analyses (growth curve, PI cell cycle FACS, Annexin V/PI FACS) for both eIF2B α and eIF2B δ knockdown, we could clearly distinguish between two phenotypes: cell cycle arrest upon eIF2B α depletion and cell death upon eIF2B δ depletion.

5. The ribosomal profiling studies are very interesting but surprising given the fact that both factors are claimed essential for sensing eIF2alpha phosphorylation by prior presented studies and have strong phenotypic impact, at least on the number of dead cells. How come then that eIF2B4 depletion has comparatively marginal effect on translation? Is this potentially a problem of higher apoptosis and the lack of these cells in polysomal analyses?

As shown in Fig. 2B and Fig. EV3B, eIF2B δ depletion leads to a strong reduction in global translation of over 30%, whereas global translation is not reduced upon knockdown of eIF2B α . In addition, eIF2B δ depletion disrupts the entire eIF2B complex, while eIF2B α knockdown leaves eIF2B $\beta\gamma\delta\epsilon$ sub-complexes present in the cells. We hypothesize that this difference is the reason for which specific transcript-level effects can be identified with depletion of eIF2B α . Indeed, the bulk repression of translation by eIF2B δ removal causes unspecific effects which cannot be interpreted by a standard DEseq2 analysis.

Regarding the absence of cells in our sample, we are able to characterize the knockdown in the sequencing data (Table EV2, EIF2B4 RPFs log2FC: -0.48, cytoplasmic RNA log2FC: -0.85). Therefore, we believe the Ribo-seq results accurately represent eIF2B δ -depleted conditions.

6. The provided GSEA analyses (Fig.5) apparently support a role of disturbed translation on overall cell viability but how these data explain the strong impact of eIF2B4 on apoptosis (?) or viability remains unclear. For eIF2B1 depletion the studies appear reasonable, but some gene sets show remarkable differences, e.g. MYC Targets V1 vs V2, why? Unfortunately, the ribosome profiling studies at best describe observations which would require further in-depth testing of at least some candidate effectors claimed for eIF2B1/4 and need to connect more precisely to more detailed phenotypic characterizations.

Regarding the regulation of MYC V1 and V2 signatures, their differential enrichment at the translational level has been described, whereby knockout of two translation initiation factors (eIF4A1 and eIF4A2) oppositely regulates only V1 (Waldron *et al.*, 2023). The two signatures are themselves strikingly different: V1 is composed of 200 genes, V2 is composed of 58 genes, and only 18 are shared by both. We performed an analysis of the overlap between either of these signatures and the well-characterized gene ontology (GO) sets on MSigDB. This shows that among the top20 scoring GO signatures overlapping with V1 or V2, only four are shared (Table screenshot below, colored lines). Whilst we acknowledge that there is some redundancy in terms, this analysis highlights that the two signatures are describing specific programs. These are known to be downstream of MYC for their transcription, as it has been well characterized. However, we posit that they are not equally regulated at the translational level.

Top20 GO MycV1 sets	Top20 GO MycV2 sets
GOCC_RIBONUCLEOPROTEIN_COMPLEX	GOBP_RIBOSOME_BIOGENESIS
GOBP_RIBONUCLEOPROTEIN_COMPLEX_BIOGENESIS	GOBP_RIBONUCLEOPROTEIN_COMPLEX_BIOGENESIS
GOCC_NUCLEAR_PROTEIN_CONTAINING_COMPLEX	GOBP_RRNA_METABOLIC_PROCESS
GOCC_CATALYTIC_COMPLEX	GOBP_NCRNA_METABOLIC_PROCESS
GOBP_PEPTIDE_BIOSYNTHETIC_PROCESS	GOCC_NUCLEOLUS
GOBP_DNA_METABOLIC_PROCESS	GOBP_NCRNA_PROCESSING
GOBP_PEPTIDE_METABOLIC_PROCESS	GOCC_PRERIBOSOME
GOBP_AMIDE_BIOSYNTHETIC_PROCESS	GOBP_RNA_PROCESSING
GOBP_ORGANONITROGEN_COMPOUND_BIOSYNTHETIC_PROCESS	GOBP_RIBOSOMAL_LARGE_SUBUNIT_BIOGENESIS
GOBP_RNA_SPLICING_VIA_TRANSESTERIFICATION_REACTIONS	GOBP_RIBOSOMAL_SMALL_SUBUNIT_BIOGENESIS
GOBP_AMIDE_METABOLIC_PROCESS	GOCC_SMALL_SUBUNIT_PROCESSOME
GOBP_MRNA_METABOLIC_PROCESS	GOCC_PRERIBOSOME_LARGE_SUBUNIT_PRECURSOR
GOBP_RNA_PROCESSING	GOCC_RIBONUCLEOPROTEIN_COMPLEX
GOBP_MRNA_PROCESSING	GOBP_MATURATION_OF_SSU_RRNA
GOBP_RNA_SPLICING	GOCC_90S_PRERIBOSOME
GOCC_INTRACELLULAR_PROTEIN_CONTAINING_COMPLEX	GOCC_CHROMOSOME
GOBP_CYTOPLASMIC_TRANSLATION	GOMF_CATALYTIC_ACTIVITY_ACTING_ON_A_NUCLEIC_ACID
GOBP_REGULATION_OF_DNA_METABOLIC_PROCESS	GOMF_ADENYL_NUCLEOTIDE_BINDING
GOBP_CHROMOSOME_ORGANIZATION	GOMF_PURINE_NUCLEOTIDE_BINDING
GOCC_SPLICEOSOMAL_COMPLEX	GOBP_DNA_METABOLIC_PROCESS

Table screenshot of MYC V1 and V2 gene sets GO analysis

As addressed in comment 1, part 3, we already showed that typical ISR genes (*DDIT3*, *PPP1R15A*) are among the translationally upregulated genes upon eIF2B δ depletion (Appendix Fig. S6B,C; Word file: page 15, line 368ff; merged PDF file: page 15, line 373ff). Additionally, we analyzed the Ribo-seq data in more detail and found *CBX4* as the most upregulated gene at the RPF level upon depletion of eIF2B δ . *CBX4* was also shown to be upregulated via the ISR (Lee *et al.*, 2008; Sikalidis *et al.*, 2011), and induction of high levels of ISR, with involvement of CHOP especially, can lead to cell death. We assume that activated ISR signaling is involved in the apoptotic phenotype of eIF2B δ -depleted cells. All three proteins (CHOP, GADD34, *CBX4*) are also upregulated upon eIF2B δ^{mut} expression, but not upon eIF2B δ^{wt} expression (Fig. 3D, 4D and Appendix Fig. S7A,B), supporting the idea that sensing p-eIF2 α is indeed crucial for CRC cells survival.

Regarding the effect of eIF2B α depletion, we believe that very strong evidence of the cell cycle effect is the almost complete downregulation at the translational level of all the genes belonging to the hallmark G2M transition. Whilst this can be already appreciated from the scatter plots showing the genes regulated (Appendix Fig. S6E), we add here an enrichment plot showing the effect on the whole gene set.

Furthermore, as addressed in comment 1, part 3, we focused on hallmarks gene sets which were translationally repressed (Fig. 5D,E), selecting genes which were preferentially downregulated by shEIF2B1 only. Among these, the HNRNP group of proteins, RNA-binding proteins mainly involved in RNA metabolism, was highly represented. Indeed, HNRNPD (also called AUF1) and HNRNPA3 were decreased by both depletion and mutation of eIF2B α (Appendix Fig. S7C; Word file: page 15, line 376ff; merged PDF file: page 15, line 381ff). Several HNRNP proteins are involved in p-eIF2 α -mediated stress granule formation, and HNRNPD, among others, is found to be downregulated in eIF2B-mutant VWM patient samples (Huyghe *et al.*, 2012; Wu *et al.*, 2014; Zhou *et al.*, 2024), suggesting a contribution of these proteins in mediating the eIF2B/p-eIF2 α non-sensing phenotype.

7. How the data presented in Fig.6 suggest eIF2B1 as a candidate target for CRC treatment remains vague for various reasons. (1) At least two shRNAs should be used for these studies; (2) the depletion of eIF2B4 should be used as a control since this would be expected to greatly impact on viability as well; (3) the change in viability on A, AK, LAKTP models versus wild type is obvious but could be explained by substantial distinct growth rates - also note the huge errors in the wild type population; (4) Major concerns are raised by Fig.6C in the AK model where barely any depletion of eIF2B1 is observed yet strong effects are reported; (5) finally, if eIF2B1 is a promising therapeutic target one would expect to have tests on other none CRC-derived cells (in addition to WT organoids) to exclude a broad toxicity of eIF2B1 targeting.

To address the mentioned concerns and to strengthen our hypothesis of eIF2B α as candidate target for CRC treatment we conducted the following experiments:

(1) As already explained in the response to comment 2, we repeated the experiments shown in Figure 6 with an additional independent shRNA against eIF2B α (see also response to comment 7). Comparable to the effects observed with the previously used mouse shEif2b1-1 or human shEIF2B1-1, both mouse shEif2b1-2 and human shEIF2B1-2 reduced the viability and p-eIF2 α levels in tumor-derived organoids (mouse A, AK, LAKTP and human T4, FAP, HD-3) (Fig. EV5A-H; Word file: page 18, line 439ff; merged PDF file: page 18, line 446ff). Only in AK organoids, no reduction in p-eIF2 α was observed with the second shRNA (Fig. EV5C). We extensively tried to infect murine WT organoids with shEif2b1-2, but we were not successful unfortunately. Therefore, we included one human WT organoid established from normal colon mucosa (Ko165), which, similar to murine WT organoids, did not show a reduction in viability upon knockdown of eIF2B α (Fig. 6E-H; Word file: page 18, line 454ff; merged PDF file: page 18, line 461ff), strongly arguing against off-target effects.

(2) We additionally depleted eIF2B δ in WT, LAKTP murine organoids as well as in T4 human PDO (Appendix Fig. S9C-H; Word file: page 18, line 457ff; merged PDF file: page 18, line 465ff). Knockdown of eIF2B δ reduced not only the viability of all tumor-derived organoids, but also of murine WT organoids validating the rather toxic cellular effects of eIF2B δ depletion.

(3) Regarding the big error bars in viability of murine WT organoids (Fig. 6B), we repeated the experiment of eIF2B α depletion with sh*Eif2b1-1* in murine WT organoids to carefully check the robustness of the data. The repeated experiment did not show this high standard deviation and, thus, we exchanged the respective graph (Fig. 6B). Regarding substantial distinct growth rates, we analyzed this carefully by comparing the growth rates of WT, A, AK and LAKTP organoids via live cell imaging (Incucyte analysis). Both AK and LAKTP organoids have a higher growth rate compared to WT, but also compared to A organoids (Reply Fig. 5A). Indeed, A organoids show a similarly slow growth rate as WT organoids (Reply Fig. 5A), but show reduced viability upon eIF2B α depletion (Fig. 6A-D). In addition, we conducted this growth analysis for all analyzed PDOs. Tumor PDOs T4, FAP and HD-3, that showed reduced viability upon knockdown of eIF2B α (Fig. 6E-H), display a comparable and even slower growth behaviour compared to WT PDO Ko165, which was not affected by eIF2B α depletion (Reply Fig. 5B, Fig. 6E-H). This suggests that the effects of eIF2B α depletion on tumor organoids are rather tumor-specific than due to higher growth rates of those.

(4) We repeated this experiment in AK organoids and carefully checked all replicates for knockdown efficiency. Accordingly, we exchanged the Western blot for another replicate showing a more obvious knockdown of eIF2B α in AK organoids (Fig. 6C). To more directly address the raised concern and to increase robustness of the data, we provided quantification of eIF2B α levels (together with p-eIF2 α and total eIF2 α levels) with corresponding errors for all replicates in all organoid lines analyzed (Fig. 6D,H and Fig. EV5D,H).

(5) As introduced in the manuscript (Word file: page 17, line 430ff; merged PDF file: page 17, line 437ff), data from yeast and mammalian cells already describe the non-essential role of eIF2B α (Elsby *et al.*, 2011; Hannig & Hinnebusch, 1988). Specifically, MEFs depleted of eIF2B α did not show any significant viability defect, which was also not observed in Hela cells (Elsby *et al.*, 2011). Thus, we argue that the effects of eIF2B α modulation are highly tumor-specific. Additionally, we have already included analysis of DepMap data in the recent manuscript showing that eIF2B α is the least essential among all other analyzed subunits (Appendix Fig. S9A). To strengthen this aspect, we used one human colon organoid line from non-CRC-derived tissue (Ko165) and depleted eIF2B α . We were able to deplete eIF2B α efficiently in these organoids, but knockdown did not have any effect on the viability (Fig. 6E-H; Word file: page 18, 454ff; merged PDF file: page 18, line 461ff).

Reply Figure 1

Western blot of indicated proteins in SW480 cells transduced with shCTRL or shRNAs against *EIF2B1-5*, representative of three biological replicates with similar results.

Reply Figure 2

(A) Western blot of indicated proteins in SW480 cells transduced with shCTRL or three independent shRNAs against *EIF2B1*, representative of three biological replicates with similar results.

(B) Crystal violet staining of SW480 cells transduced as described in (A), representative of two biological replicates with similar results.

(C) Western blot of indicated proteins in SW480 cells transduced with shCTRL or four independent shRNAs against *EIF2B4*, representative of three biological replicates with similar results.

(D) Crystal violet staining of SW480 cells transduced as described in (C), representative of two biological replicates with similar results.

(E) Western blots of inducible shRNA clones (1x for shCTRL, 2x for sh*EIF2B1*, and 2x for sh*EIF2B4*). Blots representative of four biological replicates, performed in parallel to harvesting cells for the actual Ribo-seq experiment. Quantification of the most important functional downstream effects (induction of ATF4 and reduction of relative p-eIF2 α levels) are also included. Quantification is presented as a normalized ratio between each induced vs. non-induced shRNA.

A shRNA seed matches searched in the Coding sequences (CDS)

B shRNA seed matches searched in the 3' UTR sequences

Reply Figure 3

Cumulative distribution functions of fold changes (FC) in total RNA levels (in log₂ scale) for the experimental conditions described on each x-axis.

For each plot, colored lines represent how many 8mer sites can be found in (A) the coding sequence (CDS) or in (B) the 3' untranslated region (3' UTR) of each gene. The yellow line always represents

the distribution of genes without any 8mer site. The legend also indicates the number of genes per group ($n =$), and the mean \log_2FC of the group (mean =).

Left column - plots represent the presence of 8mer(s) for sh*EIF2B1* (first row) or sh*EIF2B4* (second row). Right column - the same fold changes in total RNA levels are displayed as the panel on the left. However, the genes are grouped depending on the presence of 8mer(s) for the shCTRL sequence, and thus serve as control.

Reply Figure 4

(A) Western blot of indicated proteins in HeLa cells transfected with 5 nM siCtrl or the indicated siRNAs against *EIF2B1-5* for 48 hrs, representative of three biological replicates with similar results.

(B) Quantification of eIF2B subunits levels from blots described in (A). Data show mean \pm s.e.m. of the three biological replicates performed.

Reply Figure 5

(A) Growth curve of murine WT, A, AK, LAKTP organoids. Organoids were seeded and live cell imaging via Incucyte[®] was performed for 120 hrs. Data show mean \pm s.d. ($n = 4$ biological replicates); Student's t -test.

(B) Growth curve of human WT Ko165 organoids and T4, FAP, HD-3 PDOs. Organoids were seeded and live cell imaging via Incucyte[®] was performed for 168 hrs. Data show mean \pm s.d. ($n = 4$ biological replicates); Student's t -test.

References

- Bai X, Ni J, Beretov J, Wasinger VC, Wang S, Zhu Y, Graham P, Li Y (2021) Activation of the eIF2alpha/ATF4 axis drives triple-negative breast cancer radioresistance by promoting glutathione biosynthesis. *Redox Biol* 43: 101993
- Berkers G, van Mourik P, Vonk AM, Krusselbrink E, Dekkers JF, de Winter-de Groot KM, Arets HGM, Marck-van der Wilt REP, Dijkema JS, Vanderschuren MM *et al* (2019) Rectal Organoids Enable Personalized Treatment of Cystic Fibrosis. *Cell Rep* 26: 1701-1708 e1703
- Elsby R, Heiber JF, Reid P, Kimball SR, Pavitt GD, Barber GN (2011) The alpha subunit of eukaryotic initiation factor 2B (eIF2B) is required for eIF2-mediated translational suppression of vesicular stomatitis virus. *J Virol* 85: 9716-9725
- Farin HF, Mosa MH, Ndreshkjana B, Grebbin BM, Ritter B, Menche C, Kennel KB, Ziegler PK, Szabo L, Bollrath J *et al* (2023) Colorectal Cancer Organoid-Stroma Biobank Allows Subtype-Specific Assessment of Individualized Therapy Responses. *Cancer Discov* 13: 2192-2211
- Fogli A, Schiffmann R, Hugendubler L, Combes P, Bertini E, Rodriguez D, Kimball SR, Boespflug-Tanguy O (2004) Decreased guanine nucleotide exchange factor activity in eIF2B-mutated patients. *Eur J Hum Genet* 12: 561-566
- Ghaddar N, Wang S, Woodvine B, Krishnamoorthy J, van Hoef V, Darini C, Kazimierczak U, Ah-Son N, Popper H, Johnson M *et al* (2021) The integrated stress response is tumorigenic and constitutes a therapeutic liability in KRAS-driven lung cancer. *Nat Commun* 12: 4651
- Grimson A, Farh KK, Johnston WK, Garrett-Engele P, Lim LP, Bartel DP (2007) MicroRNA targeting specificity in mammals: determinants beyond seed pairing. *Mol Cell* 27: 91-105
- Hannig EM, Hinnebusch AG (1988) Molecular analysis of GCN3, a translational activator of GCN4: evidence for posttranslational control of GCN3 regulatory function. *Mol Cell Biol* 8: 4808-4820
- Huyghe A, Horzinski L, Henaut A, Gaillard M, Bertini E, Schiffmann R, Rodriguez D, Dantal Y, Boespflug-Tanguy O, Fogli A (2012) Developmental splicing deregulation in leukodystrophies related to EIF2B mutations. *PLoS One* 7: e38264
- Kashiwagi K, Yokoyama T, Nishimoto M, Takahashi M, Sakamoto A, Yonemochi M, Shirouzu M, Ito T (2019) Structural basis for eIF2B inhibition in integrated stress response. *Science* 364: 495-499
- Kastner C, Hendricks A, Deinlein H, Hankir M, Germer CT, Schmidt S, Wiegering A (2021) Organoid Models for Cancer Research-From Bed to Bench Side and Back. *Cancers (Basel)* 13
- Kenner LR, Anand AA, Nguyen HC, Myasnikov AG, Klose CJ, McGeever LA, Tsai JC, Miller-Vedam LE, Walter P, Frost A (2019) eIF2B-catalyzed nucleotide exchange and phosphoregulation by the integrated stress response. *Science* 364: 491-495
- Koromilas AE (2015) Roles of the translation initiation factor eIF2alpha serine 51 phosphorylation in cancer formation and treatment. *Biochim Biophys Acta* 1849: 871-880
- Lee JI, Dominy JE, Jr., Sikalidis AK, Hirschberger LL, Wang W, Stipanuk MH (2008) HepG2/C3A cells respond to cysteine deprivation by induction of the amino acid deprivation/integrated stress response pathway. *Physiol Genomics* 33: 218-229
- Li W, Wang X, Van Der Knaap MS, Proud CG (2004) Mutations linked to leukoencephalopathy with vanishing white matter impair the function of the eukaryotic initiation factor 2B complex in diverse ways. *Mol Cell Biol* 24: 3295-3306
- Lo YH, Karlsson K, Kuo CJ (2020) Applications of Organoids for Cancer Biology and Precision Medicine. *Nat Cancer* 1: 761-773
- Peter S, Bultinck J, Myant K, Jaenicke LA, Walz S, Muller J, Gmachl M, Treu M, Boehmelt G, Ade CP *et al* (2014) Tumor cell-specific inhibition of MYC function using small molecule inhibitors of the HUWE1 ubiquitin ligase. *EMBO Mol Med* 6: 1525-1541

- Sato T, Stange DE, Ferrante M, Vries RG, Van Es JH, Van den Brink S, Van Houdt WJ, Pronk A, Van Gorp J, Siersema PD *et al* (2011) Long-term expansion of epithelial organoids from human colon, adenoma, adenocarcinoma, and Barrett's epithelium. *Gastroenterology* 141: 1762-1772
- Sato T, Vries RG, Snippert HJ, van de Wetering M, Barker N, Stange DE, van Es JH, Abo A, Kujala P, Peters PJ *et al* (2009) Single Lgr5 stem cells build crypt-villus structures in vitro without a mesenchymal niche. *Nature* 459: 262-265
- Schmidt S, Gay D, Uthe FW, Denk S, Paauwe M, Matthes N, Diefenbacher ME, Bryson S, Warrander FC, Erhard F *et al* (2019) A MYC-GCN2-eIF2alpha negative feedback loop limits protein synthesis to prevent MYC-dependent apoptosis in colorectal cancer. *Nat Cell Biol* 21: 1413-1424
- Schoof M, Boone M, Wang L, Lawrence R, Frost A, Walter P (2021) eIF2B conformation and assembly state regulate the integrated stress response. *Elife* 10
- Shin S, Solorzano J, Liauzun M, Pyronnet S, Bousquet C, Martineau Y (2022) Translational alterations in pancreatic cancer: a central role for the integrated stress response. *NAR Cancer* 4: zcac031
- Sikalidis AK, Lee JI, Stipanuk MH (2011) Gene expression and integrated stress response in HepG2/C3A cells cultured in amino acid deficient medium. *Amino Acids* 41: 159-171
- Waldron JA, Kanellos G, Smith RCL, Knight JRP, Munro J, Alexandrou C, Vlahov N, Pardo-Fernandez L, Moore M, Gillen SL *et al* (2023) eIF4A1 is essential for reprogramming the translational landscape of Wnt-driven colorectal cancers. *bioRxiv*: 2023.2011.2010.566546
- Wang X, Wortham NC, Liu R, Proud CG (2012) Identification of residues that underpin interactions within the eukaryotic initiation factor (eIF2) 2B complex. *J Biol Chem* 287: 8263-8274
- Wortham NC, Stewart JD, Harris S, Coldwell MJ, Proud CG (2016) Stoichiometry of the eIF2B complex is maintained by mutual stabilization of subunits. *Biochem J* 473: 571-580
- Wu S, Lin L, Zhao W, Li X, Wang Y, Si X, Wang T, Wu H, Zhai X, Zhong X *et al* (2014) AUF1 is recruited to the stress granules induced by coxsackievirus B3. *Virus Res* 192: 52-61
- Zhou Y, Panhale A, Shvedunova M, Balan M, Gomez-Auli A, Holz H, Seyfferth J, Helmstadter M, Kayser S, Zhao Y *et al* (2024) RNA damage compartmentalization by DHX9 stress granules. *Cell* 187: 1701-1718 e1728

Dear Dr. Wiegering,

Thank you for submitting your revised manuscript (EMBOJ-2024-116846R) to The EMBO Journal. Please accept my sincere apologies for the delay in getting back to you at this time, which is due to protracted referee input as well as detailed discussions in the editorial team. As mentioned, your amended study was sent back to the three referees for their scientific re-evaluation, and we have received feedback from one of them which I enclose below. As you will see, expert #2 states that the work has been substantially enhanced by the revisions and s/he is now in favour of publication, pending minor revision.

Please note that the two other referees got delayed and despite repeated chasers did not provide their recommendations. We have editorially assessed your response to the critique raised by reviewer #1 and found the issues to be addressed satisfactorily. Further, we have asked referee #2 to check your response to reviewer #3, and this expert found the initial critical points to be well addressed (please compare his-her additional comment below).

Thus, we are pleased to inform you that your manuscript has been accepted in principle for publication in The EMBO Journal.

Please consider the remaining point by referee #2 carefully and adjust the literature citations where appropriate.

We also now need you to take care of a number of issues related to formatting and data presentation as detailed below, which should be addressed at re-submission.

Please contact me at any time if you have additional questions related to below points.

Thank you for giving us the chance to consider your manuscript for The EMBO Journal. I look forward to your final revision.

Again, please contact me at any time if you need any help or have further questions.

Best regards,

Daniel Klimmeck

>> Author Contributions: Remove the author contributions information from the manuscript text. Note that CRediT has replaced the traditional author contributions section as of now because it offers a systematic machine-readable author contributions format that allows for more effective research assessment. and use the free text boxes beneath each contributing author's name to add specific details on the author's contribution.

More information is available in our guide to authors.
<https://www.embopress.org/page/journal/14602075/authorguide>

>> Data availability section: please remove the referee token for the Ribo-seq GEO data set and make sure that data privacy is released.

>> Remove the Reagents and Tools table from the manuscript and provide as a separate file using the existing template in the

Guide For Authors, listing key reagents, experimental models, software and relevant equipment.

>> Add complete annotation of animal husbandry -mouse ethics to the Methods and adjust the Author Checklist accordingly.

>> Please indicate redisplay of data from Fig EV5A in the figure legend of Appendix Figure S9C.

>> Dataset EV legends: Table EV1-EV3 should be renamed to Dataset EV1-EV3 with the corresponding callouts, legends correctly included as separate tabs in each Excel file, just the labels need to be updated. Dataset EV1 should be renamed to Table EV1.

>> Please recheck references for the bioRxiv entry Korotkevich et al. (2021) and update the citation if in the meantime published as regular article.

>> Consider additional changes and comments from our production team as indicated below:

- Data citations: no comments

- Figure Legends (main + EV): 1. Please indicate the statistical test used for data analysis in the legends of figures 5d-e.

2. Please note that the box plots need to be defined in terms of minima, maxima, centre, bounds of box and whiskers, and percentile in the legend of figure EV 4c.

3. Please note that the scale bar is missing for figure 1a.

4. Please note that the asterisk is not defined in the legend of figure EV 5c. This needs to be rectified.

Referee #2:

The authors conducted additional experiments and included data to improve their model on the role of eIF2B in CRC. The reviewer appreciates the complexity of the subject and acknowledges that the findings would be useful for investigating eIF2B function in cancer. While the study would greatly benefit from the inclusion of a mouse model, the reviewer accepts that this may be part of future experimental plans.

A minor suggestion: it would be helpful to cite PMID 31086176 in line 483 to provide stronger context on the anti-tumor effects of phosphorylated eIF2.

Referee #2, additional comment on response to authors' response to Referee #3's concerns:

I reviewed the responses to Reviewer #3 and found them very reasonable.

I feel that the manuscript should be accepted for publication.

The authors addressed the remaining editorial issues.

Dear Dr. Wiegering,

Thank you for submitting the revised version of your manuscript. I have now evaluated your amended manuscript and concluded that the remaining minor concerns have been sufficiently addressed.

I am thus pleased to inform you that your manuscript has been accepted for publication in the EMBO Journal.

Related, I would like to hereby ask your consent on keeping the referee figures included in this file.

On a different note, I would like to alert you that EMBO Press offers a format for a video-synopsis of work published with us, which essentially is a short, author-generated film explaining the core findings in hand drawings, and, as we believe, can be very useful to increase visibility of the work. Please see the following link for representative examples and their integration into the article web page:

<https://www.embopress.org/doi/full/10.15252/embj.2019103932>

Best regards,

Daniel Klimmeck

Daniel Klimmeck, PhD
Senior Editor
The EMBO Journal
EMBO
Postfach 1022-40
Meyerhofstrasse 1
D-69117 Heidelberg
contact@embojournal.org